DOI: 10.1038/s41467-017-02750-3　　**OPEN**

# Heuristic and optimal policy computations in the human brain during sequential decision-making

Christoph W. Korn[1,2,3] & Dominik R. Bach[1,2,4]

Optimal decisions across extended time horizons require value calculations over multiple probabilistic future states. Humans may circumvent such complex computations by resorting to easy-to-compute heuristics that approximate optimal solutions. To probe the potential interplay between heuristic and optimal computations, we develop a novel sequential decision-making task, framed as virtual foraging in which participants have to avoid virtual starvation. Rewards depend only on final outcomes over five-trial blocks, necessitating planning over five sequential decisions and probabilistic outcomes. Here, we report model comparisons demonstrating that participants primarily rely on the best available heuristic but also use the normatively optimal policy. FMRI signals in medial prefrontal cortex (MPFC) relate to heuristic and optimal policies and associated choice uncertainties. Crucially, reaction times and dorsal MPFC activity scale with discrepancies between heuristic and optimal policies. Thus, sequential decision-making in humans may emerge from integration between heuristic and optimal policies, implemented by controllers in MPFC.

---

[1] Division of Clinical Psychiatry Research, Department of Psychiatry, Psychotherapy, and Psychosomatics; Psychiatric Hospital, University of Zurich, Lengstrasse 31, 8032 Zurich, Switzerland. [2] Neuroscience Center Zurich, University of Zurich, Winterthurerstrasse 190, 8057 Zurich, Switzerland. [3] Institute for Systems Neuroscience, University Medical Center Hamburg-Eppendorf, Martinistrasse 52, 20246 Hamburg, Germany. [4] Wellcome Trust Centre for Neuroimaging, University College London, 12 Queen Square, London WC1N 3BG, United Kingdom. Correspondence and requests for materials should be addressed to C.W.K. (email: c.korn@uke.de)

In biological scenarios, decision-makers should normatively evaluate the long-term consequences of their actions with respect to appropriate reward or cost functions. Foraging for food, for example, requires sequential decision-making to avoid starvation: foraging choices should ensure that current and future metabolic reserves remain within homeostatic boundaries. Humans in particular can use their quite sophisticated—but bounded—decision-making capabilities to take multiple future outcomes into account[1–4]. For instance, when current conditions are bad, decision-makers need to determine whether to forage immediately, or to wait until conditions improve; if current internal energy resources are low, however, waiting may lead to starvation. Conceptually similar multi-step decision problems arise in many different real-world contexts, for example, in business decisions when investors have to balance immediate and delayed threats of bankruptcy.

Sequential decision-making requires searching over a tree of probabilistic future states. During foraging, avoiding starvation depends on the success of current and future foraging attempts, on the internal energy state of the decision-maker, as well as on momentary and expected foraging opportunities. A model-based tree search should take all these variables into account—with more branches to be evaluated the longer the considered time horizon. These intricate computations may exceed the capacities of the human brain and entail costs in terms of opportunities foregone due to the passage of time[5–7]. Decision-makers could simplify such quandaries by resorting to model-free heuristics[8, 9] and by restricting the set of considered options[10] and actions[11], in order to approximate optimal multi-step policies without full tree search. For example, foraging options' momentary probabilities or magnitudes imperfectly signal expected starvation probability. A considerable literature has addressed how humans make choices on simple, economic gambles with one to three steps[12–16],

but it is much less well known how humans evaluate deeper sequences[2, 17].

Here, we asked to what degree humans rely on optimal vs. heuristic decision policies, and how these are computed neurally. We hypothesized optimal policy computation in multimodal regions of the medial prefrontal cortex (MPFC) known to integrate economic decision variables and to evaluate prospective outcomes[18, 19]. Our results show that participants rely on a heuristic decision policy as well as on the optimal policy—both of which relate to blood-oxygen-level dependent (BOLD) signals in MPFC regions. The discrepancies between the two employed policies scale with reaction times (RTs) and dorsal MPFC activity.

## Results

**Sequential decision-making was framed as virtual foraging.** We developed a novel task to investigate how humans evaluate complex sequences of decisions (Fig. 1). This virtual foraging task embodied a Markov decision process[20] (Fig. 2) and contained decision-making aspects of foraging while neglecting biophysical affordances such as actual energy consumption and physical efforts. In line with similar monkey and human tasks[2–4, 17, 21–25], it thus abstracts from actual foraging[26, 27] to reflect generic properties of sequential decision-making.

In our task, participants were endowed with varying "energy resources," depicted graphically as an energy bar. Participants were financially rewarded if they "survived" over a maximum of five time steps, called "days," within a given mini-block of trials, called "forest" (Fig. 1). That is, participants only received a payoff if at the end of five consecutive decisions within a forest their final energy level was above zero. On each day, they decided between "foraging" and "waiting." Foraging entailed either an energy gain or loss, with a graphically signaled probability. Waiting resulted

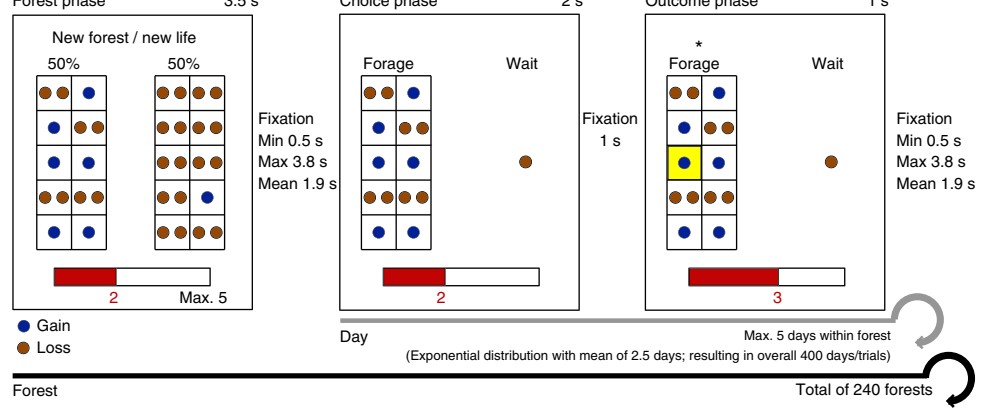

**Fig. 1** Sequential decision-making task in a virtual foraging frame. Participants "forage" within 240 different types of mini-blocks, called "forests". They are monetarily rewarded for averting "starvation" (keeping "energy bar" above zero) at day five in the forest (i.e., the last trial within a mini-block). The initial energy is reset in new forests. In the forest phase, participants see the two foraging environments or "weather types," that define the forest. This information is necessary so that participants can infer all possible future states. No choice is required during the forest phase. In the following time steps, called "days," within a forest, each of the two weather types occurs with a probability of 0.5. We depict foraging environments in the form of spatial grids (to allude to food distribution in natural environments). The numbers of colored dots within a subfield of the grid illustrates the magnitudes of potential gains and losses. If participants choose to forage, each of the 10 subfields of the grid has a 0.1 probability of being realized. Thus, the probability of foraging success is the proportion of subfields containing gains. Here, the probability of foraging success is 0.6 in the left grid and 0.1 in the right grid; and the gain magnitude is 1 point in both. Loss magnitudes are always 2 points. In the example, the left grid depicts "good" weather and the right grid "bad" weather. In the choice phase of each day, participants decide between foraging and waiting, which entails a sure loss of one energy point. In the outcome phase, participants see the impact of their choice on their energy state. In the depicted example, the participant chose to forage and successfully gained 1 point. The yellow square depicts the realized subfield. After the choice phase and a fixation period, either a new day in the same forest or a new forest starts. The number of days past in a forest is not shown to participants. See Fig. 2 for mathematical description of the task. See Table 1 and Table 2 for explanations and example parameters according to Fig. 1

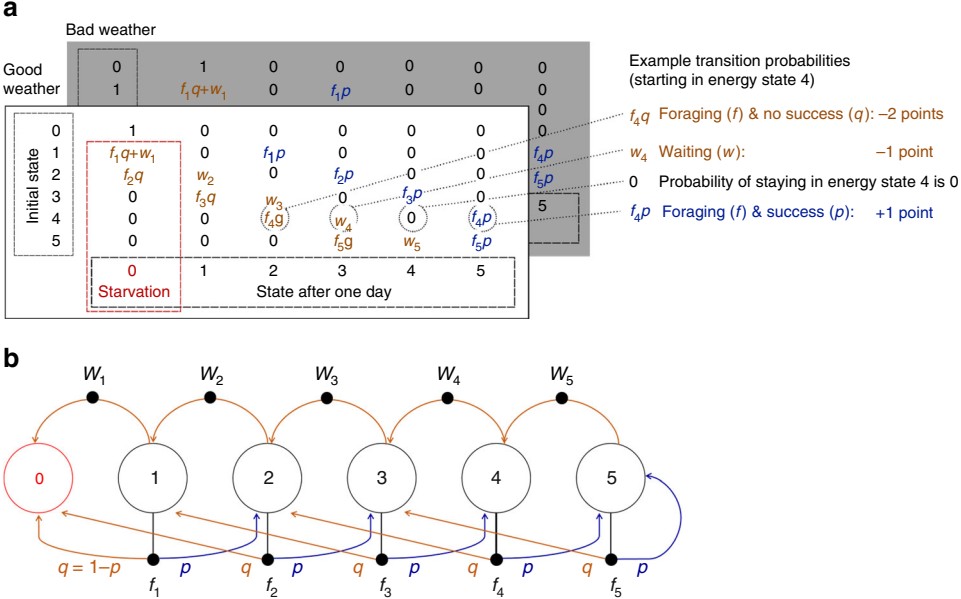

**Fig. 2** Markov decision process underlying the sequential decision-making task. **a** Example transition matrices used for determining the value differences between the two choice options according to the optimal policy (corresponding to the forest in Fig. 1). The entries in the matrices are the probabilities to transition from the initial energy state (rows) to the final energy state (columns) in one step. These entries (and their positions within the matrix) depend on the weather types of the forests and on the choice patterns. Many of the entries are 0, which means that transitions between the respective initial and final states are impossible. The forest is specified by the probability of foraging success $p$ (with $q = 1−p$) and the magnitude of gains and losses. These magnitudes are reflected in the positions of the probabilities within the matrix. Choices are reflected by the probabilities for foraging $f$ and waiting $w$ (with $f = 1−w$). The probabilities $f$ and $w$ depend on the initial state and are indexed accordingly. Additionally, the optimal policy depends on the number of remaining days in the forest since there is a finite horizon with a maximum number of five days. Starvation is absorbing, which is why the probability of staying in state zero is 1. There are two corresponding transition matrices for the two weather types. Backward computation is used to determine the value differences between the two choice options according to the optimal policies. That is, the values of the two choice options are first evaluated according to the last day in the forest, then according to the second-last day, etc. **b** The state transition diagram (for one weather type) corresponding to the example transition matrix in **a**. Large circles depict (energy) states and small filled circles depict the two actions to choose from. Arrows indicate transitions between states. For clarity, only one weather type is shown. In total, each forest type comprises 12 states = 6 (energy states) × 2 (weather types)

in a certain but smaller energy loss. To create non-trivial optimal policies, one of two environmental conditions, called "weather types," occurred randomly with a probability of 0.5 on a given day within a forest. The current weather type was signaled before participants made their choice. Each forest was specified by an initial energy endowment and a distribution of gain magnitudes and probabilities for each of the two weather types (Fig. 2; see Table 1 for a list of all heuristic variables and Table 2 for all variables related to the optimal policy; see Supplementary Note 1 for written task instructions).

We computed the a priori optimal policy that minimizes starvation probability according to a finite time horizon of five days for each combination of energy state, weather type, and day within a forest. Unless otherwise specified, we refer to the optimal policy according to the normative finite time horizon of five steps. To account for noise in the decision process, we used a probabilistic version of the optimal policy (i.e., the underlying value difference between foraging and waiting, which is the decision variable) to explain participants' choices, RTs, and fMRI data.

**Participant's choices followed primarily a heuristic policy.** We tested how well the optimal policy and ten different heuristic decision variables explained participants' choices (see Table 1 and Table 2 for explanations and examples of these variables, respectively). Normatively, participants should compute the optimal policy according to the remaining number of days, out of the five days within a forest. By definition, heuristics are myopic

and do not rely on a full evaluation of all possible upcoming outcomes. As heuristics, we considered variables related to the momentary foraging options (probability of foraging success, magnitude of the possible gain, their combination in form of expected value, EV), the current internal energy state (continuous or binary), the current weather conditions, and the number of days in the current forest. Additionally, we tested two heuristic policies ("change in states" and "win-stay-lose-shift") that derive from past states. We finally included a myopic policy that is optimal for a horizon of one day. By design, the optimal policy shares some predictions with several of the considered heuristics most notably the probability of foraging success (Supplementary Figs. 1, 2). However, average shared variance, derived on a trial-by-trial basis across participants, was sufficiently low to dissociate which variables accounted for participants' decisions (Supplementary Figs. 1, 2).

In a Bayesian model comparison of eleven logistic regression models, the signaled probability of foraging success emerged as best single predictor of participants' choices across two samples of participants (fMRI sample: $n = 28$; Bayesian model comparison, protected exceedance probability of winning model, PEP = 0.984; Fig. 3a; Supplementary Table 1; behavioral sample: $n = 21$; PEP = 0.999; Supplementary Fig. 3a; Supplementary Table 2). Visual inspection of plots showing posterior predictive checks demonstrate that, also qualitatively, no other model with a single predictor captured choice data as well as the model with the probability of foraging success (fMRI sample: Fig. 4a; behavioral sample: Supplementary Fig. 4a). This included models relying on the optimal policy or on a heuristic based on expected value (see

**Table 1 Overview of heuristic variables**

| Variable name | Explanation | Theoretically possible values of this variable in the task | Example value of this variable (as in Fig. 1 choice phase) | Grand mean of variable across fMRI participants (mean within participant SD) |
|---|---|---|---|---|
| Probability of foraging success ($p$ with $q = 1-p$) | Momentary probability that the participant can gain a certain magnitude of energy points (vs. losing two energy points). The participant can infer this probability by counting the number of subfields with gains (i.e., subfields with blue dots in Fig. 1) in the grid that contains ten subfields. | 0.1–0.9 in steps of 0.1 | 0.6 | 0.55 (0.24) |
| Magnitude of foraging gain $g$ | Momentary magnitude of the possible gain if foraging is successful. This is depicted by the number of (blue) "gain dots" per subfield of the grid. | 0–4 in steps of 1 | 1 | 1.97 (1.41) |
| Expected value (EV) of the foraging option | Momentary probability of foraging success multiplied by the corresponding magnitude of foraging gain $g$ plus (1-probability of foraging success) multiplied by the loss incurred for unsuccessful foraging, which is always −2. The EV of the waiting option is always −1. | −1.8 to 3.8 | −0.2 | −0.14 (1.13) |
| Continuous energy state $s$ | Current state of the energy bar. (An energy state of zero is synonymous with starvation and therefore participants cannot make choices at an energy state of zero.) | 1–5 in steps of 1 | 2 | 2.97 (1.09) |
| Binary energy state | When the continuous energy state is one, waiting leads to sure death. In higher energy states, waiting will never lead to starvation. The variable "binary energy state" distinguishes between these situations (1=energy state is one; 0=energy state is two or higher). | Binary variable: 1 or 0 | 0 "waiting does not lead to starvation" | 0.07 (0.26) |
| Weather type | Each forest type specifies two weather types that can be roughly classified as "good" or "bad" depending on whether they imply a lower or higher probability of starvation. Weather types are relative to each other (i.e., a given combination of $p$ and $g$ can be the "good" weather type if paired with a relatively worse weather type with lower $p$ and $g$, or the "bad" weather type if paired with a relatively better weather type, higher $p$ and $g$). | Categorical variable: 1 "bad" or 2 "good" | 2 "good" | 1.50 (0.5) |
| Days past in a forest (i.e., number of time steps $t$) | Participants remain within a given forest (i.e., mini-block) for up to 5 days (i.e., trials). The number of days is not explicitly depicted on the screen but participants can easily infer it by counting the number of choice phases after the last occurrence of the forest phase. | 1–5 in steps of 1 | 1 | 1.57 (0.91) |
| Change between past and current energy states | Participants might track the difference between their energy states in the past trial and the current trial (within and across forests). | −2 to +4 in steps of 1 (maximum loss was 2 energy points & maximum gain was 4 energy points) | Not available in Fig. 1 because the change depends on the previous trial that is not depicted. In the next choice phase, the change in energy states is +1. | −0.90 (1.55) |
| "Win-stay-lose-shift" (WSLS) strategy | Participants might use a strategy, which prescribes foraging if the energy state increased with respect to the past trial and waiting if the energy state decreased. WSLS is a binarized version of the change between past and current energy states | 1 "energy state increased" or 0 "energy state decreased" | Not available in Fig. 1 because previous trial not depicted. In next choice phase WSLS is 1 "energy state increased" | 0.38 (0.48) |

**Table 2 Overview of variables related to the optimal policy, choice uncertainties, and discrepancy**

| Variable name | Explanation |
|---|---|
| Optimal policy (h-5), i.e., value difference between foraging and waiting according to the optimal policy with a horizon of five days. | Ideally, participants should minimize the probability of starvation after five days. The optimal policy per se specifies the probabilities with which participants should forage (or wait) given the current internal state and the current time step. Since the optimal policy per se relies on taking the "true" maximum over the value difference between the two choice options, it either prescribes waiting or foraging (or is indifferent between the two choice options). We therefore use the continuous value difference between foraging and waiting as predictors of participants' choice, RT, and fMRI data. The optimal policy can also be calculated according to a horizon different from the five days incentivized in our task. These horizons, and notably a horizon of only one step (1-h), are not normative in our task (see Supplementary Fig. 10 for the prescriptions according to different horizons). |
| Choice uncertainty: probability of foraging success | Cases in which the prescriptions of the employed heuristic policy are closer to 0 (i.e., waiting) or 1 (i.e., foraging) are less uncertain than cases in which the prescriptions lie in-between. We used the mean parameter estimates of the behavioral sample to derive the relevant logistic function (cf. Supplementary Fig. 3c). The derivative of this logistic function is used to index choice uncertainty. |
| Choice uncertainty: optimal policy (h-5) | In analogy to the choice uncertainty of the employed heuristic, the optimal policy can confer more or less choice uncertainty. In some cases, the absolute value difference between foraging and waiting is small (i.e., it does not matter which option is chosen). In other cases, the value differences clearly indicate that foraging or waiting should be chosen. As for the choice uncertainty of the heuristic, derivatives of the logistic function obtained from the mean parameter estimates of the behavioral sample are used (cf. Supplementary Fig. 3d). |
| Discrepancies., absolute differences in the prescriptions of the two policies | The optimal policy and any heuristic policy make prescriptions about whether foraging or waiting should be chosen (according to logistic functions that relate the respective decision variables to choices). In some cases, optimal and heuristic policies make quite distinct prescriptions (high discrepancy), whereas in others they make quite similar prescriptions (low discrepancy). |

also Supplementary Note 2). Since decisions were not perfectly explained by this heuristic, we next asked if the optimal policy or any other heuristic contributed to participants' choices.

**The optimal policy explained additional variance in choices.** After accounting for the best heuristic, choices in the larger fMRI sample ($n = 28$) were best explained by the a priori optimal policy with a horizon of five time steps (Bayesian model comparison, PEP = 0.936; Fig. 3b; Supplementary Table 1). This model with two predictors also won in an extensive model comparison of all 55 pairs of candidate variables (Bayesian model comparison, PEP = 0.840; Supplementary Table 3). Visual inspection of plots showing posterior predictive checks confirm that the winning model qualitatively captured participants' behavior (see Fig. 3c, d; Fig. 4 for the winning model; see Supplementary Figs. 5–8 for extended posterior predictive checks of all models and for parameter estimates of a full model including all candidate variables). Additionally, models including interactions between the most important heuristics did not provide a better fit than the model with the optimal policy (i.e., the model including the probability of foraging success and the optimal policy reached PEP = 0.984; Supplementary Table 1). There was no decisive evidence (according to protected exceedance probability; PEP = 0.549) that a model with three variables explained choices better than the best model with two variables (Supplementary Table 1).

In the (smaller) behavioral sample ($n = 21$), we found that protected exceedance probabilities did not decisively distinguish between two-variable models including the optimal policy, or the binary energy state heuristic, respectively (Bayesian model comparison, PEP = 0.298). However, log-group Bayes factors provided decisive evidence for the same model as in the fMRI

sample, including the best heuristic and the optimal policy (log-group Bayes factors between best and second-best model >3; Supplementary Table 2; Supplementary Table 4; Supplementary Fig. 3c, d; Supplementary Fig. 4b, c; Supplementary Fig. 9). Thus, overall our model comparisons across the two groups decisively favored the optimal policy as a predictor of participants' choice in our task. Nevertheless, it is possible that on specific subsets of trial types (e.g., with energy state one), different variables predicted behavior better (see Supplementary Figs. 6–8; see Supplementary Note 3 for exploratory analyses on the relationship of task behavior, IQ, and questionnaires scores).

Taken together, model comparisons suggested that participants' choices were often predicted by a heuristic policy, but additionally choices followed the normatively optimal policy.

**Participants' choices took five future states into account.** Using the optimal policy implies computing future states. We performed detailed analyses to determine how many days participants thought ahead in the task. Participants were incentivized to consider a time horizon of five days. In the main task they would often not finish five days (to enhance fMRI design efficiency), but they were instructed that their payoff depended only on a random subset of ten forests that would be completed outside the scanner. When comparing models with different horizons, we found that participants' choices were indeed best described by a time-horizon of five days (Bayesian model comparison, fMRI sample: $n = 28$; PEP = 1.000; behavioral sample: $n = 21$; PEP = 0.986; see Supplementary Fig. 10 for an illustration of the different prescriptions made by optimal policies with different time horizons and Supplementary Fig. 11 for model comparisons). This finding was corroborated when analyzing subsets of

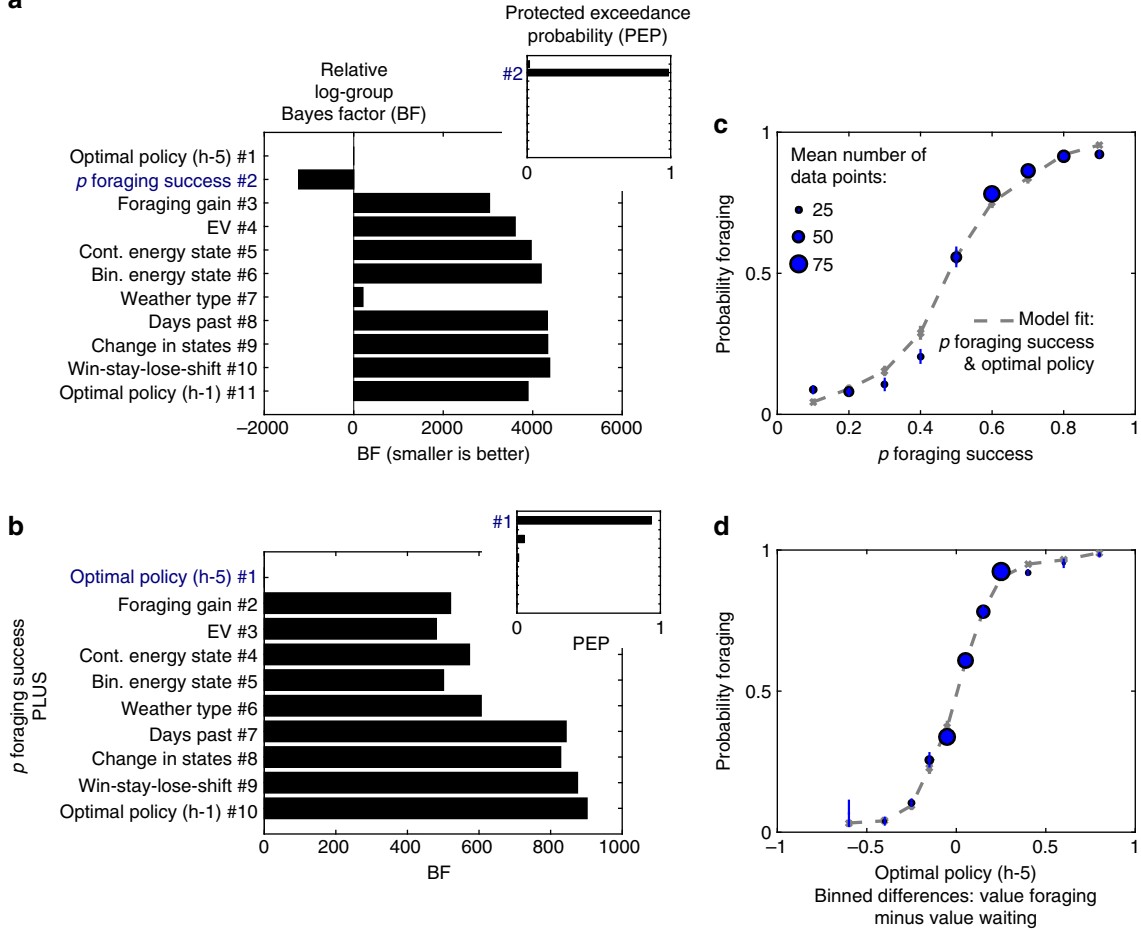

**Fig. 3** Choice data and models of the fMRI sample. **a** Model comparisons show that the probability of foraging success was the best single predictor of participants' behavior. Main plots depict fixed-effects analyses using log-group Bayes factors based on Bayesian Information Criterion (BIC) relative to model #1. Insets show random-effects analyses using protected exceedance probabilities (PEP) with the winning model marked. See Table 1 and Table 2 for lists that specify the task variables and thus the models tested here. **b** Crucially, the a priori optimal policy according to a time horizon of five days best explained the remaining variance in participants' choices. **c** The winning model, which includes the probability of foraging success and the optimal policy, captures the empirical relationship between participants' average choices and the probability of foraging success. **d** The winning model captures the relationship between participants' average choices and the optimal policy according to a horizon of five days (binned value differences of foraging vs. waiting). In the left-hand panels, blue font denotes the winning models. In the right-hand panels, error bars are SEM. Per data bin, circles depict mean empirical data points and lines and crosses depict mean model predictions (averaged for simulated data according to each participant's model fit). In several cases, error bars are smaller than the circles, which scale with the average number of trials contributing to the respective data points. See Supplementary Fig. 3 for the behavioral sample. See Supplementary Tables 1–4 for model comparisons. h-5: horizon of 5 days; h-1: horizon of 1 day; cont.: continuous; bin.: binary

trials in which policies with different horizons made opposing prescriptions (*t*-tests against midpoint of choice proportion: fMRI sample: $n = 28$; absolute *t*-values > 5.2, all *p*'s < $10^{-4}$; behavioral sample: $n = 21$; absolute *t*-values > 1.9, all *p*'s < 0.065; Supplementary Table 5).

**Participants used the overall best heuristic available.** Does the probability of foraging success constitute a useful heuristic? This heuristic shared the largest amount of variance (0.37) with the optimal policy on a trial-by-trial basis and thus the two were related by design (Supplementary Figs. 1, 2). Additionally, simulations showed that basing one's decision exclusively on this metric would lead to a lower starvation rate (0.15) than any other of the considered metrics (all other starvation rates >0.17; Supplementary Fig. 12; Supplementary Note 4). Thus, participants relied primarily on the best available heuristic variable. Notably,

this heuristic was neither primed by our task instructions nor particularly visually salient (Supplementary Note 5).

**Reaction times increased with choice uncertainties.** Models of choice data indicate that participants used both a heuristic policy, i.e., the probability of foraging success, and the optimal policy. Consequently, we predicted that RTs should reflect the choice uncertainties associated with these two variables. (These uncertainties were calculated on the basis of independent data from the behavioral sample.) Indeed, RTs were slower when choice uncertainties were high (see Fig. 5a–d for the fMRI sample and Supplementary Fig. 13a–d for the behavioral sample; see Supplementary Fig. 6 for mean parameter estimates and Supplementary Table 6 for statistics from a linear mixed effects model; fMRI sample: $n = 28$; uncertainty of probability of foraging success: $t = 15.34$, $p < 10^{-15}$; uncertainty of optimal policy: $t = 2.68$, $p < 0.05$; behavioral sample: $n = 21$; uncertainty of probability of

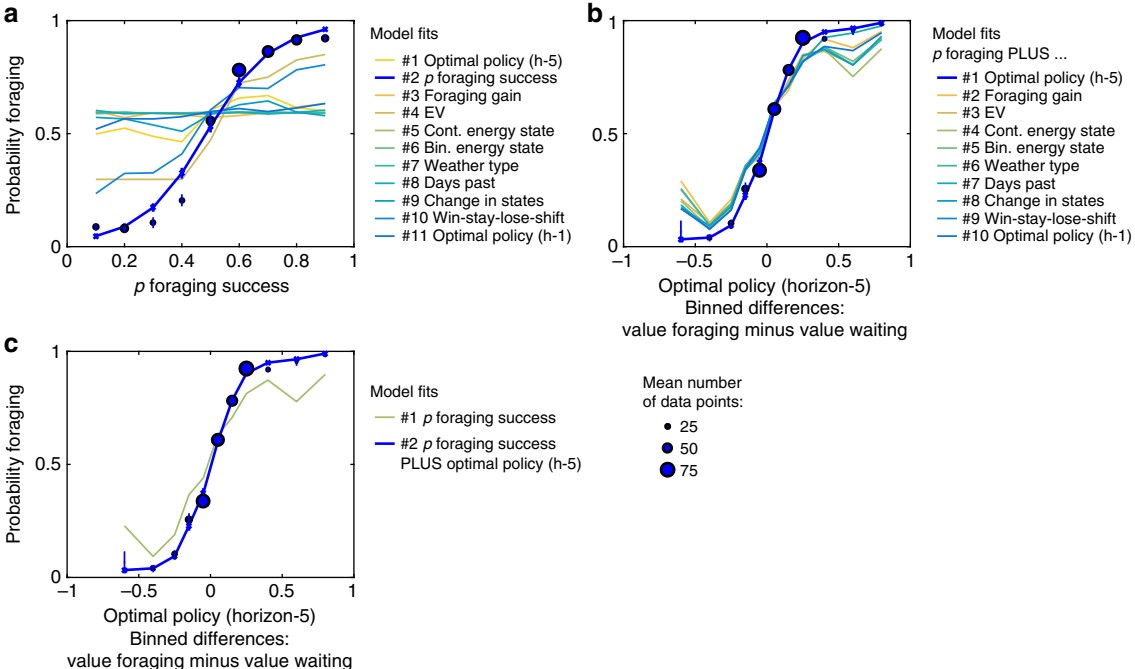

**Fig. 4** Comparison of choice data to different model predictions in the fMRI sample. **a** Posterior predictive checks show that—among the models with a single predictor—the model comprising the probability of foraging success captures choice data better than any of the other models. **b** Posterior predictive checks show that—among the models with two predictors—the model comprising the probability of foraging success and the optimal policy (at a horizon of 5 time steps) captures choice data better than any of the other models. Overall most models make quite similar predictions since they all include the probability of foraging success. **c** Posterior predictive checks show that the model comprising both the probability of foraging success and the optimal policy (horizon-5) provides a better fit to the data than the model that only comprises the probability of foraging success. Error bars are SEM. Per data bin, circles depict mean empirical data points and colored lines and crosses depict mean model predictions (averaged for simulated data according to each participant's model fit). In several cases, error bars are smaller than the marker sizes, which scale with the average number of trials contributing to the respective data points. See Table 1 and Table 2 for lists of all variables. See Supplementary Fig. 4 for the behavioral sample. See Supplementary Fig. 5 and Supplementary Fig. 9 for posterior predictive checks of the winning model with choice data split according to the nine other heuristics and combinations thereof. See Supplementary Fig. 6 for parameter estimates of a full model including all candidate variables. See Supplementary Fig. 7 and Supplementary Fig. 8 for further posterior predictive checks of the winning model with choices split jointly according to the energy state and the probability of foraging success or the weather type. See Supplementary Fig. 10 and Supplementary Fig. 11 for comparisons of different time horizons. h-5: horizon of 5 days; h-1: horizon of 1 day; cont.: continuous; bin.: binary

foraging success: $t = 5.80$, $p < 10^{-5}$; uncertainty of optimal policy: $t = 1.86$, $= 0.062$).

**Following the heuristic entailed fast choices.** Given that participants' choices appear to use heuristic and optimal policy for making their decision, the question arises how these two are computed in relation to each other. As the heuristic is easier to determine than the optimal policy, we predicted that choices following the heuristic policy should be faster than those following the optimal policy. We identified the subset of trials in the fMRI sample in which the two policies made opposite prescriptions. More specifically, we computed the policies from the behavioral sample and applied these policies to the fMRI sample to select trials with opposite choice prescriptions. Within this subset (mean proportion of trials over participants ± standard deviation, SD = 0.19 ± 0.02), mean RTs for trials in which participants' choices followed the heuristic policy were faster than for trials in which choices followed the optimal policy (mean difference ± SD = 90.7 ± 109.2 ms; $t(27) = 4.40$; $p < 0.001$; $p < 0.001$; mean predicted choices according to the used policy did not differ between these trials; $p > 0.4$).

In addition, a linear mixed effects model of RT data provided evidence for a relatively more pronounced influence of the choice uncertainty of the heuristic compared to the choice uncertainty of the optimal policy (as can be seen by comparing the $t$-values for the uncertainty of the heuristic vs. the $t$-values for the uncertainty of the optimal policy; fMRI sample: 15.34 vs. 2.68; behavioral sample: 5.80 vs. 1.86; Supplementary Table 6). That is, choice uncertainty under the heuristic policy related more strongly to RTs than choice uncertainty under the optimal policy.

**Reaction times increased with discrepancies between policies.** Next, we addressed whether both policies are integrated. In this case, decisions should take longer when the two policies make discrepant prescriptions. We quantified these discrepancies between the two variables as the absolute differences in choice probabilities. Indeed, decisions were slower when discrepancies between the decision variables were larger (linear mixed effects model, fMRI sample: $n = 28$; $t = 6.55$, $p < 10^{-5}$; behavioral sample: $n = 21$; $t = 2.94$, $p < 0.01$). This effect was present in addition to influences of choice uncertainties (fMRI sample: Fig. 5e; behavioral sample: Supplementary Fig. 13e).

Visual inspection of plots showing posterior predictive checks indicate that log-transformed RT data were qualitatively captured by a model that included the heuristic and the optimal policies themselves, their associated choice uncertainties, and the discrepancies in the choice probabilities of the two policies (see Fig. 4; Supplementary Figs. 6, 7; Supplementary Fig. 13).

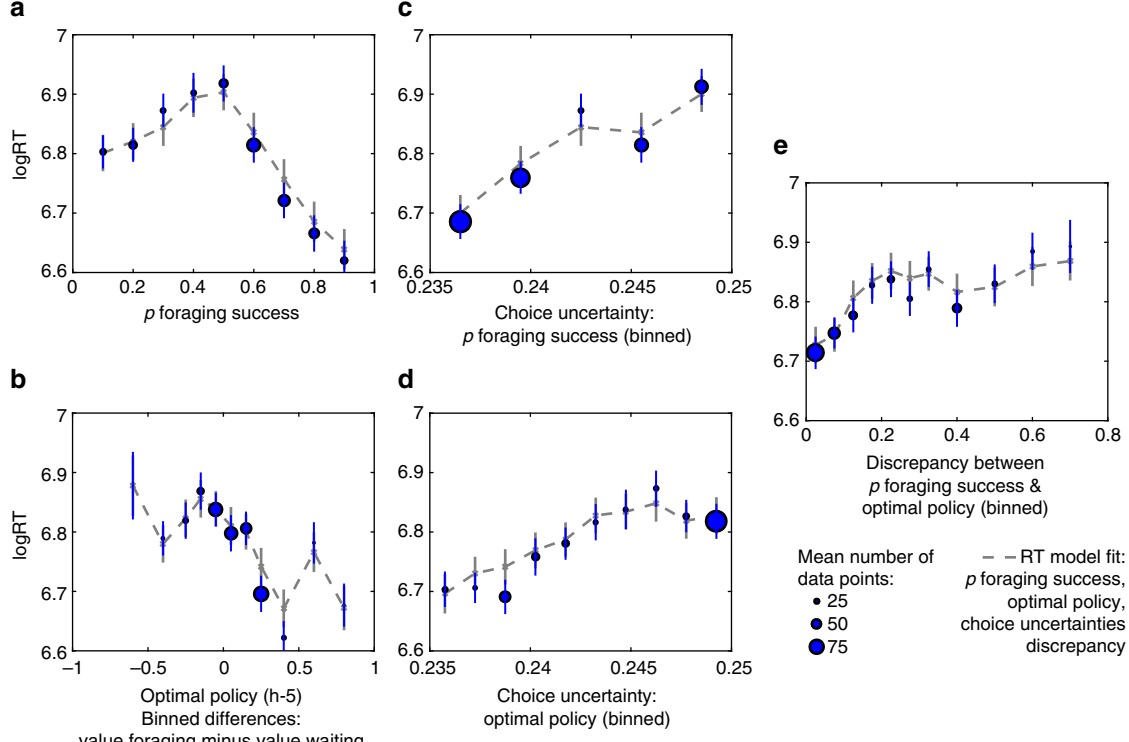

**Fig. 5** Reaction time data and models of the fMRI sample. We tested the relationship between RTs and variables associated with the heuristic and optimal policies. Since the probability of foraging success emerged as the best predictor of participants' choices, we only included this but not any other heuristic in the model of RT data. RTs relate to **a** the probability of foraging success and also weakly to **b** the optimal policy. Importantly, RTs become slower with **c**, **d** increasing choice uncertainties of these two variables and **e** higher discrepancies in their prescriptions. Posterior predictive checks show that RT data were well captured by a model that includes the five depicted variables. Error bars are SEM. Per data bin, circles depict mean empirical data points and lines and crosses depict mean model predictions (averaged for simulated data according to each participant's model fit). Circles scale with the average number of trials contributing to the respective data points. See Supplementary Fig. 13 for the behavioral sample, Supplementary Fig. 6 for the parameter estimates of the full RT model, and Supplementary Fig. 7 for posterior predictive checks of the RT model with data split differently. See Supplementary Table 6 for statistical inferences obtained from a linear mixed effects model

Motivated by this model of RT data, we also tested whether interactions of choice uncertainties or discrepancy influenced choices but found no decisive effects (Bayesian model comparison, fMRI sample: $n = 28$; highest PEP = 0.521; Supplementary Table 1; behavioral sample: $n = 21$; highest PEP = 0.202; Supplementary Table 2).

**MPFC reflected variables guiding choice**. We next assessed how heuristic and optimal policies are neurally computed (on a trial-by-trial basis). To do so, we implemented a main general linear model (GLM) that included the variables from the winning choice model and the RT model as parametric modulators during the choice phase: the probabilities of foraging success, the value differences between foraging and waiting according to the optimal policy (time horizon of five days), their associated choice uncertainties, and the discrepancies in choice probabilities, as well as log-transformed RTs. Given the central role parts of the MPFC in decision-making, we specifically focus on this region in the presentation of our results. All described clusters are whole-brain family-wise error (FWE) corrected for multiple comparisons at $p < 0.05$ with a cluster-defining threshold of $p < 0.001$.

The momentary probability of foraging success, i.e., the variable underlying the heuristic policy, showed a positive relation with BOLD signals in a posterior part of the dorsal MPFC (DMPFC, extending into pre-supplementary motor area, pre-SMA; peak voxel $x;y;z$(MNI) = 8; 29; 50; $t = 5.80$), in bilateral

intraparietal sulcus (IPS; left: $x;y;z$(MNI) = −42; −45; 51; $t = 4.42$; right: $x;y;z$(MNI) = 47; −42; 50; $t = 5.37$), and the left frontal pole ($x;y;z$(MNI) = −42; 47; −5; $t = 4.28$), among other regions (Fig. 6a, see Supplementary Table 7 for fMRI results in the choice phase). The same variable showed a negative relation with signals in the perigenual anterior cingulate cortex (ACC), extending into the ventral MPFC (VMPFC; $x;y;z$(MNI) = 6; 33; 6; $t = 6.27$; Fig. 6b).

The optimal policy showed a positive relation with activity in perigenual ACC (extending into VMPFC; $x;y;z$(MNI) = 6; 50; 6; $t = 4.53$) and mid-cingulate cortex ($x;y;z$(MNI) = 2; 14; 29; $t = 5.31$; Fig. 6c). That is, parts of the MPFC were relatively more active when waiting was favored by the heuristic and when foraging was favored by the optimal policy. This suggests an overall involvement of the MPFC in computing differences in choice value of the variables that explained participants' behavior.

**MPFC reflected choice uncertainties and discrepancies**. Lower choice uncertainty of the heuristic was related to increased BOLD signals in an anterior part of the VMPFC ($x;y;z$(MNI) = 9; 59; −2; $t = 6.24$), dorsal MPFC regions ($x;y;z$(MNI) = 3; 56; 33; $t = 4.50$) as well as to the inferior frontal gyrus (IFG; $x;y;z$(MNI) = 48; 35; −5; $t = 7.37$) and the posterior cingulate cortex ($x;y;z$(MNI) = 15; −27; 39; $t = 7.63$), among other regions (Fig. 7a; Supplementary Table 7).

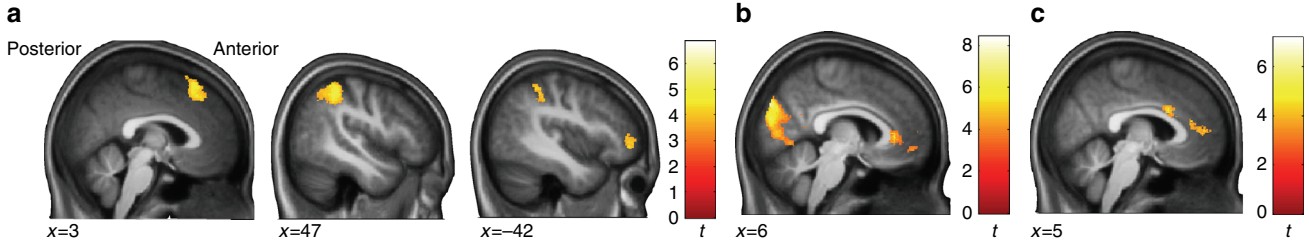

**Fig. 6** BOLD signals related to heuristic and optimal policies during the choice phase. **a** The probability of foraging success, the employed heuristic policy, showed a positive relation with BOLD signals in DMPFC (extending into pre-SMA; peak voxel $x;y;z$(MNI) = 8; 29; 50; $t$ = 5.80), in bilateral IPS (left: $x;y;z$ (MNI) = −42; −45; 51; $t$ = 4.42; right: $x;y;z$(MNI) = 47; −42; 50; $t$ = 5.37), and the left frontal pole ($x;y;z$(MIN) = −42; 47; −5; $t$ = 4.28) among other regions. **b** The probability of foraging success showed a negative relation in the perigenual ACC extending into VMPFC ($x;y;z$(MNI) = 6; 50; 6; $t$ = 4.53). **c** The optimal policy showed a positive relation in perigenual ACC ($x;y;z$(MNI) = 6; 50; 6; $t$ = 4.53) and mid-cingulate cortex ($x;y;z$(MNI) = 2; 14; 29; $t$ = 5.31). Overlay on group average T1-weighted image in MNI space; clusters are whole-brain family-wise error (FWE) corrected for multiple comparisons at $p <$ 0.05 with a cluster-defining threshold of $p <$ 0.001. Color bars depict $t$-values. See Supplementary Table 7 for a list of all clusters

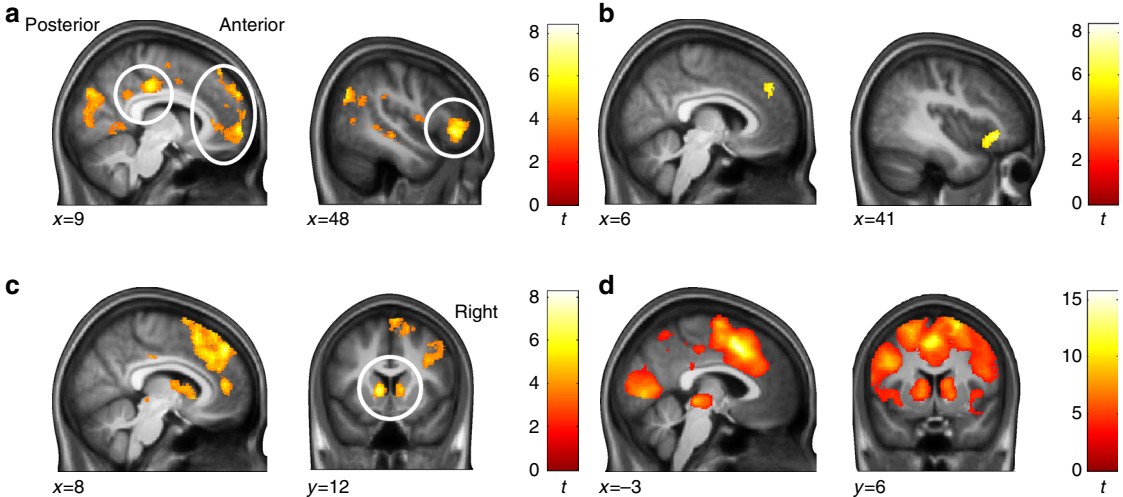

**Fig. 7** BOLD signals related to choice uncertainties, discrepancies, and RTs during the choice phase. **a** Choice uncertainty of the heuristic (i.e., probability of foraging success) showed a negative relation with BOLD signals in VMPFC (peak voxel: $x;y;z$(MNI) = 9; 59; −2; $t$ = 6.24), DMPFC ($x;y;z$(MNI) = 3; 56; 33; $t$ = 4.50), IFG ($x;y;z$(MNI) = 48; 35; −5; $t$ = 7.37), and the posterior cingulate cortex ($x;y;z$(MNI) = 15; −27; 39; $t$ = 7.63), among other regions. **b** Choice uncertainty of the optimal policy exhibited a negative correlation with DMPFC/ACC ($x;y;z$(MNI) = 6; 42; 38; $t$ = 4.81) and IFG ($x;y;z$(MNI) = 48; 26; −9; $t$ = 5.88). **c** Discrepancies between the two policies showed a positive relation with DMPFC (extending into pre-SMA and ACC; $x;y;z$(MNI) = −9; 36; 38; $t$ = 8.25), bilateral dorsal striatum (left: $x;y;z$(MNI) = −11; 12; 3; $t$ = 7.16; right: $x;y;z$(MNI) = 11; 20; −3; $t$ = 7.66), and bilateral IFG (left: $x;y;z$(MNI) = −30; 26; −5; $t$ = 6.17; right: $x;y;z$(MNI) = 50; 26; −11; $t$ = 6.85). **d** For completeness, (positive) correlations with log-transformed RTs are depicted. Overlay on group average T1-weighted image in MNI space; clusters are whole-brain FWE corrected for multiple comparisons at $p <$ 0.05 with a cluster-defining threshold of $p <$ 0.001. Color bars depict $t$-values. See Supplementary Table 7 for a list of all clusters (as well as Supplementary Tables 8–10; Supplementary Fig. 14; and Supplementary Fig. 15 for further analyses of the choice phase). See Supplementary Table 11 and Supplementary Fig. 16 for BOLD signals during the outcome phase

Lower choice uncertainty of the optimal policy scaled with activity in DMPFC, extending into ACC ($x;y;z$(MNI) = 6; 42; 38; $t$ = 4.81), and in the IFG ($x;y;z$(MNI) = 48; 26; −9; $t$ = 5.88; Fig. 7b; Supplementary Table 7). That is, we found increased BOLD signals with increasing choice certainty of both heuristic and optimal policy in regions of the MPFC.

Crucially, as in our RT data, we found evidence for an integrated computation of the heuristic and the optimal policy: DMPFC activity correlated in a trial-by-trial fashion with the discrepancies between the two policies (i.e., the absolute differences in their decision variables; $x;y;z$(MNI) = −9; 36; 38; $t$ = 8.25; Fig. 7c; Supplementary Table 7). This DMPFC region extended into pre-SMA and ACC. The same metric correlated with BOLD signals in bilateral dorsal striatum

(left: $x;y;z$(MNI) = −11; 12; 3; $t$ = 7.16; right: $x;y;z$(MNI) = 11; 20; −3; $t$ = 7.66) and bilateral IFG (left: $x;y;z$(MNI) = −30; 26; −5; $t$ = 6.17; right: $x;y;z$(MNI) = 50; 26; −11; $t$ = 6.85).

**Additional fMRI analyses corroborated MPFC involvement.** All relationships of the relevant model variables—as described above for the main GLM—emerged with log-transformed RTs as parametric modulator in the GLM (Fig. 7d). Overall, the same regions described above were also identified in a second GLM, which additionally included participants' choices themselves as parametric modulators during the choice phase (Supplementary Table 8). The primary qualitative difference between the GLMs with and without choices as additional parametric modulator was

that in the GLM including choices the DMPFC cluster related to lower choice uncertainty of the optimal policy failed to reach significance with FWE correction for multiple comparisons at $p < 0.05$ (with a cluster-defining threshold of $p < 0.001$; cf. Fig. 7b).

For the sake of completeness, we set up a third GLM that only included participants' choices as parametric modulator (please refer to Supplementary Fig. 14 and Supplementary Table 9).

In a fourth GLM, we specifically analyzed how the values of the chosen options related to BOLD signals (Supplementary Fig. 15; Supplementary Table 10). The probabilities of foraging success according to the options chosen by participants scaled positively with the left frontal pole ($x;y;z$(MNI) = −47; 39; 6; $t = 4.63$), in a similar region as described above for the probabilities of foraging success according to the presented foraging options (i.e., $x;y;z$ (MNI) = −42; 47; −5; $t = 4.28$). The values of optimal policy according to the chosen options' values were positively related to the VMPFC ($x;y;z$(MNI) = 2; 63; 11; $t = 4.76$), in a similar region as described above for the value differences according to the optimal policy (i.e., $x;y;z$(MNI) = 6; 50; 6; $t = 4.53$). These findings corroborate that BOLD signals in the left frontal pole and in the VMPFC were associated with variables on which participants based their choices.

Finally, classic reward regions[14, 15] (including VMPFC, striatum, and posterior cingulate cortex) tracked the realized outcomes, that is the impact of participants' decisions on their internal energy state (overall peak voxel of the cluster: $x;y;z$(MNI) = 0; −26; 48; $t = 9.01$; Supplementary Fig. 16; Supplementary Table 11).

Taken together, our results show that variables relevant for participants' choices and RTs scale with activity in multiple brain regions involved in decision-making—in particular in ventral and dorsal MPFC areas.

## Discussion

This study addresses the neural computations required to make sequential decisions during virtual foraging, when participants had to navigate above a lower homeostatic boundary over a number of consecutive steps with probabilistic outcomes. We show that humans rely both on an easy-to-access heuristic and on an optimal policy that integrates over probabilistic future states. More specifically, we demonstrate that participants took advantage of a model-free heuristic available at the time point of decision-making. This heuristic of relying on the probability of foraging success performed best overall in explaining participants' choices among a large set of alternatives (including the foraging options' expected values). This finding is in line with studies showing that participants often base their choices on the overall probability of winning in another type of sequential decision-making task[16, 28]. Importantly, choices in our task were also explained by the normatively optimal policy. Participants' choices were therefore best explained by the combination of two metrics that constitute the end points of a spectrum between sloppy (but easy) and exact (but difficult) solutions.

We did not find decisive evidence that any linear combination of two candidate policies and variables explained our choice data better than the probability of foraging success and the optimal policy. Although it is theoretically possible that participants use a yet unknown decision policy (for discussion see refs. [29, 30]), such policies do not follow from the given task variables in an obvious way. Any such model would thus likely require higher complexity than the linear combination of the probability of foraging success and the optimal policy (or it would only apply to a less complex setting than the one investigated here). It is an interesting question for follow-up research whether on specific trial types

(possibly under-sampled here) participants may have used a more (or less) complex model. Also, it appears possible that in a more (or less) challenging task, participants may abandon the optimal policy in favor of a combination of two or more heuristics.

For our task, we can exclude that participants resorted to a simple heuristic of looking just one time step ahead, which would have been clearly suboptimal given that they finite time horizon was five steps in our task. Imposing different time constraints or varying the number of time steps in our task may alter the time horizon considered and shift the balance between heuristic and optimal computations. Evidence for the possibility of such dynamic alterations has been provided in a recent study by Keramati et al.[31] which demonstrated that humans adaptively adjust the depth of planning and the reliance on habits in a three-step learning task—with time pressure leading to shallower planning.

Our behavioral data suggest interdependent—but partly distinct—computations of heuristic and optimal processes during sequential choice. But what is the relationship between the two policies? Analyses of RTs showed that the choice uncertainty of both policies—but in particular of the heuristic—slowed RTs. Intriguingly, the discrepancies in choice probability between the two policies also led to longer RTs, which provides crucial evidence for computation of heuristic and optimal policies, and their integration. We interpret these findings as pointing toward progressive computational processes such that the approximations provided by the heuristic are abandoned if the associated choice uncertainties turn out to be too high. That is, an insufficient choice certainty of the heuristic metric may suggest that it is worthwhile to engage in a deeper search in form of a full-blown optimal policy computation.

Integrated computation of heuristic and optimal policies could be understood in terms of a process that accumulates from increasingly sophisticated policies, until the associated choice uncertainties become sufficiently small. This could explain why the uncertainty of the heuristic had a larger effect on RTs than the uncertainty of the optimal policy, which may be computed later during the trial. Integration of opposing predictions may then engage an additional time-consuming step, leading to increased RTs. In some cases, the requirement to integrate two specific types of information may be especially pertinent (for example at the energy state boundaries integrating information about the probability of foraging success and about the energy state can be crucial). Identifying the precise temporal requirements of information integration processes is an interesting and challenging avenue for future research.

Our fMRI data revealed that the heuristic variable, i.e., the probability of foraging success, was positively associated with BOLD signals in the frontal pole, the IPS, and a posterior part of the DMPFC. The same heuristic was negatively related to signals in a region of perigenual ACC extending into VMPFC. By definition, appropriate heuristics are related to the given task demands and therefore brain regions associated with any heuristic variable likely depend on the particular heuristic used. For example, the involvement of the IPS observed in the current study might be related to its role in processing numbers[32]. On the other hand, frontal pole, DMPFC, perigenual ACC and VMPFC play roles in several decision-making process[19]. Since participants were more likely to choose waiting rather than foraging when the probabilities of foraging success were small, the identified region of the perigenual ACC and VMPFC scaled positively with the relative value of "waiting" vs. "foraging," as signaled by the heuristic. This finding may thus accord with the role of a similar region in a conceptually different sequential decision-making task: In the task used by McGuire and Kable[33] participants had to adaptively decide how long to keep waiting for future rewards

with different, uncertain timings. The temporal unfolding of the subjective value during different waiting periods was related to the VMPFC. In both tasks, "waiting" (either as a discrete action as in our task or as temporal persistence as in the task by McGuire and Kable[33]) trades off current (opportunity) costs against potential gains in the future. Still, the fact that our analyses also identify regions outside the VMPFC that scale with the value of foraging (notably in the DMPFC) constitutes a difference between our findings and those by McGuire and Kable[33]. Interestingly, the uncertainty of the heuristic variable was negatively (and not positively) related to BOLD signals in several regions, in particular in the posterior cingulate cortex, an anterior part of the VMPFC, dorsal MPFC, and IFG. Put differently, these regions showed a positive association with the "easiness" of making a decision according to the heuristic policy. Negative relations of an uncertainty metric have previously been identified a region of the posterior cingulate cortex, which was slightly more posterior than the cluster identified here[12, 34].

Multimodal integration regions such as MPFC and IFG were associated with the optimal policy and also the choice uncertainty of the optimal policy. These regions have often been observed in studies testing for brain activity related to model-based learning processes[35, 36]. Indeed, optimal decision-making in our task bears considerably resemblance with model-based learning[36–38] since both involve searching across a tree of probabilistic future states. Here, we demonstrate brain activity when participants evaluated decision trees with extended time horizons independent of the uncertainties arising during learning. In particular, we found that a dorsal region of the MPFC correlates negatively with the choice uncertainty of the optimal policy (i.e., positively with the "easiness" of making a decision according to the optimal policy). This region seems to be especially well positioned to integrated different types of decision signals related to reward values and actions[18, 19]. An intriguing possibility is that this region may generally be related to the uncertainties in recursive evaluations of a tree of probabilistic future states, which is a key feature for inferring optimal solutions in many realistic tasks.

The discrepancies between the two employed policies showed a positive trial-by-trial relationship with a prominent cluster in the region of the DMPFC, extending into the pre-SMA and the ACC (in addition to relationships bilateral dorsal striatum and IFG). The DMPFC cluster overlaps with regions classically associated with multiple types of decision discrepancies[39, 40]. Thus, our finding relating discrepancies between the heuristic and optimal policies to a part of the DMPFC could potentially indicate that this DMPFC region becomes increasingly engaged when the computations of the two policies prescribe divergent choices such that progressive evidence accumulation and competition processes are required for making a decision.

Our virtual foraging task does not require exploring or inferring unknown, uncertain, or unobservable, states, which distinguishes it from learning tasks aimed at comparing explicitly signaled vs. previously trained values and tasks designed to assess model-based vs. model-free reinforcement learning[6, 35–38, 41–45]. Our approach could be extended to include learning or information seeking such that the optimal policy would require reducing uncertainty about environmental states (cf. [45, 46]).

We have previously shown that formal models incorporating homeostasis maintenance can explain economic choices better than standard economic models[4]. In the present study, we address the underlying choice mechanism. We show that humans minimize the probability of virtual starvation via combining heuristic and optimal policies. In general, our delineation of heuristic and optimal policy computations informs a substantial body of behavioral work supporting notions of nuanced trade-offs

between heuristic and optimal solutions[47–49] or of fine-grained specifications of rationality[5, 8, 50, 51]. Our behavioral findings corroborate recent proposals that the optimality of decision-making depends on cognitive limitations of the decision-maker as well as the energy and opportunity costs incurred during the decision process[5–8, 49, 52], such that a theoretically optimal model-based system ceases to be optimal under some circumstances. Our fMRI results relate activity in the MPFC to the model-based tree search implicated in the computation of the optimal policy. Brain activity associated with the discrepancies between heuristic and optimal policies points towards a mechanism of integrated computation that may well generalize over many decision scenarios with multiple probabilistic states. Thus, our study paves the way towards better understanding the computations of the neural systems involved in making ecologically optimal choices in the face of complex multi-step decision-making scenarios.

## Methods

**Participants**. Participants were recruited from a student population via mailing lists of local universities. Twenty-one participants participated in the behavioral study (15 female; mean ± standard deviation (SD): age = 25.8 ± 4.3 years). Thirty participants were recruited for the fMRI study. Two had to be excluded (one due technical problems with the scanner and one due to excessive motion >4 mm). This resulted in a final sample of 28 participants (13 female; age = 23.5 ± 3.9 years) for the fMRI study. Due to time constraints, one participant in the fMRI sample only performed eight out of ten sessions and another only performed nine out of ten sessions. Participants were paid a show-up fee (behavioral sample: CHF 20; fMRI sample: CHF 50) plus a variable amount (see Instructions and task). Both groups performed exactly the same task with the only difference that one group underwent MR scanning.

The study was conducted in accord with the Declaration of Helsinki and approved by the governmental research ethics committee (Kantonale Ethikkommission Zürich, KEK-ZH-Nr. 2013-0328). All participants gave written informed consent using a form approved by the ethics committee.

**Mathematical framework of virtual foraging task**. To probe sequential decision-making, we propose a toy scenario and a corresponding Markov decision process (MDP) for a hunter-gatherer or any foraging agent that aims at dynamically maintaining homeostasis over time. See Fig. 1 for an example trial and Fig. 2 for the corresponding transition matrix, and the associated state transition diagram. See Table 1 and Table 2 for lists of all variables described in this and the following sections. The decision-making agent has to keep its internal energy state $s$ above zero, i.e., the agent "dies from starvation" upon reaching the energy state zero at any time step. Here, the energy state can have discrete values equaling one to five energy points (but our model easily extends to additional numbers of energy states without loss of generality). At each time step $t$, or "day," the agent can chose to "wait" and incur a sure loss $c_w$ (of one energy point) or it can "forage" in which case the agent probabilistically gains an amount $g$ (of zero to four energy points) or incurs a cost for foraging $c_f$ (of two energy points). We denote the probability of foraging success as $p$ (i.e., the probability of gaining points during foraging). The probability of unsuccessful foraging and thus of incurring $c_f$ is $q = 1 - p$. The maximum energy is capped; in the highest energy state the agent cannot gain more but simply stays in the highest state if foraging is successful.

The agent "lives" within a given "forest" in which all relevant variables are specified (presented to participants during the "forest phase," see Fig. 1). We included good and bad environmental conditions denoted as "good and bad weather types" that each occur with a probability of 0.5 on a given time step in a given forest. That is, per forest there are total of 12 states in the MDP: 6 (energy states) × 2 (weather types).

A core component of an MDP is the transition matrix between these different states (see Fig. 2 for one example). Starvation, i.e., the state of having zero energy is absorbing (and thus the entry in the transition matrix leading from energy state zero to energy state zero is 1). Waiting leads to a sure loss $c_w$ of one point, which is formalized by the associated probabilities $w$ in the off-diagonal below the main diagonal. Foraging can lead to either success ($fp$) or not ($fq$). These probabilities constitute the entries in the associated off-diagonals (depending on the magnitudes of $g$ and $c_f$).

How should the agent's optimal policy look? The agent should minimize the probability of starvation $p_{starve}(n, s, t)$; i.e., it should minimize the probability of reaching zero energy points within a fixed and finite time horizon of $n$ days when starting with the internal energy state $s$. In our case, the instructed finite time horizon $n$ is always 5 days and the starting energy state $s$ at the first day can be 2, 3, or 4 energy points. Participants were incentivized accordingly, i.e., they received a monetary payoff (for a random subset of forests), if their energy was above zero at day 5 and nothing otherwise. This corresponds to a simple implementation of a

reward function within the framework of MDPs: All transitions to zero are associated with a "reward" of −1 and all other transitions with a reward of 0.

Starvation probability $p_{starve}(n,s,t)$ and thus the optimal policy depend on the agent's choice at the current time step $t$ and the next $n−1$ time steps—in addition to the dependence on $n$ and $s$. In our finite-horizon scenario, the starvation probability thus depends on the number of time steps $t$, i.e., days within a forest.

A rough and basic intuition for a more or less prototypical optimal policy under such circumstances is that in many types of forests the agent should forage when the weather is good and wait when the weather is bad; unless waiting leads to sure death on the next time step. When the energy state is high enough so that starvation is impossible within the remaining number of days (e.g., energy state is 4 and only 1 day is left of the 5 days within a forest), the agent is indifferent between the two choice options.

**Derivation of optimal policy in Markov decision process**. We derived the optimal policies analytically for a large number of different forests, i.e., for all combinations of $p$ (from 0.1 to 0.9 in steps of 0.1) and gains $g$ (from 0 to 4 in steps of 1) for a finite time horizon of $n = 5$ days. In our MDP, the possible policies consist of the probabilities of foraging $f$ in each energy state $s$, environmental condition ("weather"), and each time step $t$ (with the probabilities of waiting $w = 1 −f$). To derive optimal policies in our finite-horizon scenario, we used backward induction: That is, we started from the final time step (i.e., day 5) and calculated the values of the two choice options (i.e., foraging or waiting) for each state for that last and final time step. These values depend on the possible transitions from the respective states (see the example transition matrix in Fig. 2a and the corresponding example state transition diagram in Fig. 2b). If the value of foraging is higher than the value of waiting, foraging is the deterministically better option in that state and at that time step—or vice versa. If both choice options have the same value, the optimal choice is indifferent between the two options. We then used the maximum of the two choice options' values to calculate the values for the second-to-last time step and determined optimal choices. This procedure was repeated until arriving at the first time step (i.e., day 1).

The optimal policy thus depends on the time horizon considered. Our task was designed such that participants should use a time horizon of 5 time steps and participants were instructed and incentivized accordingly. Nevertheless, it is a possibility that participants use a different time horizon. Therefore, we calculated optimal policies for the following time horizons: $n = 7$, $n = 6$, …, $n = 1$. Participants may employ a longer (i.e., 7 or 6 days) or shorter (i.e., 4, 3, 2, or 1 days) time horizon than instructed. Time horizons are flexible in the sense that it is assumed that participants start in a new forest with a given time horizon $n$ and then reduce their horizon on each day by one. If the horizon has reached one (i.e., $n = 1$), it will remain one. Supplementary Fig. 10 provides an illustration of how the proportions of actions prescribed by the optimal policy change according to different finite horizons.

Since the optimal policies binarize the value differences over the two choice options (either foraging or waiting is better, or they are exactly the same), they do not allow for variability in the decision process (i.e., in some cases waiting and foraging entail large value differences whereas in other cases the two choice options have quite similar values). We therefore used the continuous value differences between the two choice options as predictors of participants' choices, RTs, and fMRI data. For brevity, we often use the term "optimal policy" to refer to the value differences between the foraging and waiting (according to the normative horizon of 5 steps in our task).

All calculations were carried out in MATLAB.

**Instructions and task**. Participants received detailed written task instructions, in which the different task components were introduced step by step (Supplementary Note 1). For each mini-block of trials called "forest," participants were first presented with a screen that depicted the foraging options for the two weather types (3.5 s; Fig. 1). Foraging options were illustrated as a grid with ten subfields. Each of the ten subfields had a probability of 0.1 to become realized. The number of colored dots within the subfields indicated the foraging costs (always 2 points) or foraging gains (ranging from 0 to 4). Sides were counterbalanced for the two weather types. An energy bar at the bottom of the screen depicted the starting energy state (which was 2, 3, or 4 points). After a fixation period (3.5 s), participants had to choose between one of the two foraging options (which represented either good or bad weather) and waiting, which was symbolized by a single dot depicting the sure energy loss of one point. Sides were counterbalanced for the two options. Participants had a maximum of 2 s to make their choice, which was then indicated by an asterisk. If they failed to respond or pressed a wrong button, the words "Too slow or wrong button" appeared. After an interval of 1 s, participants saw the outcome of their choice (1 s): If they had foraged one of the ten subfields turned yellow and the corresponding energy points were added to or subtracted from their energy bar. If they had waited, one point was subtracted from the energy bar. After a variable fixation interval (between 0.5 and 3.8 s), a new day or a new forest was depicted. Participants were not explicitly cued about the current day within a forest; but they knew that they always remained a maximum of five days within a forest (so they could count down the number of days when entering a new forest). If participants had died from starvation, an empty energy bar was depicted and they were asked to

press one of the two buttons to elicit a motor response. The task was presented using the MATLAB toolbox Cogent (www.vislab.ucl.ac.uk).

Participants performed a short first training session of four forests with five days. To ensure that participants could get accustomed to looking five days ahead, they performed a second training session, in which they remained for five days in each of 24 forests. Participants performed ten sessions of the actual experimental task either on a PC (behavioral sample) or in the MR scanner (fMRI sample). Participants were incentivized to avoid starvation over five days in a forest. To strike a balance between design efficiency and the requirement to make participants integrate over a number of future time steps, the number of days in a forest followed an exponential distribution with a mean of 2.5 resulting in 40 days in 24 forests per session. Importantly, participants were instructed to always consider staying five days in a forest and were monetarily incentivized accordingly. In the end, we randomly selected one of the 24 forests for each of the ten sessions. To determine participants' payment, they were again presented with the two weather types and their current energy state and completed the 5 days within these forests. For each forest in which they survived (regardless of the precise energy state) they received additional CHF 1.50.

**Analyses of choice data and model comparison procedures**. Of the total possible number of 400 days, participants in the behavioral sample had reached the starvation state on $22.2 \pm 6.0$ trials (behavioral sample) and $21.1 \pm 5.9$ trials (fMRI sample). In the remaining trials they did not make a response within the time limit in $8.6 \pm 10.7$ trials (behavioral sample) and $6.4 \pm 8.6$ trials (fMRI sample). This left $369.1 \pm 13.0$ trials (behavioral sample) and $368.3 \pm 18.8$ trials (fMRI sample) for analyses.

We used logistic regression models (implemented in the MATLAB function mnrfit) of the following form:

$$p_{forage} = \frac{1}{1 + e^{-DV}},$$

with the following form of the decision variable DV:

$$DV = \beta_0 + \beta_1 * policy.$$

As policy, we first considered the following potential optimal or heuristic policies (see Table 1 and Table 2 for annotated lists): (1) the optimal policy according to a time horizon of 5 days (h-5). In contrast to the optimal policy, the following variables should be regarded as heuristics since they do not necessitate integration over the relevant future time points: (2) momentary probability of foraging success $p$; (3) momentary magnitude of foraging gain $g$; (4) expected value (EV) of foraging; (5) current continuous energy state $s$ (ranging from 1 to 5 points); and (6) binary energy state (indicating whether waiting would lead to sure death or not). Since waiting leads to sure death when the continuous energy state is one, the optimal policy never prescribes waiting in these situations (i.e., even when the probability of foraging success is small, the optimal policy always favors a non-zero starvation probability). Therefore, at a continuous energy state of one the prescriptions of the binary energy state variable are always in line with the optimal policy. We also included the following variables: (7) current weather type; (8) number of days past in a forest; (9) change between past and current energy states (within and across forests); and (10) a type of "win-stay-lose-shift" (WSLS) strategy, which prescribes foraging if the energy state increased in the past trial (i.e., after a foraging win) and waiting if the energy state decreased (i.e., after waiting and after a foraging loss). In other words, WSLS is a binarized version of the change between past and current energy states. Additionally, we tested for a myopic optimal policy: (11) the optimal policy according to a time horizon of 1 day (h-1).

Since the probability of foraging success emerged as best single predictor of participants' choices, we tested whether any other decision variable explained remaining variance in models that included the probability of foraging success, $p$, as first predictor and one of the other policies as second predictor (Supplementary Tables 1, 2):

$$DV = \beta_0 + \beta_1 * p + \beta_2 * policy.$$

According to the same logic, we tested all possible models with two predictors (Supplementary Tables 3, 4) and additionally models with three predictors (Supplementary Tables 1, 2). We also tested interaction models of the general form (Supplementary Tables 1, 2):

$$DV = \beta_0 + \beta_1 * p + \beta_2 * policy + \beta_3 * p * policy.$$

For each model we approximated model evidence by calculating the Bayesian Information Criterion (BIC), which penalizes model complexity. We performed both fixed-effects and random-effects analyses. The latter assume that different participants may use different models. We used the Bayesian model selection (BMS) procedure implemented in SPM12 (http://www.fil.ion.ucl.ac.uk/spm/) to calculate protected exceedance probabilities, which measure how likely it is that any given model is more frequent than all other models in the population[53].

**Calculation of variables related to participants' choices**. Both RTs and brain activity may also relate to the choice uncertainties of the employed decision variables. Choice uncertainties can be quantified by the logistic functions that relate participants' empirical choices to the decision variables (cf. Fig. 3; Supplementary Fig. 3). That is, choice uncertainty is higher at the inflection point of the logistic function (i.e., at the point at which participants are indifferent between the two choice options) than at the ranges of the decision variable where the values of the logistic function are close to zero (prescribing waiting) or close to one (prescribing foraging). As metrics of choice uncertainties, we therefore took the derivatives the logistic functions related to the two variables included in the winning model of participants' choice: the probabilities of foraging success and the value differences according to the optimal policy (with a horizon of 5 days). To base the quantification of choice uncertainties for the fMRI sample on independent data, we used the mean parameter estimates of the choice models from the behavioral sample to derive the logistic functions.

To investigate the interplay between the two variables of the winning choice model, we were specifically interested in the discrepancies in the prescriptions of the two decision variables. Therefore, we took the absolute differences between the predicted choices under the heuristic of using the probabilities of foraging success and under the value differences according to the optimal policy. That is, we calculated the absolute differences between the points on the logistic functions derived from the mean parameter estimates of the choice models from the behavioral sample. We refer to this metric as "discrepancy" between the two policies.

We included choice uncertainties and discrepancies as trial-by-trial estimates in RT and fMRI models (see below). Additionally, we explored whether choice uncertainties and discrepancy guided the use of the decision variables included in the winning model. Specifically, we tested the following interaction models (Supplementary Tables 1, 2):

$$DV = \beta_0 + \beta_1 * p * (1 - \text{choice uncertainty of } p) + \beta_2 * \text{optimal policy}$$

$$DV = \beta_0 + \beta_1 * p + \beta_2 * \text{optimal policy} \\ * (1 - \text{choice uncertainty of optimal policy})$$

$$DV = \beta_0 + \beta_1 * p * (1 - \text{choice uncertainty of } p) + \beta_2 * \text{optimal policy} \\ * (1 - \text{choice uncertainty of optimal policy})$$

$$DV = \beta_0 + \beta_1 * p * \text{discrepancy} + \beta_2 * \text{optimal policy}$$

$$DV = \beta_0 + \beta_1 * p + \beta_2 * \text{optimal policy} * \text{discrepancy}.$$

For additional analyses, we used the relevant logistic functions derived from mean parameter estimates (from the behavioral sample) to select trials in which the heuristic policies of using the probability of foraging success and of using the EV made opposite prescriptions for choice. That is, we binarized the prescriptions of the two policies by splitting them into those above and below the midpoint of 0.5. Logistic functions were derived from the mean parameter estimates of the behavioral sample (Supplementary Note 2).

To analyze subsets of trials, in which different horizons of the optimal policy made opposing prescriptions, we simply selected trials in which the (a priori computable) optimal policy for a horizon of 5 time steps prescribed foraging and the optimal policy for another time horizon prescribed waiting; or vice versa (Supplementary Table 5; Supplementary Fig. 10).

**Models of RTs**. We analyzed log-transformed RTs using linear mixed effects models as implemented in the R package lmer[54] (http://cran.r-project.org/web/packages/lme4/index.html). The independent variables in the mixed effects model were the probabilities of foraging success, the value differences of foraging vs. waiting according to the optimal policy (with a horizon of five days), the choice uncertainties of the two policies, and the discrepancies between the prescriptions of the two policies. Random effects for participants included a random intercept and random slopes for all variables. Since we had no hypotheses for interactions, we did not include any interaction terms as fixed- or random-effects. Significance levels of the fixed-effects were determined by performing log-likelihood tests, which compared the full model to models without the respective factor (Supplementary Table 6). Additionally, we performed the same analyses using untransformed RTs, which resulted in qualitatively similar results (mean ± SD of the skewness of the RT distributions: behavioral sample: untransformed 0.751 ± 0.404; log-transformed: −0.067 ± 0.928; fMRI sample: untransformed: 0.873 ± 0.335; log-transformed −0.038 ± 0.843).

To compare RTs in a subset of trials in which the heuristic policy of using the probabilities of foraging success and the value differences according to the optimal policy made opposing prescriptions, we binarized the prescriptions of the two policies by splitting them into those above and below the midpoint of 0.5. Logistic functions were derived from the mean parameter estimates of the behavioral sample. We then selected trials in which the two policies differed (mean ± SD of resulting number of

trials per participant: fMRI sample: 70.4 ± 8.1; behavioral sample: 70.4 ± 6.8; the minimum number of trials included per participant was 47 trials).

**MRI data acquisition**. Data were recorded in a 3 T (Philips Achieva, Best, The Netherlands) MRI scanner using a 32-channel head coil. Functional images were recorded using a T2*-weighted echo-planar imaging (EPI) sequence (TR 2.1 s; TE 30 ms; flip angle 80°). A total of 37 axial slices were sampled for whole brain coverage (matrix size 96 × 96; in-plane resolution 2.5 × 2.5 mm²; slice thickness 2.8 mm; 0.5 mm gap between slices; slice tilt 0°). Imaging data were acquired in ten runs of 170 volumes each. The first five volumes of each run were discarded to obtain steady-state longitudinal magnetization. Field maps were acquired with a double echo gradient echo field map sequence, using 32 slices covering the whole head (TR 349.11 ms; TE 4.099 and 7.099 ms; matrix size, 80 × 80; in-plane resolution 3 × 3 mm²; slice thickness 3 mm; 0.5 mm gap between slices; slice tilt 0°). Anatomical images were acquired using a T1-weighted scan (FoV 255 × 255 × 180 mm; voxel size 1 × 1 × 1 mm³).

**Preprocessing and nuisance regressors**. All fMRI analyses were performed in SPM12. The FieldMap toolbox was used to correct for geometric distortions caused by susceptibility-induced field inhomogeneities. Preprocessing of EPI data included rigid-body realignment to correct for head movement, unwarping, and slice time correction. EPI images were then coregistered to the individual's T1 weighted image using a 12-parameter affine transformation and normalized to the Montreal Neurological Institute (MNI) T1 reference brain template. Normalized images were smoothed with an isotropic 8 mm full width at half-maximum Gaussian kernel.

We corrected for physiological noise using RETROICOR as implemented the MATLAB PhysIO toolbox[55, 56] (open source code available as part of the TAPAS software collection: http://www.translationalneuromodeling.org/tapas/). We collected electrocardiogram (ECG), pulse oximeter, and breathing belt data during scanning. After quality checks, we used ECG and breathing belt data for noise correction in 19 participants, pulse oximeter and breathing belt data in two participants, and no physiological noise correction in six participants. The corresponding confound regressors as well as the six motion correction parameters estimated from the realignment procedure were entered as covariates of no interest. Regressors were convolved with the canonical HRF and low frequency drifts were excluded using a high-pass filter with a 128 s cutoff.

**General linear models**. The three distinct phases of the task (forest, choice, and outcome phases; see Fig. 1) were entered as events with a duration of 0 s (i.e., as stick functions) into the GLMs. Choice and outcome phases in which participants had starved or for which they did not reply were not explicitly modeled.

We were mostly interested in the choice phase and ran a main GLM with a combination of variables that emerged in our analyses of behavioral and RT data. Specifically, the following variables were entered on a trial-by-trial basis as parametric modulators of the choice phase: The probabilities of foraging success, the value differences of foraging vs. waiting according to the optimal policy (with a horizon of 5 days), the choice uncertainties of the two policies, the discrepancies between the prescriptions of the two policies, and RTs. A second GLM included participants' choices (as a binary variable) as a parametric modulator in addition to the variables entered in the main GLM. A third GLM only included participants' choices as a single parametric modulator. A fourth GLM included parametric modulators in terms of chosen (and unchosen) options (and not in terms of the presented options as in the first three GLMs). Specifically, this fourth GLM contained the a parametric modulator for the chosen values of the employed heuristic (i.e., the current probability of foraging success for trials in which the foraging option was chosen and a probability of zero when the waiting option was chosen) and a parametric modulator for the chosen values according to the optimal policy (i.e., the current values of the foraging or the waiting options) as well as parametric modulators for the corresponding unchosen values along with parametric modulators for participants' choices and RTs.

In all four GLMs, the forest phase was parametrically modulated by the current energy state and the overall starvation probability across five days. The outcome phase was parametrically modulated by the change in energy state.

We report analyses in which regressors competed for variance (i.e., without serial orthogonalization). In the main GLM, the maximum average shared variance was below 0.5 for all combinations of regressors.

We performed a second-level one-sample t-test on contrast images from all participants. All reported clusters are family-wise error (FWE) corrected for multiple comparisons at $p < 0.05$ using the SPM random field theory based approach. The cluster-defining threshold was $p < 0.001$. At this voxel-inclusion threshold the random-field theory approach in SPM correctly controls the false positive rate[57].

**Data availability**. The behavioral data that support the findings of this study are publicly available at https://figshare.com/s/1a4d75cb4176a3fef040. The neuroimaging data that support the findings of this study are publicly available at https://neurovault.org/collections/3242/.

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

## Acknowledgements

The Wellcome Trust Centre for Neuroimaging is supported by a strategic grant from the Wellcome Trust [09,1593/Z/10/Z]. C.W.K. was supported by the SFB TRR 169 during final stages of manuscript preparation. We thank Thorsten Kahnt, Gabriela Rosenblau, and Robb B. Rutledge for insightful comments on an earlier version of this manuscript as well as Giuseppe Castegnetti, Lars Kasper, Saurabh Khemka, Tessa Rusch, Matthias Staib, and Athina Tzovara for discussions, help with data acquisition, and technical support.

## Author contributions

C.W.K. and D.R.B. designed the experiment, developed the analysis procedures, and wrote the paper. C.W.K. collected and analyzed the data.

## Additional information

**Competing interests:** The authors declare no competing financial interests.

