## [Peer Review File · Nature Communications]

Reviewers' comments:

Reviewer #1 (Remarks to the Author):

Korn and Bach study human choices and neural signals in a deep, sequential foraging problem. They find evidence for contributions of both simple heuristics and deeper, more exact computation; then evidence in reaction times and BOLD that speaks to their competition and the arbitration between them. I found this to be a terrifically interesting and promising study. The task and models intersect issues that have been studied in a number of other domains (typically, learning tasks) but offer a fresh approach in a number of respects that provide both convergent and new evidence and approaches, and promise for followups. That said, the substantial ambition of this project is also accompanied by equally substantial challenges and I think there is a certain overreach as yet. Without diminishing my enthusiasm for the project, I have two major concerns about the paper, one philosophical and one technical. I very much hope the authors are able to make some headway on them so as to deliver on the promise of the study.

1. I do not think enough data are yet presented to make the reader confident in the article's basic framing that the results are best understood as the combination of two discrete strategies – one simple heuristic plus the optimal policy. As the authors are no doubt well aware, many different decision variables and approximate decision variables will be highly mutually correlated. Indeed the optimal policy (and its other cousins) is itself a deterministic (nonlinear but quite possibly monotonic) function of the heuristic variables probability, gain, and energy. The probability heuristic and the optimal policy might be thought of as two endpoints spanning the range between sloppy and exact solutions, which might in part explain their success together. A few thoughts on this point:

* I'm not crazy about the stepwise model-building approach (first greedily locking in p as the best among a set of possible candidates, then adding other variables only together with it). It certainly does not demonstrate that the best combination of any two variables is the pair it finds, nor that two is better than three, etc. Indeed given that all of the candidates are highly correlated I wouldn't necessarily expect to be able to find a unique combination of variables that best explained the behavior better than all other combinations. It is probably just too much to expect BIC etc. to adjudicate among variables in this type of situation.

Further, there are a number of promising heuristic candidates that weren't considered in the first wave (but are only measured, if at all, with p already in the model). These include the horizon-1 optimum, or the EV, probability times gain, both of which are related to the probability but also form more of a continuum with the optimal policy. Low horizon myopic optimal policies seem particularly nice options, but may not have fit well since they were only tested with p (the ultimate myopic variable) already in the model. A possible hypothesis is a variable depth optimization story, ranging between (or combining) horizon-1 and deeper solutions; this is also similar to the recent Keramati and Dayan study.

* This is a very difficult problem and I don't have great advice for the authors how to overcome it. It is probably too much to expect that it be overcome completely, and to some

extent it can also be finessed by more judicious, cautious, interpretation and more caveats. But I think one step that absolutely needs to be taken is to more concretely demonstrate how the choice behavior would change as a function of some of these different candidates, and how (and whether) they can be distinguished in choice space. The plots of optimal vs. actual policies in Supp fig 1 are a start, but it doesn't make clear how the behavior can distinguish these from the more myopic heuristics, or all the horizons from each other. How many of the trials actually distinguish these?

* I of course understand the points about reaction times and fMRI correlates of the different decision variables, which provide some (quite tantalizing) support for the authors' interpretation. But again in the context of a large family of highly correlated candidates, and without more exhaustively comparing the possibilities and more carefully demonstrating their separability in data space I think it's hard to make too much of different neural correlates and RT effects. I would note that the RT issue cuts both ways in the paper – on the one hand we are told that only parallel computation can explain the RT choice difficulty effects (I think this is an awfully strong conclusion to draw so categorically, though it is certainly also a neat piece of evidence and a novel approach) – but on the other hand we are told that heuristic use is faster, which should not be true in a fully parallel model. I think it's quite likely that some progressive computation model like Keramati and Dayan's, where approximations are refined if some features of the problem suggest that deeper search is worthwhile, might in principle be able to explain both results.

2. My technical issue is that as far as I can tell, the optimal policies used here are not computed correctly. More specifically, there are really two versions of the task used here – one with stochastic termination (i.e. exponentially distributed time in the forest), and the other a fixed horizon-5 game. Neither of these corresponds to the "optimal" policy computed here, which isn't really optimal for any problem as far as I can tell.

For the fixed, finite horizon-5 game, the key issue is that the optimal action choice will in general depend on the number of steps remaining, and so it is incorrect to use a fixed probability p of foraging (and take the transition matrices to the fifth power). For instance, when there is only one move left, if there is still sufficient energy left, then either forage or wait will be equally successful in terms of the objective function. But this might not be true at earlier steps in the forest with the same energy and weather. To find the true optimal policy, one would normally use dynamic programming, working backward from the last move and treating the different horizon steps as separate states with separate optimal policies. (That is, it is still a Markov problem, but only in a state space that distinguishes the steps toward the horizon.)

Conversely, if the game terminated stochastically (exponentially distributed game length) with some probability which I will call $(1-\gamma)$ then the states are all equivalent independent of depth – the optimal policy is indeed fixed over periods within the forest, as a function of energy remaining and weather. However, in this case it is also not correct to optimize with respect to the fifth power of the transition matrix – one must account for the distribution of termination times. This is equivalent to an exponentially discounted, (in

principle) infinite horizon problem, where the event of the task terminating with survival is, effectively, a second absorbing state (with 0 reward), which equivalently introduces a discount factor of γ over the infinite horizon version of the problem. This can then again be computed with value iteration / dynamic programming. This is discussed in standard texts, e.g. Sutton & Barto, or Puterman's MDP book. (Note that if the exponential distribution is truncated, e.g. at maximum 5 steps, then this would violate the Markov property and a finite horizon model like the one above would actually apply: the policy would again depend on the step. The paper isn't really clear what was done here.)

I doubt these details will make too much difference, but I do think this is an error that pervades essentially all the results, which needs to be corrected and the results need to be revisited.

Two side points arise from these considerations. First, while it is a theorem that all MDPs have a deterministic optimal policy (thus it is no accident that the optimal policies here are deterministic), that policy is not in general unique. Here, for instance, I strongly suspect there are states where foraging and waiting both ensure survival for sure (e.g. in the last move as discussed above) and here any policy, including stochastic ones, will be equivalently optimal. The analysis needs to take this into account. Second, the discussion of value iteration in the supplement seems a bit off. As I hope I have described above, when properly framed, the original avoid-death problem is nothing other than an MDP which can be solved by dynamic programming, i.e. value (or policy) iteration over the appropriate state space and the appropriate finite or infinite horizon. Thus the discussion in the supplement appears not to really be about the solution method, but about the nature of the objective (ie, reward) function, i.e. maximizing some new intake function vs avoiding starvation.

Reviewer #2 (Remarks to the Author):

Korn and Bach build on two emerging lines of research: one focused on planning across deep decision trees, and the other on using ethologically relevant paradigms to understand the neural mechanisms of decision making. The authors develop a task that probes foraging behavior in a simulated environment where optimal performance requires planning through a deep and probabilistic tree of possible future states. The authors collect fMRI data during task performance and use the behavioral and imaging data to support the following key claims:

- 1) Human foraging behavior deviates from optimal planning and is better explained by a heuristic (win probability) that is occasionally abandoned in favor of an optimal solution when the heuristic is difficult to implement.

- 2) Both optimal and heuristic strategies are computed in the brain in parallel and arbitrated in medial prefrontal cortex, with the optimal policy scaling with BOLD activity in anterior MPFC, conflict between optimal and heuristic policies scaling with BOLD signal in dorsal

MPFC, and the difficulty of the heuristic model reflected in the BOLD signal in anterior VMPFC and mid cingulate.

My feelings about this paper are somewhat mixed; on one hand I feel that the authors have taken on a difficult question and made strides toward pinning down an answer. However, on the other hand, in part because of the difficulty of the question and the complexity of the task, I am unconvinced that the authors' interpretation of the data is the only, or even the best, one. In particular I have concerns regarding the interpretation of the behavior as emerging from two separate strategies implemented in parallel and reflected computed through different neural systems. This interpretation is at odds with the standard rationale for why heuristics are adopted, and thus, if true, would constitute a surprising and important finding. But based on the data presented by the authors, I think it is more likely that subjects are implementing a serial strategy in which they first pay attention to the foraging success probability, and then, if not sufficiently swayed by this information, move on to consider the current "energy state". This interpretation would be more in line with standard ideas about multi-attribute decision-making, as far as I can tell in line with the behavioral analyses, and would require major reinterpretation of the fMRI results. A complete listing of my concerns is below:

Major Issues:

1) It is not clear that the primary behavioral claims are supported by the data. As far as I can tell, the authors do not compare a model of the decision variable from the optimal model (difference in survival across forage/wait) against the heuristic model. This is an important comparison. The binary optimal policy variable that the authors use for comparison does not include any information about the magnitude of model preferences, and thus has no way of accounting for noise in the decision process. It is very clear from the behavioral figures (Fig 2 & supplementary figure 1) that subjects change behavior according to the energy state suggesting that the suggested heuristic (foraging success probability) could not provide a good fit to the data. In general, the authors have tested a large number of models, but done very little to show the extent to which these models capture the subject behavior. Posterior predictive checks to show that the authors preferred models can capture the behaviors shown in Fig 2 and supplementary figure 1 would be helpful in this regard. The behavioral figures clearly indicate that behavior changes as a function of energy state, meaning that the heuristic strategy favored by the authors does not capture a key behavioral feature of the data.

2) I am not fully convinced by the authors' claim that the optimal and heuristic computations are done in parallel. I typically think of a heuristic as offering computational savings afforded by avoiding costly computations associated with more complex solutions. Here the authors are saying that people spend the computational resources to evaluate both strategies... but then choose a worse one much of the time. The authors' state on lines 212-213:

"To address this issue, we calculated optimal and heuristic policies and analyzed whether a discrepancy between the two influenced participants' behavior. This would only occur if both

were computed in parallel, on the same trials.”

The authors find that RT increases (somewhat non-monotonically) with the difference in choice-probabilities from the optimal and heuristic models. But this could come about in many ways other than parallel computation. For example, if the models diverge for particular trials that are systematically slower. My guess is that this is the case. In particular, I think that participants are likely slowest when they are required to combine two pieces of information to make a successful decision (eg. energy and probability). This would occur for low probabilities when energy is low (1) or for high probabilities when energy is high (5). These are also conditions when model predictions would diverge most, since the probability assessment gets overridden in the optimal model by the proximity to the energy boundary. In order to address this, the authors could plot RT as a function of conditions, which, along with the posterior predictive checks described above, would test this alternative explanation for the RT relationship (if the relationship is explained by similarity across task conditions, it is consistent with my interpretation). Clarifying this point seems critical to interpreting data from the neural GLM, since the authors include both optimal and heuristic terms as modulators under the assumption that they are separately represented in the brain.

3) As far as I can tell, the fMRI GLM does not include choice (forage/wait), but the other variables in the model are highly correlated with it. I can certainly understand why the authors would be reluctant to include choice along with the other terms, given that the input correlations in the model are already high (also, as a side point, it would be useful if VIF were listed instead of max pairwise shared variance). However, an interpretation of the current model may be that a large swath of the brain responds to the decision to forage... and that, through noise and thresholding, the authors have identified separate clusters for optimal policy and probability of foraging success regressors. To this effect, assuming that correlations with forage/wait decisions are high, I would recommend that the authors first do a whole brain analysis to identify regions responsive to choice (which is presumably informed by both measures), and test the encoding of each model based term within the ROIs responsive to the decision itself. It would also be useful if the authors could test whether representations of the terms are significantly different from one another through a normalized subtractive contrast.

4) One important point made by the authors is that the conflict signal that they identify in ACC/DMPFC is conflict across different policies (optimal/heuristic) rather than a signal related to overall decision difficulty/uncertainty. However, the authors do not include the best possible decision difficulty regressor in the model, and given the recent heated debate on the issue, it seems important that the authors clearly show that they are explaining variance beyond simple choice difficulty. If the authors want to make claims about “controller conflict” above and beyond simpler measures of decision difficulty, they should estimate choice difficulty using the best-logistic regression model and include this term as regressor in their GLM. Given that the “probability of foraging under optimal policy” regressor also loads on a similar region, it would be good to know something about the overall rates of foraging (are foraging decisions or wait decisions more frequent?)

4) As in the behavioral analyses, the authors have portrayed the optimal policy as binary and the heuristic one to be continuous. I see no reason why this should be the case, and I wonder to what extent the differences in the variance explained by these two terms could simply be explained by this analysis decision. If the authors are looking for a representation of an optimal decision variable, it seems that they should include the difference in projected starvation for forage versus stay decisions in the neural GLM, rather than a binary variable that reflects the optimal choice.

5) The authors should include a methods section in the main text that conveys the information necessary to interpret the results.

Minor comments

The authors describe the process of arbitrating between the heuristic and optimal strategies as depending on the difficulty of implementing the heuristic strategy. However, it seems equally valid (and perhaps more intuitive) to say that the heuristic strategy is abandoned when it provides uncertain information. This interpretation is completely in line with other theories of how the brain combines information from segregated modules in other domains (eg. combining RL with working memory, combining model-based with model free information).

The authors seem to interpret the model fits more in terms of their global meanings (optimal versus heuristic) and less in terms of what aspects of the data they actually capture. I think that the paper would be much easier to understand if they focused on the latter, rather than the former.

Line 21: easy-to-?

Figure 2: This figure would be much improved if spent less real-estate on which models were fit and more on how well they fit. For example, showing the heuristic, optimal (using the correct DV, as described above), and heuristic and optimal models along with posterior predictive checks to show how they fit the actual choice curves (figure 2&supplementary figure 1) would provide more insight into the modeling than all of these bar graphs.

Figure 4A, right: I have no idea what the x-axis is here. It is labeled "prescriptions of the optimal policy" but surely that is the y-axis. As far as I understand the implementation of "optimal policy" it was binary, and thus the lines seem somewhat superfluous. Though, as I said above, I think that a continuous decision variable would be more reasonable.

Line 288: It seems that the neural results would be easier to digest if there were some description of the GLM that was used to interrogate the imaging data at the beginning of this section.

Line 293: Maybe I missed something, but it is unclear to me why the probab

Given recent work showing slightly elevated family-wise error rates when using the multiple

comparisons techniques described by the authors 1, it would be good if the authors could validate clusters using a permutation testing procedure 2.

Supplementary Figure 1: this information should be included in the main text.

1. Eklund, A., Nichols, T. E. & Knutsson, H. Cluster failure: Why fMRI inferences for spatial extent have inflated false-positive rates. *Proceedings of the National Academy of Sciences* 113, 7900–7905 (2016).
2. Nichols, T. E. & Holmes, A. P. Nonparametric permutation tests for functional neuroimaging: a primer with examples. *Hum. Brain Mapp.* 15, 1–25 (2002).

Reviewer #3 (Remarks to the Author):

This manuscript is concerned with a 'homeostatic' sequential decision making task and its underlying neural correlates. As such there is some novelty in both the decision models and neural findings. Particularly, the potential of having interdependent but partly distinct computations of more heuristic and more optimal processes emerge over time during sequential choice, with more optimal behavioral flexibility being driven by ACC, is exciting. It also adds to the current debate over the nature of value signals in foraging-like decision tasks, rejecting a view of one unitary factor driving activity.

However, the framing and interpretation of results are not always straightforward and should be improved upon. Furthermore, specifically the description of their findings in regards to the current literature are quite selective and convoluted, neglecting to highlight some major differences between the current study any the majority of other decision tasks, while also trying to adopt terminology from many different approaches, presumably in an attempt to keep many different factions happy. While I also had some methodological concerns, about half of my comments relate to the framing. Therefore, Points 1-6 and 8 should all mostly be addressable through textual changes i.e. improvement in framing. To help with this, I tried to point out specific sections in the manuscript that could be reframed/reworded (These specific recommendations also makes the review appear unusually long)

Surprisingly the authors appear to argue a heuristic and optimal policy is always computed in parallel, but not sufficiently address the question of why to even bother with heuristics if an optimal computation is possible. The alternative explanation that optimal policies are only computed selectively, particularly when heuristics fail and thus in part ACC being more engaged when there is a need for optimal computations is wholly compatible with their results and more likely.

Furthermore, their conflict or "difficulty" representation effects can be described equally well with a more neurophysiologically grounded explanation, explaining decision making using concept such as evidence accumulation and mutual inhibition, without having to resort to a representation of a psychological experience of conflict or explicit signaling of difficulty itself. More importantly, there are many different aspects of the task that could be labeled

difficulty or conflict including decision horizon, and therefore a description of the results in terms of absolute value differences or choice uncertainty/value driven choice probability would be more appropriate.

(More detailed comments below)

Major Comments:

1) The current manuscript is relatively unusual within the decision domain, as value is solely based on whether participants "survive" a series of five sequential decisions after which they receive a fixed point independent reward. As such the point of the task is to maximize the probability of "survival" or homeostasis rather than maximizing the overall number of points in the experiments, as is normally optimal in other decision experiments. Therefore it is tricky to compare this study's value regressors with the ones from other studies, which should be sufficiently highlighted. Furthermore, it might deserve a mention in the title and definitely in the abstract. The task is furthermore lacking some of the sequential elements of other experiments as participants accumulate points at every step. However, the authors note that already in the manuscript.

2) As mentioned above, the authors should change the language to follow more of a value terminology, as this is closer to the data in a value based decision task and allows for the use of more mechanistic descriptions related to increasingly detailed models of decision making as an evidence accumulation process. E.g. value, choice probability and uncertainty make more sense than decision difficulty as the later is a meta process/judgment beyond the decision process itself. This is important as the circuit is presumably trying to make a decision, and adapt optimally when a heuristic value estimate is insufficient. Related to this the concept of comparing heuristic with optimal in parallel every trial is strange. Why should the agent bother with a heuristic if it eventually compares it the optimal anyway? At some point in the manuscript this statement is qualified and it is suggested that the optimal is partly conditional on the heuristic, but it is not nearly clear enough in many parts of the manuscript (see also point 9).

Related to this, the authors have a quite contorted description P17, L282-4 "This provides crucial evidence that both heuristic and an optimal policies determined participants behavior, with the relative contribution of the two depending on the difficulty or reliability of the heuristic". However, the heuristic isn't more difficult, but rather does not guide behavior in one way or other, i.e. there is no clear value in either response. A better description of the results would be that lower order decision uncertainty or lack of reliability forces the heuristic system to engage other systems and compute higher order value instead to increase choice certainty and also accuracy.

Additionally, the ecological approach overall suggests that the animal is trying to gather things of value, and ensure survival. Therefore, the terminology should be guided by value, not split into different forms of difficulty, particularly as the whole concept of conflict and difficulty is that it is an atomic unit that can't be split.

Furthermore to prove the psychological phenomenon of difficulty representation, one would

have to show any changes that are commonly associated with difficulty, such as increases of reaction time through non-value related manipulations, induce equivalent effects (unlikely given Stoll et al. [2016, Nature Communication]). If that is not done, the resulting neural findings should more accurately be described as value and certainty related, as that is what is being explicitly manipulated. Additionally, there is no point in using the term difficulty, which is meant to be a catch-all for many different factors combined in just how hard it is and then discuss only one bit of the task. If one talks about difficulty it has to be overall difficulty of doing the task or nothing.

Looking at the text itself, specifically, P21 L363 "Difficulty was represented in several brain regions..." is a fallacy of assuming representation from a correlation. More specifically, only because the signal was larger when no quick clear decision could be made that doesn't mean there is a representation of difficulty. Any competitive system should be sensitive to value similarity and uncertainty. Also, it isn't the difficulty of the heuristic policy.

3) Furthermore, I think the concepts of competition and accumulation of evidence as well as decision thresholds are far better suited to describe the data. In other words, if the agent hasn't made a decision quickly with a heuristic because there is no clear value or a lack of a clearly superior option, additional mechanisms such as dACC are engaged. Presumably, although I couldn't find it in the manuscript there is also an effect of reaction time in ACC, as many studies have found decision length associated activity with other factors riding on top. This could then mean that ACC is increasingly active when heuristic certainty is low and therefore decisions aren't made quickly (therefore also a larger effect of heuristic on reaction time), eventually coding the relative gain from using a better understanding of the task structure (task model etc). This is encapsulated both by the neural effects of the difference in choice probability between heuristic and optimal, and the overlapping increase by optimal policy value. This in itself is potentially a very interesting finding if described in those terms.

Related to the idea of overlapping or interrelated accumulation processes, the repeated statement of Parallel policies is slightly misleading e.g. P20 L336-8 "computed in parallel-although when they are conflicting, only one of them can be implemented. " As outlined above (and further in point 4) it is very tricky to show parallel processes and I think the data can be explained without going hard on the idea of always both policies being computed.

Furthermore, related to both aspects of this point P23 L433-5 "Brain activity associated with conflict between heuristic and optimal policies points towards a mechanism of parallel computation"

Could be more mechanistically described as possibly competitive value/decision evidence accumulation processes, although there is also the possibility of one integrated process that changes over time as additional types of information get fed in.

P20 L340-2 "The higher the decision difficulty of the heuristic policy the more the optimal policy was applied..." Could be rephrased as rather saying when neither option is clearly more valuable on a heuristic level, then a more sophisticated computation is made/ a more

optimal policy is computed.

4) More generally, the authors substantiate their parallel processing claim mostly on one RT analysis. There is weak evidence for the pure “conflict” of two strategies rather than a process or interrelated processes, which change over time and between trials. Rather it could be an increasingly sophisticated computation over time, possibly with an increasing probability of optimal policy computation being initiated meaning computations aren't completely sequential either.

Independently of that concern, showing truly parallel computations rather than an averaged mixture is chronically difficult to prove and participants might just have been additionally slowed in parts where the two diverge because that corresponds to the parts of decision space that is misfit or because, although both processes aren't parallel the optimal policy is increasingly more likely to be computed and therefore slow down decisions when the heuristic policy is likely to fail (e.g. long horizons or large dot ranges making p-forage success a poor substitute).

Additionally, it looks like the effect in figure 4 B isn't linearly increasing but only to a point and then decreasing again, suggesting it might be driven by another potentially somewhat related variable like horizon, energy level or trial number in forest. If the authors want to prove truly parallel computations, they should resort to a method with higher temporal resolution or else settle with talking about interrelated, but not necessarily parallel computations.

5) Optimal policy effect in ACC/MPFC is interesting as it suggests positive value related effects do exist in ACC, particularly for the more sophisticated computations. This is important, as the rest of the manuscript reads a little as if there was no positive value effects in ACC. Furthermore, the lack of a strong heuristic difficulty effects or other definitions of difficulty such as horizon length should be mentioned.

6) The negative abs. value difference/ choice probability effects based on p-forage success in perigenual ACC and vmPFC are quite surprising given a wealth of studies showing positive value effects in vmPFC. This should be at least discussed, as it is quite unusual to have this bit of medial prefrontal cortex not activate with value. Furthermore, when looking at the supplementary material, confusingly it says negative p-foraging success, which is not quite the same as choice difficulty using p-foraging success, which was used in the regression. This should definitely be clarified, as a negative effect of foraging success is quite different conceptually than a negative abs. value difference effect/choice p effect.

7) It is a bit strange that p forage success is such a strong driver of the behavioural effects and very close to optimal. It means that the other aspects of value feeding into overall survival probability do not have very large range. From the manuscript it wasn't clear whether there was just little variance in some of the other factors, such as magnitude. It might be nice to see a bit more of the schedule they ran, such as magnitude spread and variance etc..

8) Title should be changed to more appropriately reflect the content of the study (see also comment 1). At the very least, it should become apparent that the study is a sequential

decision task. "Emerging computations of heuristic and optimal decision value in the human cortex" or "The neural signature of heuristic and optimal decision value in homeostatic sequential decisions", might convey this point a bit more (both are only meant as illustrations not firm suggestions. I do think however, the title should be less general)

9) There is an interesting analysis relating to heuristic choice certainty modulating the use of heuristic vs optimal model. However, this suggests rather what I discussed above, a conditional computation of higher order value if and when necessary, not a purely parallel computation. This should be elaborated on and maybe an equivalent neural analysis should be run.

Also, a bit more basic concern, if true, is that it looks from the supplementary methods that an individually fit softmax is used in order to derive the difficulty of the heuristics in order to scale the value of foraging success. If this is true choices are used twice for fitting, making statistics biased and potentially invalid as the difficulty adjustment upscales the part of p-foraging success with the largest changes in decision based on p-foraging success. (See P12 L295 in supplements)

This would also explain why the scaling with difficulty is positive. Conceptually, I would have expected the opposite, i.e. less use of the heuristic information if it is uncertain, relying instead on the optimal policy, while the opposite appears to be the case from the formula. However, I am not so sure about the difficulty effect being positive, although this is how the formula is written as the authors explicitly state the more intuitive correct sign of effect P20 L340-2 "The higher the decision difficulty of the heuristic policy the more the optimal policy was applied..."

10) I think it would be important to run RT and decision regression with all the important/potentially relevant effects, plotting all of them on a group lvl (mean and error bars). This would be 'difficulty' (abs value difference) for heuristic and optimal and 'difficulty' of heuristic vs. optimal on RT and all the potential heuristics for the logistic regression on the decision data. This would give the reader a better sense of the data and how the difference of policies, not the main effects separately, drive RT and the relative effect sizes for aspects of the task that could combine to be the thing going beyond the p-forage heuristic. It might furthermore, be useful to bin the data according both optimal and heuristic separately (e.g. separate lines for different optimal bins and points on x axis for heuristic) to show the different effects and potential scaling of effect size (see point 9).

11) The fMRI part of the manuscript is a bit thin. For example, if they believe the ACC is involved in implementing the optimal policy, shouldn't they run a PPI with the value of the optimal policy to test whether it functionally connects more strongly to other value related areas when it tries to steer the system towards more optimal policies?

Minor Comments:

A) P21 L375 "found neural representation" Isn't this the scaling of the signal by value or a transfer of the same into p forage success? It is unusual to talk about a scalar value signal

as a representation, as that might have come from a pattern analysis etc.. Therefore, I would suggest just describing it as "IPS and frontal pole signals scaled/increased with heuristic value/p-foraging success". Also it should be discussed why this positive effect exists, together with negative effects related to overall survival probability in frontal pole in the forest phase (called starvation probability).

B) I am not sure whether it is mentioned in the manuscript but I couldn't find it. Do subjects need to count the days by themselves or is this cued somehow?

C) P20 L357 " conflict-related brain activity" a bit empty phrase. Also, I don't think the conclusion of parallel rather than interrelated can be proven.

D) How does the pregenual cingulate/vmPFC effect compare to McGuire and Kable's Nat Neuro paper, as they had a simple sequential paradigm and argued for positive value effects in these regions.

E) Both optimal policy positive signals as the other effect are as much in the ACC as other, yet one is called MPFC and other ACC in figure 5, although this is better in the figure legends.

F) As a general note, the authors should consider larger delays between events. As it is, it is impossible to dissociate some of the stages temporally and overall the timing was rather crowded. For their current analyses it is ultimately not too bad, as the important decision event has information that isn't presented before, but other analyses would have been possible with a bit more generous spacing.

G) Participants are strangely bad (Figure 2). Even when they should be at 100% in bad weather for foraging they are essentially random. Is this because the overall p survival is so low that they don't care? Might be useful to plot the p survival for good and bad weather by energy state as the temperature might be effected by the value and fMRI behavioural data might have a higher temperature accentuating that modulation even further..

H) How likely is the heuristic strategy to give higher p-choice estimates than the optimal? Furthermore, is, after fitting the softmax etc one strategy more likely to be more certain the majority of trials? If so, then the unsigned difference converges with a signed variant and it is therefore important to know the degree of correlation between signed and unsigned within every subject.

I) They claim that macroscopically different brain regions encode the two policy but never show the respective other policies effect size in the regions they are referring to making it hard to judge that statement P17 L288-9.

J) The abstract has an incomplete sentence. It says "resort to easy-to-heuristics"

Reviewer #1 (Remarks to the Author):

Korn and Bach study human choices and neural signals in a deep, sequential foraging problem. They find evidence for contributions of both simple heuristics and deeper, more exact computation; then evidence in reaction times and BOLD that speaks to their competition and the arbitration between them. I found this to be a terrifically interesting and promising study. The task and models intersect issues that have been studied in a number of other domains (typically, learning tasks) but offer a fresh approach in a number of respects that provide both convergent and new evidence and approaches, and promise for followups. That said, the substantial ambition of this project is also accompanied by equally substantial challenges and I think there is a certain overreach as yet. Without diminishing my enthusiasm for the project, I have two major concerns about the paper, one philosophical and one technical. I very much hope the authors are able to make some headway on them so as to deliver on the promise of the study.

Thank you very much for the overall positive evaluation of our study. Please see below for details how we addressed your concerns.

- 1. I do not think enough data are yet presented to make the reader confident in the article's basic framing that the results are best understood as the combination of two discrete strategies – one simple heuristic plus the optimal policy. As the authors are no doubt well aware, many different decision variables and approximate decision variables will be highly mutually correlated. Indeed the optimal policy (and its other cousins) is itself a deterministic (nonlinear but quite possibly monotonic) function of the heuristic variables probability, gain, and energy. The probability heuristic and the optimal policy might be thought of as two endpoints spanning the range between sloppy and exact solutions, which might in part explain their success together.**

We entirely agree that the heuristic policy and the optimal policy could be regarded as two possible endpoints of a continuum and that this is one reason why a mixture between them explains behavior in our task. We are confident that after adding new analyses, we can now firmly conclude that no linear combination of two candidate policies and variables explains the data better than the two discrete strategies highlighted in the previous version. Although it is theoretically possible that the neural system computes a yet unknown decision strategy that directly reflects this mixture, such strategies do not follow from the given task variables in an obvious manner. Any such model would thus require higher complexity than the linear combination of strategies considered here.

First of all, we now explicitly mention this notion in the discussion: Page 21:

Participants' choices were thus explained by two metric that can be regarded as points falling on spectrum between sloppy (but easy) and exact (but difficult) solutions.

The reviewer expected a “nonlinear but quite possibly monotonic” relationship between the optimal policy and the different heuristics. To directly address this, we now plot the optimal policy (i.e., the average value differences between the two choice options according to the optimal policy) for each heuristic across all trials (**Fig S1**). As expected by the reviewer, monotonic relationships emerge for several heuristics—notably for the heuristic that explains behavior best (i.e., the probability of foraging success). Still, the optimal policy is not at all identical with any of the heuristics, which is also mirrored in a quite reasonable range of average shared variances between the optimal policy and the other candidate variables (with a maximum of 0.37; shown in **Fig S1**). For some heuristics, no strictly monotonic relationship emerges (i.e., energy states, days past, changes in states).

Fig S1. Relation of optimal policy and heuristic variables

Shared variance between all 10 candidate variables and relationships with the optimal policy (according to the normative horizon of five time steps) with all 9 heuristic variables considered. Data are binned and arranged in the same way as below in **Figs S3 & S4**, which show the empirical relationship between data and fitted model. “Weather type” and “wins-stay-loose-shift” are binary variables. See **Table 1** for a list of all variables.

A few thoughts on this point:

- **I'm not crazy about the stepwise model-building approach (first greedily locking in p as the best among a set of possible candidates, then adding other variables only together with it). It certainly does not demonstrate that the best combination of any two variables is the pair it finds, nor that two is better than three, etc. Indeed given that all of the candidates are highly correlated I wouldn't necessarily expect to be able to find a unique combination of variables that best explained the behavior better than all other combinations. It is probably just too much to expect BIC etc. to adjudicate among variables in this type of situation.**

Thank you for giving us an opportunity to expand on this point. Indeed, the stepwise approach, although motivated by a priori considerations, is not guaranteed to expose the best combination of policies in the set considered. To address this point, we now test all possible pairs of the 10 policies considered, resulting in 45 models. We show that no pair of two policies is better than the model that includes the optimal policy and the winning heuristic (i.e., probability of foraging success). The finding that the original model wins in two independent samples (using both fixed and random effects model comparison procedures) lends robustness to its explanatory power for behavior in our task (**Tables S1 & S2**). Thus, we can now firmly conclude that within the set considered, there exists a unique linear combination of variables that best explains behavior, and this is the combination previously identified in the step-wise approach. We mention this point in the results section (page 12):

The behavioral model that included both the probability of foraging success and the optimal policy also won in an extensive model comparison across all 45 pairs of candidate variables (**Tables S1 & S2**).

We specifically tested for interactions of the most promising heuristics with the winning heuristic and found that the model including p foraging success and the optimal policy clearly outperformed these interaction models (**Table S3**). Page 12:

Additionally, we made sure that models including interactions between the most important heuristics did not provide a better fit than the model with the optimal policy (**Table S3**).

We also report analyses testing whether a linear combination of three variables provides a better explanation of participants' choices than the combination of one heuristic (p foraging success) and the optimal policy. A comparison among models including these two policies and any of the remaining eight candidate models did not provide decisive evidence for a winning model among these eight models (especially when considering protected exceedance probabilities and both samples, **Table S3**). Also, when testing the most promising "three-variables-model" (p foraging success, optimal policy, and energy state), we did not find decisive evidence for this model being better than the winning "two-variable-model" (p foraging success and optimal policy; **Table S3**). Page 12:

In the two samples tested here, there was no decisive evidence that a model with three variables outperformed the model with the probability of foraging success and the optimal policy (**Table S3**). We deem it possible that specific selections of different trial types could identify the use of more complicated models, such as models including a variable for energy states.

To mitigate concerns of high correlations between the different variables, we added a matrix plot with the average shared variance in R^2 across participants (see **Fig S1** above). The highest shared variance was between the newly included expected value (EV) and the possible gain magnitude (0.84). The second highest shared variance was between two variables, which anyway did not explain behavior well and were a priori rather unlikely and do not relate to the optimal policy (changes in states & win-stay-loose-shift). Shared variance between the probability of foraging success and the weather type was 0.51. All others were well below 0.5.

We now emphasize in the introduction that we specifically wanted to test whether participants follow an optimal policy. The prominence of the winning heuristic, probability of foraging success, is

underscored by the fact that it performs second-best in simulations (after the optimal policy). Pages 3-4:

Here, we asked to what degree humans rely on optimal versus heuristic decision policies, and how these are neurally computed, during a sequential decision-making task that challenges decision-makers to integrate over an extended horizon of multiple probabilistic states. We hypothesized optimal policy computation in multimodal regions of the medial prefrontal cortex (MPFC) known to integrate economic decision variables and to evaluate prospective outcomes^{16,17}. To test the interplay between heuristic and optimal policies, we developed a novel sequential choice task which embodied a Markov decision process¹⁸ (**Fig 1A**).

Additionally, we now plot data according to the predictions of all 10 relevant variables to provide further support that the probability of foraging success and the optimal policy best explain behavior. See below for **Fig 2** for the main variables of the fMRI sample and **Fig S3** for the other variables. See **Figs S2 & S4** for the same plots with data from the behavioral sample.

Fig 2 for the two main variables:

Fig 2. Choice data of the fMRI sample

- (A)** Model comparisons show that the probability of foraging success was the best single predictor of participants' behavior. Main plots depict fixed-effects analyses using log-group Bayes factors based on Bayesian Information Criterion (BIC) relative to model #1. Insets show random-effects analyses using protected exceedance probabilities (EP) with the winning model marked. See **Table 1** for a list that specifies the task variables and thus the models tested here.
- (B)** Crucially, the *a priori* optimal policy according to a time horizon of five days best explained the remaining variance in participants' choices.
- (C)** Posterior predictive checks show that the winning model, which includes the probability of foraging success and the optimal policy, captures the empirical relationship between participants' average choices and the probability of foraging success. Marker sizes which scale with the average number of trials contributing to the respective data points.
- (D)** Posterior predictive checks show that the winning model captures the relationship between participants' average choices and the optimal policy according to a horizon of five days (binned value differences of foraging versus waiting).

Data are binned. Error bars are standard errors of the mean (SEM). In several cases, error bars are smaller than the marker sizes. See **Fig S2** for the same plots with data of the behavioral sample. See **Fig S3 & S4** for posterior predictive checks of the winning model with choice data split according to the 8 other heuristics and combinations thereof, and for parameter estimates of a full model including all candidate variables. See **Fig S5** for further posterior predictive checks of the winning model with choices split jointly according to the energy state and the probability of foraging success or the weather type. See **Fig S6** for comparisons of different time horizons. See **Tables S1, S2, & S3** for further model comparisons.

Fig S3 (and Fig S4) show all other 8 variables:

Fig S3. fMRI sample: Posterior predictive checks according to all heuristics and parameter estimates for full models including all candidate variables

(A) Posterior predictive checks show that the winning model, which includes the probability of foraging success and the optimal policy, captures the empirical relationship between all other 8 other heuristic variables. “Weather type” and “wins-stay-loose-shift” are binary variables. See **Table 1** for a list of all variables.

[...]

See **Fig S4** for the same plots for the behavioral sample.

See **Fig S5** for more plots, which show data split according to two heuristics.

A fMRI sample: energy state & p foraging success

B Behavioral sample: energy state & p foraging success

C fMRI sample: weather type & energy state

D Behavioral sample: weather type & energy state

Fig S5. Both samples: Further posterior predictive checks according to two combined variables for models of choice and RT data.

(A & B) Split according to energy state and probability of foraging success.

(C & D) Split according to weather type and energy state.

See legend of **Fig S3** for more information on the logic of these plots. See **Fig 2**.

Further, there are a number of promising heuristic candidates that weren't considered in the first wave (but are only measured, if at all, with p already in the model). These include the horizon-1 optimum, or the EV, probability times gain, both of which are related to the probability but also form more of a continuum with the optimal policy. Low horizon myopic optimal policies seem particularly nice options, but may not have fit well since they were only tested with p (the ultimate myopic variable) already in the model. A possible hypothesis is a variable depth optimization story, ranging between (or combining) horizon-1 and deeper solutions; this is also similar to the recent Keramati and Dayan study.

Thank you for these insightful suggestions. We now added the horizon-1 optimum and the EV in to the model space and also tested them separately (without the probability of foraging gain in the model). The horizon-1 optimum performed decisively worse in both samples (see **Fig 2** above).

Please note that in accordance with your later comment we corrected the calculation of the optimal policies (see below). Please also note that in response to one of your later comments and to a comment by reviewer 2, we now use continuous versions of the correctly calculated optimal policies according to different time horizons (i.e., the value difference between the two choice options).

This also addresses in part the “variable depth optimization story:” The time horizon in our task is fixed and finite. That is, participants have to first consider a time horizon of maximum 5 time steps and should reduce their horizon with each time step passed. This is adjustment of the time horizons is exactly what the corrected optimal policy captures mathematically and what explains participants' behavior.

We explicitly compared different time horizons and found that a horizon of 5 steps explained behavior best. Results section (pages 12-13 and **Fig S6**).

Participants' choices took five future states into account

Using the optimal policy implies computing future states. We performed detailed analyses to determine how many days participants looked ahead in the task. Participants were incentivized to consider a time horizon of five days: although in the main task they would often not finish five days (to enhance fMRI design efficiency), they were instructed that their pay off depended only on a subset of ten forests from the main task which they would then have to complete outside the scanner. When comparing models with horizons between one and seven time steps, we found that that participants' choices were indeed best described by a time-horizon of five days (**Fig S6**).

Fig S6. Both samples: Model comparisons for different time horizons of the optimal policy

Follow-up Bayesian model comparison suggested that participants used the time horizon of five days that was normative in our task. Main plots depict fixed-effects analyses using log-group Bayes factors (based on Bayesian Information Criterion, BIC) relative to model #1. Insets show random-effects analyses using protected exceedance probabilities with the winning model marked.

We now discuss the findings in our sequential decision-making task in relation to the results of Keramati et al. in a learning task with three steps. We think the intriguing study by Keramati et al. highlights at least two important points with respect to our findings: First, the considered time horizon of the optimal policy is likely to be flexibly adapted under different task constraints. Second, the balance between optimal and heuristic solutions can likely be shifted. Page 22:

Notably, participants did not simply resort to looking just one time step ahead, which would have been clearly suboptimal given that their finite time horizon was five steps in our task. A recent study by Keramati et al.²⁶ demonstrated that humans adaptively adjust the depth of planning and the reliance on habits to in a three-step learning task—with time pressure leading to shallower planning. We hold it likely that imposing different time constraints or varying the number of time steps in our task will alter the time horizon considered and shift the balance between heuristic and optimal computations.

The EV (probability of foraging gain * gain magnitude) had worse explanatory power than the probability of foraging success. The two variables were not very much correlated with each other across the trials used in our study (shared variance below 0.5; see **Fig S1**). Conversely, EV was quite highly correlated with the possible gain magnitude (shared variance of 0.84). This somewhat limits the possibility to separate the influences of EV and gain magnitude. We argue that this is less critical given that the probability of foraging gain explained behavior better than these other two variables.

- **This is a very difficult problem and I don't have great advice for the authors how to overcome it. It is probably too much to expect that it be overcome completely, and to some extent it can also be finessed by more judicious, cautious, interpretation and more caveats. But I think one step that absolutely needs to be taken is to more concretely demonstrate how the choice behavior would change as a function of some of these different candidates, and how (and whether) they can be distinguished in choice space. The plots of optimal vs. actual policies in Supp fig 1 are a start, but it doesn't make clear how the behavior can distinguish these from the more myopic heuristics, or all the horizons from each other. How many of the trials actually distinguish these?**

We thank the reviewer for encouraging us to provide a better evaluation of all candidate models. We now provide plots for behavior binned according to all candidate variables to allow visual evaluation. The size of the data points in these plots depicts the number of trials within the different bins, which provides an informal indication of statistical power across the different conditions and may guide follow-up studies. Importantly, these plots also contain posterior predictive checks for the winning model (see **Figs 2 & S3** above for the fMRI sample and **Figs S2 & S4** for the behavioral sample).

To be more judicious and cautious in our interpretation, we mention the caveats raised by all reviewers in the third paragraph of the discussion section. Page 21:

We did not find evidence that any linear combination of two candidate policies and variables explained our choice data better than the probability of foraging success and the optimal policy. Although it is theoretically possible that participants use a yet unknown decision policy, such policies do not follow from the given task variables in an obvious way. Any such model would thus likely require higher complexity than the linear combination of the probability of foraging success and the optimal policy. Our analyses did not provide decisive evidence for a more complicated model. We deem it an interesting question for follow-up research whether a different selection of trials or variations of our task design would result in more (or less) complex models being identified. For example, it could well be that more challenging tasks would lead participants to abandon the optimal policy in favor of a combination of two heuristics (such as a combination of the momentary probabilities of foraging success and the current energy state).

- **I of course understand the points about reaction times and fMRI correlates of the different decision variables, which provide some (quite tantalizing) support for the authors' interpretation. But again in the context of a large family of highly correlated candidates, and**

without more exhaustively comparing the possibilities and more carefully demonstrating their separability in data space I think it's hard to make too much of different neural correlates and RT effects.

As mentioned above, the correlations between the different candidates are within reasonable ranges that are common for (neuroimaging) studies on decision-making and learning (e.g., Kolling et al., 2014, Neuron). We now provide more exhaustive comparisons of both choice and RT data in terms of statistical analyses and visual depictions (see above). To be rigorous, we only include variables in the neural analyses that are robustly related to choice or RT data.

I would note that the RT issue cuts both ways in the paper – on the one hand we are told that only parallel computation can explain the RT choice difficulty effects (I think this is an awfully strong conclusion to draw so categorically, though it is certainly also a neat piece of evidence and a novel approach) – but on the other hand we are told that heuristic use is faster, which should not be true in a fully parallel model. I think it's quite likely that some progressive computation model like Keramati and Dayan's, where approximations are refined if some features of the problem suggest that deeper search is worthwhile, might in principle be able to explain both results.

The reviewer's suggestion to look more closely into RTs has helped us to elaborate on this point. After correcting the computation of optimal policy, we find that RTs are actually quite monotonically related to conflict (way more so than our previous analyses suggested). Plotting RTs depending on the heuristic (probability of foraging gain) reinforces this idea (see **Fig 3** below for the fMRI sample and **Fig S8** for the behavioral sample): RTs are highest when the probability of foraging gain is 0.5, i.e., when uncertainty about the outcome is highest. RTs decrease as the probability of foraging gain approaches 0.1 or 0.9. This may suggest that the heuristic itself reflects the “feature(...) of the problem suggest[ing] that deeper search is worthwhile.” Low (or high) probabilities of foraging gain seem to entail a deeper search strategy, which relates to the progressive computation in the line of the results by Keramati and Dayan.

We now are considerably more careful in our conclusions and use the more general term “integrated computation” instead of parallel computation throughout the manuscript. Please note that following a suggestion by reviewer 3 we now use the terms “choice uncertainties” and “discrepancy” instead of “difficulties” and “conflict.”

Results section (pages 13-16):

Reaction times increased with choice uncertainties of the heuristic and optimal policies

Models of choice data indicate that participants used both a heuristic policy, i.e., the probability of foraging success, and the optimal policy. Consequently, we predicted that reaction times (RTs) should reflect the choice uncertainties associated with these two variables. Indeed, RTs were slower when choice uncertainties were high (see **Fig 3A-D** for the fMRI sample and **Fig S8A-D** for the behavioral sample, see **Table S4** for statistics). For example, RTs were highest when the probability of foraging success was 0.5 and thus neither foraging nor waiting was clearly favored by this metric. Choice uncertainties were quantified on the basis of the mean parameter estimates of the choice models from the behavioral sample. For the fMRI sample, choice uncertainty calculations thus rely on independent data.

Choices based on the heuristic were faster than those based on the optimal policy

Given that participants' choices seem to integrate heuristic and optimal policy computations, the question arises how these computations relate to each other. A first straightforward prediction is that choices following the heuristic policy should be faster than those following the optimal policy, if the heuristic is easier to compute than the optimal policy. In many trials, both variables made the same prescriptions and thus these trials cannot be used to disambiguate choices made according to one or the other variable. Therefore, we identified

the subset of trials in which the two made opposite prescriptions (i.e., trials with opposite choice probabilities according to the mean parameter estimates of the choice model from the independent behavioral sample). Within this subset, mean RTs for trials in which participants' choices followed the probability of foraging success were faster than mean RTs for trials in which choices followed the optimal policy (mean difference \pm SD: fMRI sample: 90.7 ± 109.2 ms; $t(27)=4.40$; $p<0.001$; behavioral sample: 147.5 ± 96.2 ms; $t(20)=7.02$; $p<0.001$; choice probabilities did not differ between these trials; both $p's>0.4$).

In addition, a linear mixed effects model of RT data provided evidence for a relatively more pronounced influence of the choice uncertainty of the heuristic compared to the choice uncertainty of the optimal policy (**Table S4**). That is, choice uncertainty under the heuristic policy related more strongly to RTs than choice uncertainty under the optimal policy.

Reaction times increased with discrepancies between the heuristic and optimal policies

Integrated computation of heuristic and optimal policy makes a second crucial prediction for RT data: Decisions should take longer when the two variables make discrepant prescriptions. For example, RTs should be slower when the heuristic prescribes waiting but the optimal policy prescribes foraging or vice versa. We quantified these discrepancies between the two variables as the absolute differences in choice probabilities (which were based on the mean parameter estimates of the choice model from the independent behavioral sample). Indeed, decisions were slower when discrepancies between the variables were larger. This effect was present in addition to influences of choice uncertainties (see **Fig 3E** for the fMRI sample and **Fig S8E** for the behavioral sample).

Overall, log-transformed RT data were well described by a linear mixed effects model that included the heuristic and the optimal policies themselves, their associated choice uncertainties, and the discrepancies in the choice probabilities of the two policies (**Figs 3 & S8**; see **Figs S3B & S4B** for the parameter estimates of the full RT model fitted on the basis of individual participants and for **Fig S5** for further posterior predictive checks; see **Table S4** for results obtained from a linear mixed effects model of log-transformed RTs, which were qualitatively the same as results on untransformed RTs). Motivated by this model of RT data, we also tested whether interactions of choice uncertainties or discrepancy influenced choices but found no decisive effects (**Table S3**).

Fig 3. Reaction time data of the fMRI sample

We tested the relationship between RTs and variables associated with the heuristic and optimal policies. Since the probability of foraging success emerged as the best predictor of participants' choices, we only included this but not any other heuristic in the model of RT data. RTs relate to (A) the probability of foraging success and also weakly to (B) the optimal policy. Importantly, RTs become slower with (C & D) increasing choice uncertainties of these two variables and (E) higher discrepancies in their prescriptions. Posterior predictive checks show that RT data were well captured by a model that includes the five depicted variables.

Data are binned. Error bars are SEM. In several cases, error bars are smaller than the size of the markers, which scale with the average number of trials contributing to the respective data points. See Fig S8 for the behavioral sample and Figs S3 & S4 for the parameter estimates of the full RT model and Fig S5 for posterior predictive checks of the RT model with data split differently. See Table S4 for statistical inferences obtained from a linear mixed effects model.

We now also highlight this possibility in the discussion: Page 22:

Our behavioral data suggest interdependent but partly distinct computations of heuristic and optimal processes during sequential choice. But what is the relationship between the two policies? Analyses of reaction times (RTs) showed that the choice uncertainty of both policies—but in particular of the heuristic—slowed RTs. Intriguingly, the discrepancies in choice probability between the two policies also led to longer RTs, which provides crucial evidence for an integrated computation of heuristic and optimal policies. We interpret these findings as pointing toward progressive computational processes such that the approximations provided by the heuristic are abandoned if the associated choice uncertainties turn out to be too high. That is, an insufficient choice certainty of the heuristic metric suggests that it is worthwhile to engage in a deeper search in form of a full-blown optimal policy computation.

- 2. My technical issue is that as far as I can tell, the optimal policies used here are not computed correctly. More specifically, there are really two versions of the task used here – one with stochastic termination (i.e. exponentially distributed time in the forest), and the other a fixed horizon-5 game. Neither of these corresponds to the “optimal” policy computed here, which isn’t really optimal for any problem as far as I can tell. For the fixed, finite horizon-5 game, the key issue is that the optimal action choice will in general depend on the number of steps remaining, and so it is incorrect to use a fixed probability p of foraging (and take the transition matrices to the fifth power). For instance, when there is only one move left, if there is still sufficient energy left, then either forage or wait will be equally successful in terms of the objective function. But this might not be true at earlier steps in the forest with the same energy and weather. To find the true optimal policy, one would normally use dynamic programming, working backward from the last move and treating the different horizon steps as separate states with separate optimal policies. (That is, it is still a Markov problem, but only in a state space that distinguishes the steps toward the horizon.) Conversely, if the game terminated stochastically (exponentially distributed game length) with some probability which I will call $(1-\gamma)$ then the states are all equivalent independent of depth – the optimal policy is indeed fixed over periods within the forest, as a function of energy remaining and weather. However, in this case it is also not correct to optimize with respect to the fifth power of the transition matrix – one must account for the distribution of termination times. This is equivalent to an exponentially discounted, (in principle) infinite horizon problem, where the event of the task terminating with survival is, effectively, a second absorbing state (with 0 reward), which equivalently introduces a discount factor of γ over the infinite horizon version of the problem. This can then again be computed with value iteration / dynamic programming. This is discussed in standard texts, e.g. Sutton & Barto, or Puterman’s MDP book. (Note that if the exponential distribution is truncated, e.g. at maximum 5 steps, then this would violate the Markov property and a finite horizon model like the one above would actually apply: the policy would again depend on the step. The paper isn’t really clear what was done here.) I doubt these details will make too much difference, but I do think this is an error that pervades essentially all the results, which needs to be corrected and the results need to be revisited.**

We are very grateful to the reviewer for making this pertinent observation and for directly offering the solution to this issue. Indeed, our previous way of computing the optimal policy (by taking the fifth power of the transition matrix) was incorrect. We now calculated the optimal policy according to a finite time horizon as described by the reviewer and revised all behavioral and fMRI analyses accordingly. Importantly, our overall findings held and often became clearer when using the correctly calculated policy. As expected by the reviewer, the correctly calculated optimal policy only differed from the previous calculation in a small subset of trials (average of 21% of the trials). These are mostly trials in which both choice options have the same optimal value according (i.e., the optimal policy does not distinguish between the two; see also next point).

Furthermore, our previous manuscript was obviously not clearly enough stating that there is just one version of the task: Although in the main part of the task participants were only presented

with an exponentially distributed number of time steps, they were instructed (and understood) that they would have to complete the maximum of “five days within a forest” afterwards (outside the scanner) for a randomly chosen subset of forests that would count toward their payoff. They therefore had to employ a maximum time horizon of five steps (for the “first days within a forest”). Since monetary reward only depended on “survival” in the final fifth time step, played outside the scanner, no discount factor was implemented in the finite horizon model and therefore the optimal policy depends on the number of steps remaining as the reviewer noted. This is now also reflected in the fact that the optimal policy according to a horizon of five time steps best accounts for participants’ behavior in the main task.

The revised explanation of the optimal policy in the introduction: Pages 4-5:

We computed the *a priori* optimal policy that minimizes starvation probability according to a finite time horizon of five days for each combination of energy state, weather type, and day within a forest. Because the two choice options, foraging or waiting, vary in their relative expected values, the optimal policy allows for some degree of stochasticity. In line with the general notion that human choices are sensitive to value differences between choice options, we assumed choice would be determined by the continuous value differences between foraging and waiting, computed according to the optimal policy. In the following, we use the term “optimal policy” to refer to these continuous value differences between the choice options and not to the deterministically better option. Unless otherwise specified, we refer to the optimal policy according to the normative finite time horizon of five steps.

Fig1 was changed accordingly and now also includes a state transition diagram.

A Schematic task design

B Example transition matrices

		Bad weather					
		0	1	0	0	0	0
initial energy state	Good weather	0	1	0	0	0	0
	Bad weather	1	f_1q+w_1	0	f_1p	0	0
0	0	1	0	0	0	0	0
1	f_1q+w_1	0	f_1p	0	0	0	f_1p
2	f_2q	w_2	0	f_2p	0	0	f_2p
3	0	f_3q	w_3	0	f_3p	0	0
4	0	0	f_4q	w_4	0	f_4p	5
5	0	0	0	f_5q	w_5	f_5p	0
starvation		0	1	2	3	4	5

Example transition probabilities (starting in energy state 4)

- f_4q foraging (f) & no success (q): minus 2 points
- w_4 waiting (w): minus 1 point
- 0 probability of staying in energy state 4 is zero
- f_4p foraging (f) & success (p): plus 1 point

C Example state transition diagram (for one weather type)

Fig 1. Virtual foraging task testing for sequential decision-making.

(A) [...]

(B) Example transition matrices used for determining the value differences between the two choice options according to the optimal policy (corresponding to the forest in A). The entries in the matrices are the probabilities to transition from the initial energy state (rows) to the final energy state (columns) in one day. These entries (and their positions within the matrix) depend on the weather types of the forests and on the choice patterns. Many of the entries are 0, which means that transitions between the respective initial and final states are impossible (e.g., from one day to the next one cannot stay in state 4 and one cannot starve in states 3 and above). The forest is specified by the probability of foraging success p (with $q=1-p$) and the magnitude of gains and losses. These magnitudes are reflected in the positions of the probabilities within the matrix (e.g., gaining an additional point after successfully foraging in state 4 is indicated by the entry of the probability f_4p at the position initial state 4 and final state 5). Choices are reflected by the probabilities for foraging f and waiting w (with $f=1-w$). The probabilities f and w depend on the initial state and are indexed accordingly. Additionally, the optimal policy depends on the number of remaining days in the forest since there is a finite horizon with a maximum number of five days. Starvation is absorbing, which is why the probability of

staying in state zero is 1. There are two corresponding transition matrices for the two weather types. Backward computation is used to determine the value differences between the two choice options according to the optimal policies. That is, the values of the two choice options are first evaluated according to the last day in the forest, then according to the second-last day, etc.

- (C) The state transition diagram corresponding to the example transition matrix in **B**. Large empty circles depict (energy) states and small filled circles depict the two actions to choose from. Arrows indicate transitions between states. For clarity, only one weather type is shown. In total, each forest type comprises 12 states = 6 (energy states) x 2 (weather types).

The revised methods section in the **SI** reads:

Sequential decision-making in a virtual foraging frame

Mathematical framework in the form of a Markov decision process

To probe sequential decision-making, we propose a toy scenario and a corresponding Markov decision process (MDP) for a hunter-gatherer or any foraging agent that aims at dynamically maintaining homeostasis over time. See **Fig 1** for an example trial, the corresponding transition matrix, and the associated state transition diagram. See **Table 1** for a list of all variables described in this and the following sections. The decision-making agent has to keep its internal energy state s above zero, i.e., the agent “dies from starvation” upon reaching the energy state zero at any time step. Here, the energy state can have discrete values equaling 1-5 energy points (but our model easily extends to additional numbers of energy states without loss of generality). At each time step t , or “day,” the agent can choose to “wait” and incur a sure loss c_w (of one energy point) or it can “forage” in which case the agent probabilistically gains an amount g (of zero to four energy points) or incurs a cost for foraging c_f (of two energy points). We denote the probability of foraging success as p (i.e., the probability of gaining points during foraging). The probability of unsuccessful foraging and thus of incurring c_f is $q=1-p$. The maximum energy is capped; in the highest energy state the agent cannot gain more but simply stays in the highest state if foraging is successful.

The agent “lives” within a given “forest” in which all relevant variables are specified (presented to participants during the “forest phase,” see **Fig 1A**). We included good and bad environmental conditions denoted as “good and bad weather types” that each occur with a probability of 0.5 on a given time step. That is, there are total of 12 states in the MDP: 6 (energy states) \times 2 (weather types).

A core component of an MDP is the transition matrix between these different states (see **Fig 1B** for one example). Starvation, i.e., the state of having zero energy is absorbing (and thus the entry in the transition matrix leading from energy state zero to energy state zero is 1). Waiting leads to a sure loss c_w of 1 point, which is formalized by the associated probabilities w in the off-diagonal below the main diagonal. Foraging can lead to either success (fp) or not (fq). These probabilities constitute the entries in the associated off-diagonals (depending on the magnitudes of g and c_f).

How should the agent’s optimal policy look like? The agent should minimize the probability of starvation $p_{starve}(n,s,t)$; i.e., it should minimize the probability of reaching zero energy points within a fixed and finite time horizon of n days when starting with the internal energy state s . In our case, the finite time horizon n is always 5 days and the starting energy state s at the first day can be 2, 3, or 4 energy points. Participants were incentivized accordingly, i.e., they received a monetary payoff (for a random subset of trials), if their energy was above zero at day 5 and nothing otherwise. This corresponds to a simple implementation of a reward function within the framework of MDPs: All transitions to zero are associated with a “reward” of -1 and all other transitions with a reward of 0.

Starvation probability $p_{starve}(n,s,t)$ and thus the optimal policy depend on the agent’s choice at the current time step t and the next $n-1$ time steps—in addition to the dependence on n and s . In our finite-horizon scenario, the starvation probability thus depends on the number of time steps t , i.e., days within a forest.

A rough and basic intuition for a more or less prototypical optimal policy under such circumstances is that in many types of forests the agent should forage when the weather is good and wait when the weather is bad; unless waiting leads to sure death on the next time step. When the energy state is high enough so that starvation is impossible within the remaining number of days (e.g., energy state is 4 and only 1 day is left of the 5 days within a forest), the agent is indifferent between the two choice options.

Derivation of the optimal policy within the Markov decision process

We derived the optimal policies analytically for a large number of different forests, i.e., for all combinations of p (from 0.1 to 0.9 in steps of 0.1) and gains g (from 0 to 4 in steps of 1) for a finite time horizon of $n=5$ days. In our MDP, the possible policies consist of the probabilities

of foraging f in each energy state s , environmental condition (“weather”), and each time step t (with the probabilities of waiting $w=1-f$). To derive optimal policies in our finite-horizon scenario, we used backward induction: That is, we started from the final time step (i.e., day 5) and calculated the values of the two choice options (i.e., foraging or waiting) for each state for that last and final time step. These values depend on the possible transitions from the respective states (see the example transition matrix in **Fig 1B** and the corresponding example state transition diagram in **Fig 1C**). If the value of foraging is higher than the value of waiting, foraging is the deterministically better option in that state and at that time step—or vice versa. If both choice options have the same value, the optimal choice is indifferent between the two options. We then used the values of the better choice options to calculate the values for the second-to-last time step and determined optimal choices. This procedure was repeated until arriving at the first time step (i.e., day 1).

The optimal policy thus depends on the time horizon considered. Our task was designed such that participants should use a time horizon of 5 time steps and participants were instructed and incentivized accordingly. Nevertheless, it is a possibility that participants use a different time horizon. Therefore, we calculated optimal policies for the following time horizons: $n=7$, $n=6$, ..., $n=1$. Participants may employ a longer (i.e., 7 or 6 days) or shorter (i.e., 4, 3, 2, or 1 days) time horizon than instructed. Time horizons are flexible in the sense that it is assumed that participants start in a new forest with a given time horizon n and then reduce their horizon on each day by one. If the horizon has reached one (i.e., $n=1$), it will remain one.

Since the optimal policies binarize the value differences over the two choice options (either foraging or waiting is better, or they are exactly the same), they do not allow for variability in the decision process (i.e., in some cases waiting and foraging entail large value differences whereas in other cases the two choice options have quite similar values). We therefore used the continuous value differences between the two choice options as predictors of participants’ choices, RTs, and fMRI data. For brevity, we often use the term “optimal policy” to refer to the value differences between the foraging and waiting (according to a horizon of 5 steps).

All calculations were carried out in MATLAB.

Two side points arise from these considerations. First, while it is a theorem that all MDPs have a deterministic optimal policy (thus it is no accident that the optimal policies here are deterministic), that policy is not in general unique. Here, for instance, I strongly suspect there are states where foraging and waiting both ensure survival for sure (e.g. in the last move as discussed above) and here any policy, including stochastic ones, will be equivalently optimal. The analysis needs to take this into account.

Thank you again for this important remark. Following this comment and a comment by reviewer 2, we no longer use the binary optimal policies as decision variables in our behavioral and fMRI models but the differences in value between the two choice options according to the optimal policy (value of “forage” minus value of “wait”). This value difference has the great advantage of being a continuum with zero as the natural indifference point. The newly provided plots of empirical choices show that on average participants are indeed indifferent between the two options when they are equivalently good (see above for the revised introduction paragraph and **Fig 2**). Thus, our new analyses capture this aspect that the optimal policy is in some cases indifferent between the two choice options.

Second, the discussion of value iteration in the supplement seems a bit off. As I hope I have described above, when properly framed, the original avoid-death problem is nothing other than an MDP which can be solved by dynamic programming, i.e. value (or policy) iteration over the appropriate state space and the appropriate finite or infinite horizon. Thus the discussion in the supplement appears not to really be about the solution method, but about the nature of the objective (ie, reward) function, i.e. maximizing some new intake function vs avoiding starvation. We agree with the reviewer that this discussion is besides the main point of the paper and does not provide additional support for the other results. We therefore removed this part of the supplementary information.

Reviewer #2 (Remarks to the Author):

Korn and Bach build on two emerging lines of research: one focused on planning across deep decision trees, and the other on using ethologically relevant paradigms to understand the neural mechanisms of decision making. The authors develop a task that probes foraging behavior in a simulated environment where optimal performance requires planning through a deep and probabilistic tree of possible future states. The authors collect fMRI data during task performance and use the behavioral and imaging data to support the following key claims:

- 1) Human foraging behavior deviates from optimal planning and is better explained by a heuristic (win probability) that is occasionally abandoned in favor of an optimal solution when the heuristic is difficult to implement.**
- 2) Both optimal and heuristic strategies are computed in the brain in parallel and arbitrated in medial prefrontal cortex, with the optimal policy scaling with BOLD activity in anterior MPFC, conflict between optimal and heuristic policies scaling with BOLD signal in dorsal MPFC, and the difficulty of the heuristic model reflected in the BOLD signal in anterior VMPFC and mid cingulate.**

Thank you for this summary of the approach and findings in our original manuscript.

My feelings about this paper are somewhat mixed; on one hand I feel that the authors have taken on a difficult question and made strides toward pinning down an answer. However, on the other hand, in part because of the difficulty of the question and the complexity of the task, I am unconvinced that the authors' interpretation of the data is the only, or even the best, one. In particular I have concerns regarding the interpretation of the behavior as emerging from two separate strategies implemented in parallel and reflected computed through different neural systems. This interpretation is at odds with the standard rationale for why heuristics are adopted, and thus, if true, would constitute a surprising and important finding. But based on the data presented by the authors, I think it is more likely that subjects are implementing a serial strategy in which they first pay attention to the foraging success probability, and then, if not sufficiently swayed by this information, move on to consider the current "energy state". This interpretation would be more in line with standard ideas about multi-attribute decision-making, as far as I can tell in line with the behavioral analyses, and would require major reinterpretation of the fMRI results. A complete listing of my concerns is below:

We now report considerably more detailed and refined analyses and also a more careful discussion of how heuristic and optimal policies are computed. Your comments were very convincing to us and we now refrain from drawing unduly strong assumptions about truly parallel computation.

Major Issues:

- 1) It is not clear that the primary behavioral claims are supported by the data. As far as I can tell, the authors do not compare a model of the decision variable from the optimal model (difference in survival across forage/wait) against the heuristic model. This is an important comparison. The binary optimal policy variable that the authors use for comparison does not include any information about the magnitude of model preferences, and thus has no way of accounting for noise in the decision process.**

Thank you very much for this pertinent comment. We agree that the binary variable does not account for relative magnitude of the two choice options and thus neglects a legitimate source of choice variability. We now use the proper decision variable (i.e., the value difference between the two choice options "forage" minus "wait") for all behavioral and fMRI analyses. We previously used this metric only for a specific behavioral model. Importantly, our main results held and actually became clearer. As the reviewer noted, this decision variable allows to assess choices on a continuum. Newly included plots show that of participants' behavior scales with this decision variable and corresponds to the winning model, which includes this continuous variable (see **Fig 2** below for the fMRI sample and **Fig S2** for the behavioral sample).

We specified this in **Table 1**. The revised explanation of the optimal policy in the introduction: Pages 4-5

We computed the *a priori* optimal policy that minimizes starvation probability according to a finite time horizon of five days for each combination of energy state, weather type, and day within a forest. Because the two choice options, foraging or waiting, vary in their relative expected values, the optimal policy allows for some degree of stochasticity. In line with the general notion that human choices are sensitive to value differences between choice options, we assumed choice would be determined by the continuous value differences between foraging and waiting, computed according to the optimal policy. In the following, we use the term “optimal policy” to refer to these continuous value differences between the choice options and not to the deterministically better option. Unless otherwise specified, we refer to the optimal policy according to the normative finite time horizon of five steps.

The revised **Fig 2** now shows the relationship between behavior and the continuous optimal policy variable (see **2D**) along with posterior predictive checks.

Fig 2. Choice data of the fMRI sample

- (A)** Model comparisons show that the probability of foraging success was the best single predictor of participants' behavior. Main plots depict fixed-effects analyses using log-group Bayes factors based on Bayesian Information Criterion (BIC) relative to model #1. Insets show random-effects analyses using protected exceedance probabilities (EP) with the winning model marked. See **Table 1** for a list that specifies the task variables and thus the models tested here.
- (B)** Crucially, the *a priori* optimal policy according to a time horizon of five days best explained the remaining variance in participants' choices.
- (C)** Posterior predictive checks show that the winning model, which includes the probability of foraging success and the optimal policy, captures the empirical relationship between participants' average choices and the probability of foraging success. Marker sizes which scale with the average number of trials contributing to the respective data points.
- (D)** Posterior predictive checks show that the winning model captures the relationship between participants' average choices and the optimal policy according to a horizon of five days (binned value differences of foraging versus waiting).

Data are binned. Error bars are standard errors of the mean (SEM). In several cases, error bars are smaller than the marker sizes. See **Fig S2** for the same plots with data of the behavioral sample. See **Fig S3 & S4** for posterior predictive checks of the winning model with choice data split according to the 8 other heuristics and combinations thereof, and for parameter estimates of a full model including all candidate variables. See **Fig S5** for further posterior predictive checks of the winning model with choices split jointly according to the energy state and the probability of foraging success or the weather type. See **Fig S6** for comparisons of different time horizons. See **Tables S1, S2, & S3** for further model comparisons.

It is very clear from the behavioral figures (Fig 2 & supplementary figure 1) that subjects change behavior according to the energy state suggesting that the suggested heuristic (foraging success probability) could not provide a good fit to the data.

Thank you for this pertinent observation. Our new analyses and further considerations offer two points with respect to this comment. First, the heuristic alone can only give a first approximation and does clearly not account fully for of participants' behavior. This is precisely the point why a model that combines the heuristic and the optimal policy provides a superior fit. The calculation of the optimal policy accounts for the energy state—and further additional variables (in non-linear ways; a new **Fig S1** depicts these relationships). Notably, energy state alone—or any combination of energy state with another variable—does not provide a better explanation of the data (as shown by testing all possible pairs of the 10 policies considered, resulting in 45 models). No pair of two policies is better than the model that includes the optimal policy and the winning heuristic (i.e., probability of foraging success; see **Table S1** below for the fMRI sample and **Table S2** for the behavioral sample).

Second, we also tested further models. Specifically, we tested models that included an interaction between the probability of foraging success and other heuristics, including the energy state (**Table S3**). These models were outperformed by the model that included the probability of foraging success and the optimal policy. Results section: Page 12:

Additionally, we made sure that models including interactions between the most important heuristics did not provide a better fit than the model with the optimal policy (**Table S3**).

We also report analyses testing whether a linear combination of three variables provides a better explanation of participants' choices than the combination of one heuristic (p foraging success) and the optimal policy. A comparison among models including these two policies and any of the remaining eight candidate models did not provide decisive evidence for a winning model among these eight models (especially when considering protected exceedance probabilities and both samples, **Table S3**). Also, when testing the “three-variables-model” that includes the energy state (p foraging success, optimal policy, and energy state), we did not find decisive evidence for this model being better than the winning “two-variable-model” (p foraging success and optimal policy; **Table S3**). We mention acknowledge that a different selection of trials might lead to the “energy state” being a clear-cut additional predictor. Results section: Page 12:

In the two samples tested here, there was no decisive evidence that a model with three variables outperformed the model with the probability of foraging success and the optimal policy (**Table S3**). We deem it possible that specific selections of different trial types could identify the use of more complicated models, such as models including a variable for energy states.

To be more judicious and cautious in our interpretation, we mention the caveats raised by all reviewers in the third paragraph of the discussion section. Page 21:

We did not find evidence that any linear combination of two candidate policies and variables explained our choice data better than the probability of foraging success and the optimal policy. Although it is theoretically possible that participants use a yet unknown decision policy, such policies do not follow from the given task variables in an obvious way. Any such model would thus likely require higher complexity than the linear combination of the probability of foraging success and the optimal policy. Our analyses did not provide decisive evidence for a more complicated model. We deem it an interesting question for follow-up research whether a different selection of trials or variations of our task design would result in more (or less) complex models being identified. For example, it could well be that more challenging tasks would lead participants to abandon the optimal policy in favor of a combination of two heuristics (such as a combination of the momentary probabilities of foraging success and the current energy state).

Please see also below (comment 2) for more details on choice behavior at the energy boundaries.

Fig S1. Relation of optimal policy and heuristic variables

Shared variance between all 10 candidate variables and relationships with the optimal policy (according to the normative horizon of five time steps) with all 9 heuristic variables considered. Data are binned and arranged in the same way as below in **Figs S3 & S4**, which show the empirical relationship between data and fitted model. “Weather type” and “wins-stay-loose-shift” are binary variables. See **Table 1** for a list of all variables.

In general, the authors have tested a large number of models, but done very little to show the extent to which these models capture the subject behavior. Posterior predictive checks to show that the authors preferred models can capture the behaviors shown in Fig 2 and supplementary figure 1 would be helpful in this regard. The behavioral figures clearly indicate that behavior changes as a function of energy state, meaning that the heuristic strategy favored by the authors does not capture a key behavioral feature of the data.

We thank the reviewer for prompting us to provide further plots and posterior predictive checks. We now plot behavior (and RTs) binned according to all considered decision variables along with the posterior predictive checks of the winning model. Overall, model predictions appear quite close to actual data across all conditions.

For the two main variables see **Fig 2** pasted above to an earlier comment.

For splits according to all other 8 variables see **Fig S3** below for the fMRI sample (and **Fig S4** for the behavioral sample):

Fig S3. fMRI sample: Posterior predictive checks according to all heuristics and parameter estimates for full models including all candidate variables

(B) Posterior predictive checks show that the winning model, which includes the probability of foraging success and the optimal policy, captures the empirical relationship between all other 8 other heuristic variables. “Weather type” and “wins-stay-loose-shift” are binary variables. See **Table 1** for a list of all variables.

[...]

See **Fig S4** for the same plots for the behavioral sample.

See **Fig S5** for more plots, which show data split according to two heuristics.

A fMRI sample: energy state & p foraging success

B Behavioral sample: energy state & p foraging success

C fMRI sample: weather type & energy state

D Behavioral sample: weather type & energy state

Fig S5. Both samples: Further posterior predictive checks according to two combined variables for models of choice and RT data.

(A & B) Split according to energy state and probability of foraging success.

(C & D) Split according to weather type and energy state.

See legend of **Fig S3** for more information on the logic of these plots. See **Fig 2**.

We now also provide posterior predictive checks for the RT model in **Fig 3**.

Fig 3. Reaction time data of the fMRI sample

We tested the relationship between RTs and variables associated with the heuristic and optimal policies. Since the probability of foraging success emerged as the best predictor of participants' choices, we only included this but not any other heuristic in the model of RT data. RTs relate to **(A)** the probability of foraging success and also weakly to **(B)** the optimal policy. Importantly, RTs become slower with **(C & D)** increasing choice uncertainties of these two variables and **(E)** higher discrepancies in their prescriptions. Posterior predictive checks show that RT data were well captured by a model that includes the five depicted variables.

Data are binned. Error bars are SEM. In several cases, error bars are smaller than the size of the markers, which scale with the average number of trials contributing to the respective data points. See **Fig S8** for the behavioral sample and **Figs S3 & S4** for the parameter estimates of the full RT model and **Fig S5** for posterior predictive checks of the RT model with data split differently. See **Table S4** for statistical inferences obtained from a linear mixed effects model.

- 2) I am not fully convinced by the authors' claim that the optimal and heuristic computations are done in parallel. I typically think of a heuristic as offering computational savings afforded by avoiding costly computations associated with more complex solutions. Here the authors are saying that people spend the computational resources to evaluate both strategies... but then choose a worse one much of the time. The authors' state on lines 212-213:

“To address this issue, we calculated optimal and heuristic policies and analyzed whether a discrepancy between the two influenced participants' behavior. This would only occur if both were computed in parallel, on the same trials.”

We thank the reviewer for highlighting this crucial point. We now offer a more careful interpretation in terms of “integrated” processing. We previously intended to imply that participants compute both policies in some trials but we did not intend to convey that this computation was done at the same moment in time within a trial, or in all trials. We have changed the sentence and the relevant sections: Results (pages 13-16):

Reaction times increased with choice uncertainties of the heuristic and optimal policies

Models of choice data indicate that participants used both a heuristic policy, i.e., the probability of foraging success, and the optimal policy. Consequently, we predicted that reaction times (RTs) should reflect the choice uncertainties associated with these two variables. Indeed, RTs were slower when choice uncertainties were high (see **Fig 3A-D** for the fMRI sample and **Fig S8A-D** for the behavioral sample, see **Table S4** for statistics). For example, RTs were highest when the probability of foraging success was 0.5 and thus neither foraging nor waiting was clearly favored by this metric. Choice uncertainties were quantified on the basis of the mean parameter estimates of the choice models from the behavioral sample. For the fMRI sample, choice uncertainty calculations thus rely on independent data.

Choices based on the heuristic were faster than those based on the optimal policy

Given that participants' choices seem to integrate heuristic and optimal policy computations, the question arises how these computations relate to each other. A first straightforward prediction is that choices following the heuristic policy should be faster than those following the optimal policy, if the heuristic is easier to compute than the optimal policy. In many trials, both variables made the same prescriptions and thus these trials cannot be used to disambiguate choices made according to one or the other variable. Therefore, we identified the subset of trials in which the two made opposite prescriptions (i.e., trials with opposite choice probabilities according to the mean parameter estimates of the choice model from the independent behavioral sample). Within this subset, mean RTs for trials in which participants' choices followed the probability of foraging success were faster than mean RTs for trials in which choices followed the optimal policy (mean difference \pm SD: fMRI sample: 90.7 ± 109.2 ms; $t(27)=4.40$; $p<0.001$; behavioral sample: 147.5 ± 96.2 ms; $t(20)=7.02$; $p<0.001$; choice probabilities did not differ between these trials; both p 's >0.4).

In addition, a linear mixed effects model of RT data provided evidence for a relatively more pronounced influence of the choice uncertainty of the heuristic compared to the choice uncertainty of the optimal policy (**Table S4**). That is, choice uncertainty under the heuristic policy related more strongly to RTs than choice uncertainty under the optimal policy.

Reaction times increased with discrepancies between the heuristic and optimal policies

Integrated computation of heuristic and optimal policy makes a second crucial prediction for RT data: Decisions should take longer when the two variables make discrepant prescriptions. For example, RTs should be slower when the heuristic prescribes waiting but the optimal policy prescribes foraging or vice versa. We quantified these discrepancies between the two variables as the absolute differences in choice probabilities (which were based on the mean parameter estimates of the choice model from the independent behavioral sample). Indeed,

decisions were slower when discrepancies between the variables were larger. This effect was present in addition to influences of choice uncertainties (see **Fig 3E** for the fMRI sample and **Fig S8E** for the behavioral sample).

Overall, log-transformed RT data were well described by a linear mixed effects model that included the heuristic and the optimal policies themselves, their associated choice uncertainties, and the discrepancies in the choice probabilities of the two policies (**Figs 3 & S8**; see **Figs S3B & S4B** for the parameter estimates of the full RT model fitted on the basis of individual participants and for **Fig S5** for further posterior predictive checks; see **Table S4** for results obtained from a linear mixed effects model of log-transformed RTs, which were qualitatively the same as results on untransformed RTs). Motivated by this model of RT data, we also tested whether interactions of choice uncertainties or discrepancy influenced choices but found no decisive effects (**Table S3**).

The authors find that RT increases (somewhat non-monotonically) with the difference in choice-probabilities from the optimal and heuristic models.

The revised analyses with the corrected calculation of the continuous decision variable of the optimal policy show a considerably more monotonic increase of RTs than our previous analyses (see **Fig 3** pasted above).

But this could come about in many ways other than parallel computation. For example, if the models diverge for particular trials that are systematically slower. My guess is that this is the case. In particular, I think that participants are likely slowest when they are required to combine two pieces of information to make a successful decision (eg. energy and probability). This would occur for low probabilities when energy is low (1) or for high probabilities when energy is high (5). These are also conditions when model predictions would diverge most, since the probability assessment gets overridden in the optimal model by the proximity to the energy boundary. In order to address this, the authors could plot RT as a function of conditions, which, along with the posterior predictive checks described above, would test this alternative explanation for the RT relationship (if the relationship is explained by similarity across task conditions, it is consistent with my interpretation). Clarifying this point seems critical to interpreting data from the neural GLM, since the authors include both optimal and heuristic terms as modulators under the assumption that they are separately represented in the brain.

We thank the reviewer for this insight. Indeed from the previously provided plots it seems that these two sub-conditions (low energy of 1 point with probabilities of foraging gain below 0.5 and high energy of 5 points with probabilities of foraging gain above 0.5) constitute instances that differ from the other conditions and are relatively less well described by the overall winning models of behavior and RTs. The new plots of choice and RT data show a similar picture (see above **Fig S3 & S5**). Importantly, these plots also show that unfortunately there were a limited number of trials within these conditions. This was actually intended by our stimulus selection: We aimed at reducing the number of trials in which participants were starved (because participants could not provide answers in these trials). Conversely, in energy state 5 the optimal policy is often indifferent with respect to foraging or waiting. Therefore, we had participants start in the energy states 2, 3, and 4 but never in 1 or 5.

Nevertheless, as described above, we conducted additional analyses to address the reviewer's point. We did not find evidence that choice data were consistently and decisively better explained by the energy state (see major comment 1). To offer the reviewer a better evaluation of the influence of energy state, we provide posterior predictive checks for choice and RT models with and without the inclusion of energy state.

This comparison shows that even including the energy state does not convincingly account for the trials the reviewer refers to.

Overall, it thus seems quite likely that there are several separate—but not mutually exclusive—explanations for RT increases in our task: First, RTs increase with higher absolute differences (“discrepancies”) in choice probabilities from the optimal and heuristic policies as we demonstrated earlier and show more convincingly in the revised version (with the newly calculated continuous optimal policy and the help of posterior predictive checks). Second, RTs increase with the choice uncertainties of the optimal and heuristic policies. RT effects of these two choice uncertainties are motivated given that behavior relies on optimal and heuristic policies. The fact that they both explain variance could be taken as an indication of a separate computation of these variables.

Third, RTs may increase when an integration of the energy state boundaries is warranted as remarked by the reviewer. In line with the reviewer’s suggestion, we would interpret these as cases in which the bounded range of the energy bar creates non-linearity, which could best be accounted for by integrating two pieces of information—in particular an additional reliance on the energy state. We agree with the reviewer that such “integration” requires extra decision time. As detailed above, our selection of trials was not optimized to address this third effect. On average there are less than 5 trials per subject in the relevant bins. We hold it likely that our current task just does not provide the necessary power. On the other hand, this makes it quite unlikely that the fMRI analyses are affected by this “integration effect.” To avoid overburdening the manuscript, we refrain from presenting the right-hand part of the above plot. Instead, we suggest energy state as an interesting variable and discuss the possibility of an “integration effect” as an important avenue for follow-up studies.

For the reviewer’s convenience, we again past this section here. Page 21:

We did not find evidence that any linear combination of two candidate policies and variables explained our choice data better than the probability of foraging success and the optimal policy. Although it is theoretically possible that participants use a yet unknown decision policy, such policies do not follow from the given task variables in an obvious way. Any such

model would thus likely require higher complexity than the linear combination of the probability of foraging success and the optimal policy. Our analyses did not provide decisive evidence for a more complicated model. We deem it an interesting question for follow-up research whether a different selection of trials or variations of our task design would result in more (or less) complex models being identified. For example, it could well be that more challenging tasks would lead participants to abandon the optimal policy in favor of a combination of two heuristics (such as a combination of the momentary probabilities of foraging success and the current energy state).

We specifically discuss a possible RT integration effect. Pages 22-23:

RTs likely reflect an additional feature relevant in the decision process: In some cases, the requirement to integrate two specific types of information may be especially pertinent (for example at the energy state boundaries integrating information about the probability of foraging success and about the energy state can be crucial). Identifying the precise temporal requirements of information integration processes is an interesting and challenging avenue for future research.

3) As far as I can tell, the fMRI GLM does not include choice (forage/wait), but the other variables in the model are highly correlated with it. I can certainly understand why the authors would be reluctant to include choice along with the other terms, given that the input correlations in the model are already high (also, as a side point, it would be useful if VIF were listed instead of max pairwise shared variance). However, an interpretation of the current model may be that a large swath of the brain responds to the decision to forage... and that, through noise and thresholding, the authors have identified separate clusters for optimal policy and probability of foraging success regressors. To this effect, assuming that correlations with forage/wait decisions are high, I would recommend that the authors first do a whole brain analysis to identify regions responsive to choice (which is presumably informed by both measures), and test the encoding of each model based term within the ROIs responsive to the decision itself. It would also be useful if the authors could test whether representations of the terms are significantly different from one another through a normalized subtractive contrast.

We now present the results of two additional and separate GLMs, in order to allow the reader to evaluate suggestions by the reviewer.

One additional GLM included only choice as a parametric modulator of the choice phase (see **Fig S9 & Table S7**). Just using ROIs on the basis of a contrast between foraging and waiting choices would in our opinion result in an unduly restricted analyses of brain activity. On the other hand, the overall contrast of choice (i.e., the onset regressor) identifies (as expected) large unspecific areas that showed activity during this task phase. This overall contrast does therefore not lend itself as the basis for identifying ROIs.

Another additional GLM included participants' choice along with all the other variables as parametric modulators of the choice phase. This GLM revealed mostly the very same regions as the main GLM without choice as a parametric modulator (see text below and a comparison of **Table S5** for the main GLM and **Table S6** for the GLM with choice as an additional parametric modulator).

In the revised version of the manuscript, we also took care not to make unqualified statements suggesting that one region is only related to a given variable (but not to another). We mention these findings in the fMRI results section. Page 9:

Overall, the same regions described above were also identified in another GLM, which additionally included participants' choices themselves as parametric modulators during the choice phase (**Table S6**). The main qualitative difference between the GLMs with and without choices as additional parametric modulator was that in the GLM including choices the DMPFC cluster related to lower choice uncertainty of the optimal policy failed to reach significance (cf. **Fig 4B**). A GLM that only included participants' choices as parametric modulator did not reveal all the regions described above to be related to the heuristic and

optimal policies, their choice uncertainties as well as the discrepancies between the two policies (**Fig S9, Table S7**). This suggests that the variables identified from choice and RT models accounted for variance in the fMRI beyond the variance explained by choice *per se*.

A GLM with participants' choice only
(foraging < waiting)

B GLM with participants' choice only
(waiting > foraging)

Fig S9. Statistical parametric maps for the choice phase (GLM with participants' choices as only parametric modulator)

Overlay on group average T1-weighted image in MNI space; clusters are whole-brain FWE corrected for multiple comparisons at $p < 0.05$ with a cluster-defining threshold of $p < 0.001$. See **Table S7** for a list of all clusters.

- 4) **One important point made by the authors is that the conflict signal that they identify in ACC/DMPFC is conflict across different policies (optimal/heuristic) rather than a signal related to overall decision difficulty/uncertainty. However, the authors do not include the best possible decision difficulty regressor in the model, and given the recent heated debate on the issue, it seems important that the authors clearly show that they are explaining variance beyond simple choice difficulty. If the authors want to make claims about “controller conflict” above and beyond simpler measures of decision difficulty, they should estimate choice difficulty using the best-logistic regression model and include this term as regressor in their GLM. Given that the “probability of foraging under optimal policy” regressor also loads on a similar region, it would be good to know something about the overall rates of foraging (are foraging decisions or wait decisions more frequent?)**

We are grateful to reviewer for raising this important point. We now included the decision uncertainties associated with the two policies into the GLM and present these contrasts as main fMRI results in **Fig 5**. Please also note that we now refrain from using the term “difficulty” (since this creates misleading and incorrect connotations as pointed out by reviewer 3). Instead we use the term “choice uncertainty” as also done here by the reviewer.

Fig 5. Statistical parametric maps for the respective uncertainties of heuristic and optimal policies, the discrepancies in their choice probabilities, and log-transformed RTs during the choice phase (overlay on group average T1-weighted image in MNI space; clusters are whole-brain FWE corrected for multiple comparisons at $p < 0.05$ with a cluster-defining threshold of $p < 0.001$). See **Table S5** for a list of all clusters. See **Fig S9** for results from a GLM that only includes participants' choices as parametric modulators. See **Fig S10** for BOLD signals during the outcome phase.

- 5) **As in the behavioral analyses, the authors have portrayed the optimal policy as binary and the heuristic one to be continuous. I see no reason why this should be the case, and I wonder to what extent the differences in the variance explained by these two terms could simply be explained by this analysis decision. If the authors are looking for a representation of an optimal decision variable, it seems that they should include the difference in projected starvation for forage versus stay decisions in the neural GLM, rather than a binary variable that reflects the optimal choice.**

We are fully convinced by the reviewer's arguments to use the continuous decision variable for both behavioral and fMRI data. We have revised all behavioral and fMRI results accordingly (see **Fig 4** below). Importantly, the overall neural results do not change considerably (which reflects the correlation of the newly used continuous variable and the previously used binary variable).

Fig 4. Statistical parametric maps for the BOLD signals related to heuristic and the optimal policies during the choice phase (overlay on group average T1-weighted image in MNI space; clusters are whole-brain FWE corrected for multiple comparisons at $p < 0.05$ with a cluster-defining threshold of $p < 0.001$). See **Table S5** for a list of all clusters.

6) The authors should include a methods section in the main text that conveys the information necessary to interpret the results.

We have no considerably specified and extended the description of the task, the employed behavioral analyses, and the fMRI analyses throughout the main text. Due to space restrictions and since the supplementary methods are published in conjunction with the online version, we were not able to include a full-fledged methods section. We have rearranged the methods sections and now provide the necessary information in the main text. We are happy to include further parts of the methods into the main text if the editor sees fit.

Minor comments

The authors describe the process of arbitrating between the heuristic and optimal strategies as depending on the difficulty of implementing the heuristic strategy. However, it seems equally valid (and perhaps more intuitive) to say that the heuristic strategy is abandoned when it provides uncertain information. This interpretation is completely in line with other theories of how the brain combines information from segregated modules in other domains (eg. combining RL with working memory, combining model-based with model free information). The authors seem to interpret the model fits more in terms of their global meanings (optimal versus heuristic) and less in terms of what aspects of the data they actually capture. I think that the paper would be much easier to understand if they focused on the latter, rather than the former.

We now carefully revised the whole manuscript and in particular the discussion section to reflect this crucial suggestion. Page 22.

Our behavioral data suggest interdependent but partly distinct computations of heuristic and optimal processes during sequential choice. But what is the relationship between the two policies? Analyses of reaction times (RTs) showed that the choice uncertainty of both policies—but in particular of the heuristic—slowed RTs. Intriguingly, the discrepancies in choice probability between the two policies also led to longer RTs, which provides crucial evidence for an integrated computation of heuristic and optimal policies. We interpret these findings as pointing toward progressive computational processes such that the approximations provided by the heuristic are abandoned if the associated choice uncertainties turn out to be too high. That is, an insufficient choice certainty of the heuristic metric suggests that it is worthwhile to engage in a deeper search in form of a full-blown optimal policy computation.

Line 21: easy-to-?

We correct to “easy-to-compute”

Figure 2: This figure would be much improved if spent less real-estate on which models were fit and more on how well they fit. For example, showing the heuristic, optimal (using the correct DV, as described above), and heuristic and optimal models along with posterior predictive checks to show how they fit the actual choice curves (figure 2&supplementary figure 1) would provide more insight into the modeling than all of these bar graphs.

We cut the number of bar plots in half by putting all data from the behavioral pilot group into the supplementary. Instead, we now provide the suggested plots of binned data along with posterior predictive checks. See **Figs 2, S3, & S5** plotted above in response to earlier comments as well as **Figs S4 & Fig S8**.

Figure 4A, right: I have no idea what the x-axis is here. It is labeled “prescriptions of the optimal policy” but surely that is the y-axis. As far as I understand the implementation of “optimal policy” it was binary, and thus the lines seem somewhat superfluous. Though, as I said above, I think that a continuous decision variable would be more reasonable.

In line with the reviewer's first comments, we now use (and plot) the continuous decision variable (i.e., the differences in value for the two choice options according to the optimal policy, see **Fig 2** plotted above).

Line 288: It seems that the neural results would be easier to digest if there were some description of the GLM that was used to interrogate the imaging data at the beginning of this section.

We included the following brief description of the GLM in the manuscript. Page 17:

Neuroimaging results

We next assessed heuristic and optimal policy computations using our fMRI data. To do so, we implemented a general linear model (GLM) that included the variables from the winning choice model and the RT model as parametric modulators during the choice phase: the probabilities of foraging success, the value differences according to the optimal policy (time horizon of five days), their associated choice uncertainties, and the discrepancies in choice probabilities, as well as log-transformed RTs. Given the central role parts of the MPFC in decision-making, we specifically focused on this region.

Line 293: Maybe I missed something, but it is unclear to me why the probab

We have deleted the sentences in this line because the described analyses have been revised version after the reviewers' comments.

Given recent work showing slightly elevated family-wise error rates when using the multiple comparisons techniques described by the authors 1, it would be good if the authors could validate clusters using a permutation testing procedure 2.

We thank the reviewer for pointing us to this work. A close inspection of the analyses by Eklund and et al. shows family-wise error (FWE) rates are actually do not seem to be elevated when using a threshold of $p < 0.001$ at the voxel-level and of $p < 0.05$ FWE at the cluster level in SPM; see Fig 1B in the article by Eklund et al. We now explicitly mention this in the methods section (SI).

All reported clusters are familywise error (FWE) corrected for multiple comparisons at $p < 0.05$ using the SPM random field theory based approach. The cluster-defining threshold was $p < 0.001$. At this voxel-inclusion threshold of $p < 0.001$, FWE-rates in SPM do not seem to be elevated⁵.

Reference ⁵ is the suggested paper by Eklund et al.

Supplementary Figure 1: this information should be included in the main text.

As detailed above, we have considerably revised the provided figures and now include overall more plots of participants' behavior binned according to the different considered variables in particular in (**Fig 2**). As suggested by the reviewer, we now use the continuous decision variable of the optimal policy and not the binary optimal policies. See **Figs 2, S3, & S5** plotted above in response to earlier comments as well as **Figs S4 & Fig S8**.

1. Eklund, A., Nichols, T. E. & Knutsson, H. Cluster failure: Why fMRI inferences for spatial extent have inflated false-positive rates. Proceedings of the National Academy of Sciences 113, 7900–7905 (2016).

2. Nichols, T. E. & Holmes, A. P. Nonparametric permutation tests for functional neuroimaging: a primer with examples. Hum. Brain Mapp. 15, 1–25 (2002).

Reviewer #3 (Remarks to the Author):

This manuscript is concerned with a ‘homeostatic’ sequential decision making task and its underlying neural correlates. As such there is some novelty in both the decision models and neural findings. Particularly, the potential of having interdependent but partly distinct computations of more heuristic and more optimal processes emerge over time during sequential choice, with more optimal behavioral flexibility being driven by ACC, is exciting. It also adds to the current debate over the nature of value signals in foraging-like decision tasks, rejecting a view of one unitary factor driving activity.

Thank you for this positive evaluation of our manuscript.

However, the framing and interpretation of results are not always straightforward and should be improved upon. Furthermore, specifically the description of their findings in regards to the current literature are quite selective and convoluted, neglecting to highlight some major differences between the current study any the majority of other decision tasks, while also trying to adopt terminology from many different approaches, presumably in an attempt to keep many different factions happy. While I also had some methodological concerns, about half of my comments relate to the framing. Therefore, Points 1-6 and 8 should all mostly be addressable through textual changes i.e. improvement in framing. To help with this, I tried to point out specific sections in the manuscript that could be reframed/reworded (These specific recommendations also makes the review appear unusually long)

We are very grateful to the reviewer for taking the time to formulate these specific comments, which have helped us tremendously to improve our manuscript. Please see below for specific answers.

Surprisingly the authors appear to argue a heuristic and optimal policy is always computed in parallel, but not sufficiently address the question of why to even bother with heuristics if an optimal computation is possible. The alternative explanation that optimal policies are only computed selectively, particularly when heuristics fail and thus in part ACC being more engaged when there is a need for optimal computations is wholly compatible with their results and more likely.

Thank you for highlighting this. We now provide analyses and careful discussions along these lines. We refrain from an interpretation in terms of parallel processing. We realized that our use of the word “parallel” was misleading. We did not want to imply that there are “always” two computations that are strictly running “in parallel.” Please see our specific answers and the textual changes below.

Furthermore, their conflict or “difficulty” representation effects can be described equally well with a more neurophysiologically grounded explanation, explaining decision making using concept such as evidence accumulation and mutual inhibition, without having to resort to a representation of a psychological experience of conflict or explicit signaling of difficulty itself. More importantly, there are many different aspects of the task that could be labeled difficulty or conflict including decision horizon, and therefore a description of the results in terms of absolute value differences or choice uncertainty/value driven choice probability would be more appropriate.

We agree and have considerably revised the whole text to reflect this. We now refrain from using the words “conflict” and “difficulty” and instead use terms “discrepancy” and “choice uncertainty,” respectively. We did not want to imply a “psychological experience” of conflict or difficulty in our previous version and appreciate the reviewer’s comment that using the words “conflict” and “difficulty” could create an incorrect understanding. We tried to clearly link the terms “discrepancy” and “choice uncertainty” to our task; both in the text and in **Table 1**, which lists all task variables.

We think that our current fMRI study does not provide the necessary neurophysiological data and the necessary temporal resolution to allow detailed conclusions about evidence accumulation and mutual inhibition. We address the intriguing notion of evidence accumulation in a new section in the discussion (see below).

(More detailed comments below)

Major Comments:

- 1) **The current manuscript is relatively unusual within the decision domain, as value is solely based on whether participants “survive” a series of five sequential decisions after which they receive a fixed point independent reward. As such the point of the task is to maximize the probability of “survival” or homeostasis rather than maximizing the overall number of points in the experiments, as is normally optimal in other decision experiments. Therefore it is tricky to compare this study’s value regressors with the ones from other studies, which should be sufficiently highlighted. Furthermore, it might deserve a mention in the title and definitely in the abstract.**

We agree. This feature allowed us to compute non-trivial types of *a priori* optimal policies. We highlight this more in the abstract and throughout the manuscript.

Abstract:

To probe the potential interplay between heuristic and optimal computations, we developed a novel sequential decision-making task in which rewards only depend on final outcomes in mini-blocks of five consecutive trials. Therefore, optimal choices necessitate evaluating five sequential decisions and probabilistic outcomes.

Introduction: Page 4

In our task, participants were endowed with varying “energy resources,” depicted graphically as an energy bar. Participants were financially rewarded if they “survived” over a maximum of five time steps, called “days,” within a given mini-block of trials, called “forest” (**Fig 1A**). That is, participants could not simply gain monetary rewards in all trials but only received a payoff if a series of five consecutive decisions and probabilistic outcomes resulted in a final energy level above zero.

Previously, we used the binary optimal policy (i.e., “foraging” or “waiting”). We now use a continuous variable for the value differences between the two choice options according to the optimal policy (i.e., the value of “foraging” minus the value of “waiting”). This notion of “value” is actually concordant with the common understanding of the term “value” in Markov decision-making tasks (e.g., in multi-step learning tasks). We now provide a more nuanced explanation in the last paragraph of the introduction. Pages 4-5

We computed the *a priori* optimal policy that minimizes starvation probability according to a finite time horizon of five days for each combination of energy state, weather type, and day within a forest. Because the two choice options, foraging or waiting, vary in their relative expected values, the optimal policy allows for some degree of stochasticity. In line with the general notion that human choices are sensitive to value differences between choice options, we assumed choice would be determined by the continuous value differences between foraging and waiting, computed according to the optimal policy. In the following, we use the term “optimal policy” to refer to these continuous value differences between the choice options and not to the deterministically better option. Unless otherwise specified, we refer to the optimal policy according to the normative finite time horizon of five steps.

We also mention this point in the first sentences of the discussion. Page 21:

This study addresses the neural computations required to make sequential decisions over multiple probabilistic time steps. That is, participants did not have to maximize their overall gains in independent trials but had to make sure to reach a given boundary in a number of consecutive and probabilistic steps. Such decisions arise in many biological or economic

contexts for example when decision makers aim at maintaining energetic homeostasis or retaining liquidity in business transactions.

The task is furthermore lacking some of the sequential elements of other experiments as participants accumulate points at every step. However, the authors note that already in the manuscript.

We now also mention this more clearly in the task description in the caption of **Fig 1A**.

Participants “forage” within 240 different types of mini-blocks, called “forests”. They are monetarily rewarded for averting “starvation” (keeping “energy bar” above zero) at day five in the forest (i.e., the last trial within a mini-block).

- 2) As mentioned above, the authors should change the language to follow more of a value terminology, as this is closer to the data in a value based decision task and allows for the use of more mechanistic descriptions related to increasingly detailed models of decision making as an evidence accumulation process. E.g. value, choice probability and uncertainty make more sense than decision difficulty as the later is a meta process/judgment beyond the decision process itself. This is important as the circuit is presumably trying to make a decision, and adapt optimally when a heuristic value estimate is insufficient.**

We thank the reviewer for this pertinent comment. We did not intend to refer to meta-judgements of difficulty or conflict and this comment made us realize that the whole study could be misunderstood due to the misleading use of the terms “difficulty” and “conflict.” We now follow the reviewer’s suggestions and revised the whole text in accordance with a “value terminology” as this indeed describes the intended meaning much better. We now use the terms “choice uncertainty” (previously “difficulty”) and “discrepancy” between the choice probabilities of the two decision variables (previously “conflict”)

Related to this the concept of comparing heuristic with optimal in parallel every trial is strange. Why should the agent bother with a heuristic if it eventually compares it the optimal anyway? At some point in the manuscript this statement is qualified and it is suggested that the optimal is partly conditional on the heuristic, but it is not nearly clear enough in many parts of the manuscript (see also point 9).

The reviewers’ comment made us realize that our previous description using the term “parallel” was misleading and conferred an unduly strong interpretation in terms of two computations that would always run “in parallel” at the same time within a trial. We now offer a more careful interpretation in terms of “integrated” processing. We previously intended to imply that participants compute both policies in some trials but we did not intend to convey that this computation was done at the same moment in time within a trial, or to the same degree in all trials. Please find the changes results sections on RT effects pasted below: Pages 13-16:

Reaction times increased with choice uncertainties of the heuristic and optimal policies

Models of choice data indicate that participants used both a heuristic policy, i.e., the probability of foraging success, and the optimal policy. Consequently, we predicted that reaction times (RTs) should reflect the choice uncertainties associated with these two variables. Indeed, RTs were slower when choice uncertainties were high (see **Fig 3A-D** for the fMRI sample and **Fig S8A-D** for the behavioral sample, see **Table S4** for statistics). For example, RTs were highest when the probability of foraging success was 0.5 and thus neither foraging nor waiting was clearly favored by this metric. Choice uncertainties were quantified on the basis of the mean parameter estimates of the choice models from the behavioral sample. For the fMRI sample, choice uncertainty calculations thus rely on independent data.

Choices based on the heuristic were faster than those based on the optimal policy

Given that participants' choices seem to integrate heuristic and optimal policy computations, the question arises how these computations relate to each other. A first straightforward prediction is that choices following the heuristic policy should be faster than those following the optimal policy, if the heuristic is easier to compute than the optimal policy. In many trials, both variables made the same prescriptions and thus these trials cannot be used to disambiguate choices made according to one or the other variable. Therefore, we identified the subset of trials in which the two made opposite prescriptions (i.e., trials with opposite choice probabilities according to the mean parameter estimates of the choice model from the independent behavioral sample). Within this subset, mean RTs for trials in which participants' choices followed the probability of foraging success were faster than mean RTs for trials in which choices followed the optimal policy (mean difference \pm SD: fMRI sample: 90.7 ± 109.2 ms; $t(27)=4.40$; $p<0.001$; behavioral sample: 147.5 ± 96.2 ms; $t(20)=7.02$; $p<0.001$; choice probabilities did not differ between these trials; both $p's>0.4$).

In addition, a linear mixed effects model of RT data provided evidence for a relatively more pronounced influence of the choice uncertainty of the heuristic compared to the choice uncertainty of the optimal policy (**Table S4**). That is, choice uncertainty under the heuristic policy related more strongly to RTs than choice uncertainty under the optimal policy.

Reaction times increased with discrepancies between the heuristic and optimal policies

Integrated computation of heuristic and optimal policy makes a second crucial prediction for RT data: Decisions should take longer when the two variables make discrepant prescriptions. For example, RTs should be slower when the heuristic prescribes waiting but the optimal policy prescribes foraging or vice versa. We quantified these discrepancies between the two variables as the absolute differences in choice probabilities (which were based on the mean parameter estimates of the choice model from the independent behavioral sample). Indeed, decisions were slower when discrepancies between the variables were larger. This effect was present in addition to influences of choice uncertainties (see **Fig 3E** for the fMRI sample and **Fig S8E** for the behavioral sample).

Overall, log-transformed RT data were well described by a linear mixed effects model that included the heuristic and the optimal policies themselves, their associated choice uncertainties, and the discrepancies in the choice probabilities of the two policies (**Figs 3 & S8**; see **Figs S3B & S4B** for the parameter estimates of the full RT model fitted on the basis of individual participants and for **Fig S5** for further posterior predictive checks; see **Table S4** for results obtained from a linear mixed effects model of log-transformed RTs, which were qualitatively the same as results on untransformed RTs). Motivated by this model of RT data, we also tested whether interactions of choice uncertainties or discrepancy influenced choices but found no decisive effects (**Table S3**).

Related to this, the authors have a quite contorted description P17, L282-4 “This provides crucial evidence that both heuristic and an optimal policies determined participants behavior, with the relative contribution of the two depending on the difficulty or reliability of the heuristic”. However, the heuristic isn’t more difficult, but rather does not guide behavior in one way or other, i.e. there is no clear value in either response. A better description of the results would be that lower order decision uncertainty or lack of reliability forces the heuristic system to engage other systems and compute higher order value instead to increase choice certainty and also accuracy.

We deleted this unclear description in the revised the whole methods accordingly (see answers to same comment above). We now included a section in the discussion that follows several of the reviewer's suggestions. Page 22.

Our behavioral data suggest interdependent but partly distinct computations of heuristic and optimal processes during sequential choice. But what is the relationship between the two

policies? Analyses of reaction times (RTs) showed that the choice uncertainty of both policies—but in particular of the heuristic—slowed RTs. Intriguingly, the discrepancies in choice probability between the two policies also led to longer RTs, which provides crucial evidence for an integrated computation of heuristic and optimal policies. We interpret these findings as pointing toward progressive computational processes such that the approximations provided by the heuristic are abandoned if the associated choice uncertainties turn out to be too high. That is, an insufficient choice certainty of the heuristic metric suggests that it is worthwhile to engage in a deeper search in form of a full-blown optimal policy computation.

In addition, we discuss the intriguing notion of an internal evidence accumulation process during decision-making in a section that immediately follows the above paragraph in the discussion. Page 22.

Integrated computation of heuristic and optimal policies could be understood in terms of an evidence accumulation process: In potentially increasingly sophisticated computations evidence for the values of the choice options is progressively acquired according to different (possibly competing) policies—probably until the associated choice uncertainties become sufficiently small. Such an evidence accumulation process would change over time within a trial and also vary between different trials.

Additionally, the ecological approach overall suggests that the animal is trying to gather things of value, and ensure survival. Therefore, the terminology should be guided by value, not split into different forms of difficulty, particularly as the whole concept of conflict and difficulty is that it is an atomic unit that can't be split.

We entirely agree that this is another important reason to refrain from using the words difficulty and conflict.

Furthermore to prove the psychological phenomenon of difficulty representation, one would have to show any changes that are commonly associated with difficulty, such as increases of reaction time through non-value related manipulations, induce equivalent effects (unlikely given Stoll et al. [2016, Nature Communication]). If that is not done, the resulting neural findings should more accurately be described as value and certainty related, as that is what is being explicitly manipulated. Additionally, there is no point in using the term difficulty, which is meant to be a catch-all for many different factors combined in just how hard it is and then discuss only one bit of the task. If one talks about difficulty it has to be overall difficulty of doing the task or nothing. Looking at the text itself, specifically, P21 L363 “Difficulty was represented in several brain regions...” is a fallacy of assuming representation from a correlation. More specifically, only because the signal was larger when no quick clear decision could be made that doesn't mean there is a representation of difficulty. Any competitive system should be sensitive to value similarity and uncertainty. Also, it isn't the difficulty of the heuristic policy.

Thank you as outlined above we now use the terms suggested by the reviewer and do no longer refer to the misleading concept of “difficulty.” We did not want to refer to the overall task difficulty but as the reviewer correctly points out to the choice uncertainty related to the heuristic and optimal policies. We also did not want to imply an explicit “representation” of (meta-cognitive) “difficulty.” Instead, we now specifically relate the choice uncertainties of both the heuristic policy and the optimal policy to RTs and BOLD signals. We did not want to imply that non-value related manipulations would induce equivalent effects.

Thank you for pointing us to the interesting paper by Stoll et al., which suggests an intriguing follow-up question for our manuscript: How would an optimal strategy look like when there is a requirement (or opportunity) to seek information about the environment? How would information seeking alter the interplay between optimal and heuristic strategies? We now mention this idea and cite the paper by Stoll et al. (ref 40) in the discussion. Page 24:

Here, we were interested in how humans make sequential decisions once they have acquired sufficiently precise estimates of the relevant candidate decision variables. This is why all variables were explicitly signaled or could be deterministically calculated. There was no requirement to learn, explore, or infer, unknown, uncertain, or unobservable, states. This distinguishes our task fundamentally from learning tasks aimed at comparing explicitly signaled versus previously trained values tasks designed to assess model-based versus model-free reinforcement learning^{6,30–33,36–40}. Our approach could be extended to include learning or information seeking (cf. ⁴⁰) such that the optimal policy would require reducing uncertainty about environmental states (as can be implemented in a partially observable Markov decision processes⁴¹).

- 3) Furthermore, I think the concepts of competition and accumulation of evidence as well as decision thresholds are far better suited to describe the data. In other words, if the agent hasn't made a decision quickly with a heuristic because there is no clear value or a lack of a clearly superior option, additional mechanisms such as dACC are engaged. Presumably, although I couldn't find it in the manuscript there is also an effect of reaction time in ACC, as many studies have found decision length associated activity with other factors riding on top.**

Yes, the reviewer is correct. The effect of (log-transformed) RTs was and is included in the GLMs. We now present this contrast in the main text in **Fig 5D** pasted below (and in **Table S5**). Importantly, the dACC was associated with the discrepancies in the prescriptions of the two policies in addition to a simple effect of RTs (see the prominent effect in **Fig 5C**).

For the reviewer's convenience, we paste most parts of the fMRI results along with **Figs 4 & 5** here. These results are also relevant for several of the following comments. Pages 17-19:

MPFC activity correlated with differences in choice values both for heuristic and optimal policies

The momentary probability of foraging success, i.e., the variable underlying the heuristic policy, correlated positively with BOLD signals in a posterior part of the dorsal MPFC (DMPFC, extending into pre-supplementary motor area, pre-SMA), in bilateral intraparietal sulcus (IPS), and the left frontal pole among other regions (**Fig 4A**, see **Table S5** for all fMRI results in the choice phase). The same variable correlated negatively with signals in the perigenual anterior cingulate cortex (ACC), extending into the ventral MPFC (VMPFC; **Fig 4B**).

The value differences of foraging versus waiting according to the optimal policy correlated positively with activity in perigenual ACC and mid-cingulate cortex (**Fig 4C**). That is, parts of the MPFC were relatively more active when waiting was favored by the heuristic and when foraging was favored by the optimal policy. This suggests an overall involvement of the MPFC in computing differences in choice value of the variables that explained participants' behavior.

MPFC activity also reflected choice uncertainties and the discrepancy between heuristic and optimal policy

Lower choice uncertainty of the heuristic was related to increased BOLD signals in an anterior part of the VMPFC, dorsal MPFC regions as well as to the inferior frontal gyrus (IFG) and the posterior cingulate cortex, among other regions (**Fig 5A**, see **Table S5** for a list of all clusters). Lower choice uncertainty of the optimal policy scaled with activity in DMPFC, extending into ACC, and in the IFG (**Fig 5B**).

Crucially, as in our RT data, we found evidence for an integrated computation of the heuristic and the optimal policy: DMPFC activity correlated in a trial-by-trial fashion with the discrepancies between the two policies (i.e., the absolute differences in their associated choice probabilities; **Fig 5C**). This DMPFC region extended into the pre-SMA and the ACC. The same metric correlated with BOLD signals in the bilateral dorsal striatum and bilateral IFG.

All correlations of the relevant model variables emerged over and above correlations with log-transformed RTs (**Fig 5D**). Overall, the same regions described above were also identified in another GLM, which additionally included participants' choices themselves as parametric modulators during the choice phase (**Table S6**). The main qualitative difference between the GLMs with and without choices as additional parametric modulator was that in the GLM including choices the DMPFC cluster related to lower choice uncertainty of the optimal policy failed to reach significance (cf. **Fig 4B**). A GLM that only included participants' choices as parametric modulator did not reveal all the regions described above to be related to the heuristic and optimal policies, their choice uncertainties as well as the discrepancies between the two policies (**Fig S9, Table S7**). This suggests that the variables identified from choice and RT models accounted for variance in the fMRI beyond the variance explained by choice *per se*.

Finally, classic reward regions^{13,14} (including ventromedial prefrontal cortex, striatum, and posterior cingulate cortex) tracked the realized outcomes, that is the impact of participants' decisions on their internal energy state (**Fig S10, Table S8**).

Fig 4. Statistical parametric maps for the BOLD signals related to heuristic and the optimal policies during the choice phase (overlay on group average T1-weighted image in MNI space; clusters are whole-brain FWE corrected for multiple comparisons at $p < 0.05$ with a cluster-defining threshold of $p < 0.001$). See **Table S5** for a list of all clusters.

Fig 5. Statistical parametric maps for the respective uncertainties of heuristic and optimal policies, the discrepancies in their choice probabilities, and log-transformed RTs during the choice phase (overlay on group average T1-weighted image in MNI space; clusters are whole-brain FWE corrected for multiple comparisons at $p < 0.05$ with a cluster-defining threshold of $p < 0.001$). See **Table S5** for a list of all clusters. See **Fig S9** for results from a GLM that only includes participants' choices as parametric modulators. See **Fig S10** for BOLD signals during the outcome phase.

This could then mean that ACC is increasingly active when heuristic certainty is low and therefore decisions aren't made quickly (therefore also a larger effect of heuristic on reaction time), eventually coding the relative gain from using a better understanding of the task structure (task model etc). This is encapsulated both by the neural effects of the difference in choice probability between heuristic and optimal, and the overlapping increase by optimal policy value. This in itself is potentially a very interesting finding if described in those terms. Related to the idea of overlapping or interrelated accumulation processes, the repeated statement of Parallel policies is slightly misleading e.g. P20 L336-8 "computed in parallel-although when they are conflicting, only one of them can be implemented." As outlined above (and further in point 4) it is very tricky to show parallel processes and I think the data can be explained without going hard on the idea of always both policies being computed.

We are very grateful to the reviewer for making these specific suggestions and include a section in the discussion. Page 24:

The discrepancies between the two employed policies showed a positive trial-by-trial relationship with a prominent cluster in the region of the DMPFC, extending into the pre-SMA and the ACC (in addition to relationships bilateral dorsal striatum and IFG). The DMPFC cluster overlaps with regions classically associated with multiple types of decision discrepancies^{33,34}. Thus, our finding relating discrepancies between the heuristic and optimal policies to a larger part of the DMPFC could potentially indicate that this DMPFC region becomes increasingly engaged when the computations of the two policies prescribe divergent choices such that progressive evidence accumulation and competition processes are required for making a decision.

The reviewer's comments have convinced us to refrain from interpreting our data in terms of parallel processing. As mentioned above, we now use the term "integrated processing" and provide discussion points along the suggestions of the reviewer.

Furthermore, related to both aspects of this point P23 L433-5 "Brain activity associated with conflict between heuristic and optimal policies points towards a mechanism of parallel computation" Could be more mechanistically described as possibly competitive value/decision evidence accumulation processes, although there is also the possibility of one integrated process that changes over time as additional types of information get fed in.

We thank the reviewer for pointing us toward the concept of evidence accumulation, which we now discuss. The strong sentence about parallel computation is deleted. See the general section on evidence accumulation, which we re-paste below for the reviewer's convenience: Page 22:

Integrated computation of heuristic and optimal policies could be understood in terms of an evidence accumulation process: In potentially increasingly sophisticated computations evidence for the values of the choice options is progressively acquired according to different (possibly competing) policies—probably until the associated choice uncertainties become sufficiently small. Such an evidence accumulation process would change over time within a trial and also vary between different trials.

P20 L340-2 "The higher the decision difficulty of the heuristic policy the more the optimal policy was applied..." Could be rephrased as rather saying when neither option is clearly more valuable on a heuristic level, then a more sophisticated computation is made/ a more optimal policy is computed.

Thank you for this suggestion, which we included in the following discussion section (re-pasted for the reviewer's convenience): Page 22:

We interpret these findings as pointing toward progressive computational processes such that the approximations provided by the heuristic are abandoned if the associated choice uncertainties turn out to be too high. That is, an insufficient choice certainty of the heuristic metric suggests that it is worthwhile to engage in a deeper search in form of a full-blown optimal policy computation.

- 4) **More generally, the authors substantiate their parallel processing claim mostly on one RT analysis. There is weak evidence for the pure “conflict” of two strategies rather than a process or interrelated processes, which change over time and between trials. Rather it could be an increasingly sophisticated computation over time, possibly with an increasing probability of optimal policy computation being initiated meaning computations aren’t completely sequential either.**

Thank you. We completely agree. As described above, we included this notion into the discussion, taking many of the formulations of the reviewer as a basis. We specifically mention the variability of such processing across time and between trials: Page 22:

Such an evidence accumulation process would change over time within a trial and also vary between different trials.

In the same paragraph, we also highlight the need to investigate the precise temporal unfolding of these processes: Page 22-23:

Identifying the precise temporal requirements of information integration processes is an interesting and challenging avenue for future research.

Independently of that concern, showing truly parallel computations rather than an averaged mixture is chronically difficult to prove and participants might just have been additionally slowed in parts where the two diverge because that corresponds to the parts of decision space that is misfit or because, although both processes aren’t parallel the optimal policy is increasingly more likely to be computed and therefore slow down decisions when the heuristic policy is likely to fail (e.g. long horizons or large dot ranges making p-forage success a poor substitute).

We completely agree that showing truly parallel processing is notoriously difficult to demonstrate for the reasons specified by the reviewer. This is especially pertinent in our task where computing the optimal policy requires some kind of processing of the information provided by the heuristic. We therefore refrain from going hard on the idea of parallel processing.

Additionally, it looks like the effect in figure 4 B isn’t linearly increasing but only to a point and then decreasing again, suggesting it might be driven by another potentially somewhat related variable like horizon, energy level or trial number in forest.

The revised analyses with the corrected calculation of the continuous decision variable of the optimal policy show a considerably more monotonic and linear increase of RTs than in the previous version of our manuscript (see **Fig 3E** below for the fMRI sample and **Fig S8** for the behavioral sample).

Fig 3. Reaction time data of the fMRI sample

We tested the relationship between RTs and variables associated with the heuristic and optimal policies. Since the probability of foraging success emerged as the best predictor of participants' choices, we only included this but not any other heuristic in the model of RT data. RTs relate to (A) the probability of foraging success and also weakly to (B) the optimal policy. Importantly, RTs become slower with (C & D) increasing choice uncertainties of these two variables and (E) higher discrepancies in their prescriptions. Posterior predictive checks show that RT data were well captured by a model that includes the five depicted variables.

Data are binned. Error bars are SEM. In several cases, error bars are smaller than the size of the markers, which scale with the average number of trials contributing to the respective data points. See Fig S8 for the behavioral sample and Figs S3 & S4 for the parameter estimates of the full RT model and Fig S5 for posterior predictive checks of the RT model with data split differently. See Table S4 for statistical inferences obtained from a linear mixed effects model.

If the authors want to prove truly parallel computations, they should resort to a method with higher temporal resolution or else settle with talking about interrelated, but not necessarily parallel computations.

We agree and settle with talking about interrelated processes in this fMRI study.

- 5) Optimal policy effect in ACC/MPFC is interesting as it suggests positive value related effects do exist in ACC, particularly for the more sophisticated computations. This is important, as the rest of the manuscript reads a little as if there was no positive value effects in ACC.**

Thank you for stressing this point about the positive value effect for the optimal policy. In our revised version, we used corrected and revised analyses (continuous value differences of foraging versus waiting according to the optimal policy instead of the deterministic optimal policy per se). We additionally included the choice uncertainty of the “continuous optimal policy” as a parametric regressor in the GLM. We now find that the optimal policy metric policy correlated positively with activity in perigenual ACC and mid-cingulate cortex. (**Fig 4C** pasted above for comment 3). The choice unchoice uncertainty of the optimal policy scaled negatively with activity in DMPFC, extending into ACC, and in the IFG (**Fig 5B** pasted above). Therefore, the ACC/MPFC region the reviewer is referring to scales with choice uncertainty (i.e., higher BOLD signals for higher choice certainty), which we discuss: Page 24:

Multimodal integration regions such as MPFC and IFG were associated with the optimal policy and also the choice uncertainty of the optimal policy. These regions have often been observed in studies testing for brain activity related to model-based learning processes^{30,31}. Indeed, optimal decisions in our task bears considerably resemblance with model-based learning^{31–33} since both involve searching across a tree of probabilistic future states. Here, we demonstrate brain activity when participants evaluated decision trees with extended time horizons independent of the uncertainties arising during learning. In particular, we found that a dorsal region of the MPFC correlates with the choice uncertainty of the optimal policy. This region seems to be especially well positioned to integrated different types of decision signals related to reward values and actions^{17,18}. An intriguing possibility is that this region may generally be related to the uncertainties in recursive evaluations of a tree of probabilistic future states, which is a key feature for inferring optimal solutions in many realistic tasks.

Furthermore, the lack of a strong heuristic difficulty effects or other definitions of difficulty such as horizon length should be mentioned.

Maybe, we misunderstand the reviewer here. There are actually neural effects of “heuristic difficulty”—or according to the revised terminology—of “choice uncertainty of the heuristic.” This metric was negatively related to several brain regions, notably parts of the MPFC (see Fig 5A and fMRI results section pasted above).

- 6) The negative abs. value difference/ choice probability effects based on p-forage success in perigenual ACC and vmPFC are quite surprising given a wealth of studies showing positive value effects in vmPFC. This should be at least discussed, as it is quite unusual to have this bit of medial prefrontal cortex not activate with value.**

Thank you very much for highlighting this finding. We now explicitly present this effect in **Fig 4B** (see pasted above in response to comment 3). We would argue that this is not necessarily a “pure negative value effect.” Indeed, we hold it likely that this region is related to the “positive” value of the alternative choice option, i.e., the action of “waiting.” Although we cannot unambiguously dissect this interpretation from other possibilities, it is actually in line with the interpretation for this region given in a previous study on virtual foraging (Kolling et al., 2012 = ref 2). We now carefully discuss this: Page 23:

The probabilities of foraging success were negatively related to signals in the perigenual ACC and the VMPFC. When probabilities of foraging success were small, the choice values of the “waiting” action (which always entailed a sure loss of the one point) were higher than the choice values of the “foraging” action (which resulted in a loss of two points or varying gains according to the probabilities of foraging success). Consequently, participants were more likely to choose waiting rather than foraging when the probabilities of foraging success were small. It is therefore a possibility that this region of the perigenual ACC and VMPFC scaled positively with the value of the waiting action which entailed a sure outcome. This notion is consistent with a general role of this region in flexible value encoding and previous findings that this region negatively relates to the values of unchosen options².

Furthermore, when looking at the supplementary material, confusingly it says negative p-foraging success, which is not quite the same as choice difficulty using p-foraging success, which was used in the regression. This should definitely be clarified, as a negative effect of foraging success is quite different conceptually than a negative abs. value difference effect/choice p effect.

We apologize for the confusing presentation, which we have now clarified (by dedicating considerable more space to the presentation of the fMRI results). The effect in the perigenual ACC / VMPFC (peak: xyz: 6, 33, 6) to which we think the reviewer refers here (**Fig 4B** pasted above) is negatively related to the probabilities of foraging success themselves—and not to the associated “choice difficulty / uncertainty”. That is, the *lower* the probability of gaining the associated gain magnitude, the *higher* the BOLD signals in this perigenual ACC / VMPFC region. The negative relation of BOLD signals to the choice uncertainty of the probabilities is presented in **Fig 5A**. Indeed, this metric also shows a negative relation to a region in the perigenual ACC / VMPFC (peak: xyz: 9, 59, -2). This region extends more anteriorly. Here, the *lower* the choice *uncertainty* (i.e., the *higher* the choice *certainty*), the *higher* the BOLD signals.

- 7) It is a bit strange that p forage success is such a strong driver of the behavioural effects and very close to optimal. It means that the other aspects of value feeding into overall survival probability do not have very large range. From the manuscript it wasn't clear whether there was just little variance in some of the other factors, such as magnitude. It might be nice to see a bit more of the schedule they ran, such as magnitude spread and variance etc.**

We now provide plots with the value difference according to the optimal policy (**Fig S1**) and with behavioral data (**Fig S3 & S4**) binned according to all candidate variables, which shows the magnitude spread and variance of all these considered variables. This plots show that *per se* limited variance was for most of the other heuristics an unlikely reason why these other heuristics performed worse than the probability of foraging success. Additionally, we added the grand mean of the parameters along with the mean within-subject SD in addition to the theoretical possible range into **Table 1**.

Value differences according to optimal policy in relation to the heuristic candidate variables:

Fig S1. Relation of optimal policy and heuristic variables

Shared variance between all 10 candidate variables and relationships with the optimal policy (according to the normative horizon of five time steps) with all 9 heuristic variables considered. Data are binned and arranged in the same way as below in **Figs S3 & S4**, which show the empirical relationship between data and fitted model. “Weather type” and “wins-stay-loose-shift” are binary variables. See **Table 1** for a list of all variables.

Choice behavior split according to those heuristic candidate variables that did not explain behavior:

Fig S3. fMRI sample: Posterior predictive checks according to all heuristics and parameter estimates for full models including all candidate variables

(C) Posterior predictive checks show that the winning model, which includes the probability of foraging success and the optimal policy, captures the empirical relationship between all other 8 other heuristic variables. “Weather type” and “wins-stay-loose-shift” are binary variables. See **Table 1** for a list of all variables.

[...]

See **Fig S4** for the same plots for the behavioral sample.

First part of **Table 1**:

Table 1. Task variables overview

Variable name	Explanation	Theoretically possible values of this variable in the task	Example value of this variable (as in Fig 1A choice phase)	Grand mean of variable across fMRI participants (mean within participant SD)
Heuristic variables				
Probability of foraging success (p with $q=1-p$)	Momentary probability that the participant can gain a certain magnitude of energy points (versus losing two energy points). The participant can infer this probability by counting the number of subfields with gains (i.e., blue dots in Fig 1A) in the grid that contains 10 subfields.	0.1 to 0.9 in steps of 0.1	0.6	0.55 (0.24)
Magnitude of foraging gain g	Momentary magnitude of the possible gain if foraging is successful. This is depicted by the number of (blue) “gain dots” per subfield of the grid.	0 to 4 in steps of 1	1	1.97 (1.41)
Expected value (EV) of the foraging option	Momentary probability of foraging success multiplied by the corresponding magnitude of foraging gain g plus (1- probability of foraging success) multiplied by the loss incurred for unsuccessful foraging, which is always -2. The EV of the waiting option is always -1.	-1.8 to 3.8	-0.2	-0.14 (1.13)
Energy state s	Current state of the energy bar. An energy state of zero is synonymous with starvation.	0 to 5 in steps of 1	2	2.97 (1.09)
Weather type	Each forest type specifies two weather types that can be roughly classified as “good” or “bad” depending on whether they imply a lower or higher probability of starvation. Weather types are relative to each other (i.e., a given combination of p and g can be the good weather type if paired with a relatively worse weather type with	Categorical variable: 1 “bad” or 2 “good”	2 “good”	1.50 (0.5)

Days past in a forest (i.e., number of time steps t)	lower p and g, or the bad weather type if paired with a relatively better weather type, higher p and g). Participants remain within a given forest (i.e., mini-block) for up to 5 days (i.e., trials). The number of days is not explicitly depicted on the screen but participants can easily infer it by counting the number of choice phases after the last occurrence of the forest phase.	1 to 5 in steps of 1	1	1.57 (0.91)
Change between past and current energy states	Participants might track the difference between their energy states in the past trial and the current trial (within and across forests).	-2 to +4 in steps of 1 (maximum loss was 2 energy points & maximum gain was 4 energy points)	Not available in the figure because the change depends on the previous trial that is not depicted. In the next choice phase, the change in energy states is +1.	-0.90 (1.55)
“Win-stay-loose-shift” (WSLS) strategy	Participants might use a strategy, which prescribes foraging if the energy state increased with respect to the past trial and waiting if the energy state decreased. WSLS is a binarized version of the change between past and current energy states	1 “energy state increased” or 0 “energy state decreased”	Not available in the figure because previous trial not depicted. In next choice phase WSLS is 1 “energy state increased”	0.38 (0.48)

- 8) **Title should be changed to more appropriately reflect the content of the study (see also comment 1). At the very least, it should become apparent that the study is a sequential decision task. “Emerging computations of heuristic and optimal decision value in the human cortex” or “The neural signature of heuristic and optimal decision value in homeostatic sequential decisions”, might convey this point a bit more (both are only meant as illustrations not firm suggestions. I do think however, the title should be less general)**

Thank you for this very helpful suggestion. We changed the title to
Heuristic and optimal policy computations in the human brain during sequential decision-making

- 9) **There is an interesting analysis relating to heuristic choice certainty modulating the use of heuristic vs optimal model. However, this suggests rather what I discussed above, a conditional computation of higher order value if and when necessary, not a purely parallel computation. This should be elaborated on and maybe an equivalent neural analysis should be run.**

In the revised version, we use the continuous value differences between the two choice options according to the optimal policy and not the binary optimal policy. Additionally, we corrected the calculations of the optimal policy and now use independent data for deriving choice uncertainties (and discrepancies). See our answer below to the same comment. With these changes, we do no longer find a decisive effect of the choice model with choice uncertainty being better than the model without (**Table S3**). Still, we find strong and clear-cut effects of choice uncertainty of the heuristic being related to RTs (**Fig 3** pasted above in response to comment 4) and to neural data (**Fig 5** pasted above in response to comment 4).

Also, a bit more basic concern, if true, is that it looks from the supplementary methods that an individually fit softmax is used in order to derive the difficulty of the heuristics in order to scale the value of foraging success. If this is true choices are used twice for fitting, making statistics biased and potentially invalid as the difficulty adjustment upscales the part of p-foraging success with the largest changes in decision based on p-foraging success. (See P12 L295 in supplements)

Thank you very much for this pertinent comment. Previously, we indeed used individual participants' data twice in the way the reviewer describes. We now changed this so that for the fMRI sample the described metrics are based on independent data from the behavioral sample. This is a crucial change in analyses of the revised manuscript.

We describe this in the main text in the results: Page 13:

Choice uncertainties were quantified on the basis of the mean parameter estimates of the choice models from the behavioral sample. For the fMRI sample, choice uncertainty calculations thus rely on independent data.

See also page 15:

Decisions should take longer when the two variables make discrepant prescriptions. For example, RTs should be slower when the heuristic prescribes waiting but the optimal policy prescribes foraging or vice versa. We quantified these discrepancies between the two variables as the absolute differences in choice probabilities (which were based on the mean parameter estimates of the choice model from the independent behavioral sample).

Methods section in the SI: Page 9 of SI:

To base the quantification of choice uncertainties for the fMRI sample on independent data, we used the mean parameter estimates of the choice models from the behavioral sample to derive the logistic functions.

This would also explain why the scaling with difficulty is positive. Conceptually, I would have expected the opposite, i.e. less use of the heuristic information if it is uncertain, relying instead on the optimal policy, while the opposite appears to be the case from the formula. However, I am not so sure about the difficulty effect being positive, although this is how the formula is written as the authors explicitly state the more intuitive correct sign of effect P20 L340-2 “The higher the decision difficulty of the heuristic policy the more the optimal policy was applied...”

Thank you. We apologize that the presentation of the formula was misleading. The effect was indeed in the way the reviewer expected and as stated by us in the text. We now present these formulas more clearly and also test two additional models with discrepancy scaling the use of the two policies. As mentioned above, in the revised version, these models do not decisively outperform the choice model that only contains the probability of foraging success (“p”) and the value difference according to the optimal policy (“optimal policy”). Page 10 of SI:

DV = decision variable

$DV = \beta_0 + \beta_1 * p * (1 - \text{choice uncertainty of } p) + \beta_2 * \text{optimal policy}.$

$DV = \beta_0 + \beta_1 * p + \beta_2 * \text{optimal policy} * (1 - \text{choice uncertainty of optimal policy}).$

$DV = \beta_0 + \beta_1 * p * (1 - \text{choice uncertainty of } p) + \beta_2 * \text{optimal policy} * (1 - \text{choice uncertainty of optimal policy}).$

$DV = \beta_0 + \beta_1 * p * \text{discrepancy} + \beta_2 * \text{optimal policy}.$

$DV = \beta_0 + \beta_1 * p + \beta_2 * \text{optimal policy} * \text{discrepancy}.$

10) I think it would be important to run RT and decision regression with all the important/potentially relevant effects, plotting all of them on a group lvl (mean and error bars). This would be ‘difficulty’ (abs value difference) for heuristic and optimal and ‘difficulty’ of heuristic vs. optimal on RT and all the potential heuristics for the logistic regression on the decision data. This would give the reader a better sense of the data and how the difference of policies, not the main effects separately, drive RT and the relative effect sizes for aspects of the task that could combine to be the thing going beyond the p-forage heuristic.

We have conducted the requested analyses for both choice and RT and provide the relevant plots (Figs S3B & S4B). The plots clearly show that the probability of foraging success is the strongest driver of choice data. The choice uncertainty associated with the probability of foraging success is the strongest driver of RTs.

Fig S3. FMRI sample: Posterior predictive checks according to all heuristics and parameter estimates for full models including all candidate variables

[...]

(D) Mean parameter estimates of full behavioral and RT models that include all 10 or 5 candidate decision variables (which were z-scored). For better visualization the intercept for the RT model is not depicted. Parameter estimates were derived by averaging across parameter estimates of models fit to individual participants. Error bars are SEM.

See **Fig S4** for the same plots for the behavioral sample.

It might furthermore, be useful to bin the data according both optimal and heuristic separately (e.g. separate lines for different optimal bins and points on x axis for heuristic) to show the different effects and potential scaling of effect size (see point 9).

We now provide plots that bin data according to the heuristic (probability of foraging success and according to the optimal policy along with posterior predictive checks (**Fig 2** for the fMRI sample and **Fig S2** for the behavioral sample). We also provide plots that split choice data according to all other heuristic variables (see **Fig S3** pasted above for comment 7). Furthermore, we plot both choice and RT data according to two combinations of heuristics (**Fig S5**).

Fig 2 for the two main variables:

Fig 2. Choice data of the fMRI sample

- (E) Model comparisons show that the probability of foraging success was the best single predictor of participants' behavior. Main plots depict fixed-effects analyses using log-group Bayes factors based on Bayesian Information Criterion (BIC) relative to model #1. Insets show random-effects analyses using protected exceedance probabilities (EP) with the winning model marked. See **Table 1** for a list that specifies the task variables and thus the models tested here.
- (F) Crucially, the *a priori* optimal policy according to a time horizon of five days best explained the remaining variance in participants' choices.
- (G) Posterior predictive checks show that the winning model, which includes the probability of foraging success and the optimal policy, captures the empirical relationship between participants' average choices and the probability of foraging success. Marker sizes which scale with the average number of trials contributing to the respective data points.
- (H) Posterior predictive checks show that the winning model captures the relationship between participants' average choices and the optimal policy according to a horizon of five days (binned value differences of foraging versus waiting).

Data are binned. Error bars are standard errors of the mean (SEM). In several cases, error bars are smaller than the marker sizes. See **Fig S2** for the same plots with data of the behavioral sample. See **Fig S3 & S4** for posterior predictive checks of the winning model with choice data split according to the 8 other heuristics and combinations thereof, and for parameter estimates of a full model including all candidate variables. See **Fig S5** for further posterior predictive checks of the winning model with choices split jointly according to the energy state and the probability of foraging success or the weather type. See **Fig S6** for comparisons of different time horizons. See **Tables S1, S2, & S3** for further model comparisons.

See **Fig S5** for more plots, which show data split according to two heuristics.

A fMRI sample: energy state & p foraging success

B Behavioral sample: energy state & p foraging success

C fMRI sample: weather type & energy state

D Behavioral sample: weather type & energy state

Fig S5. Both samples: Further posterior predictive checks according to two combined variables for models of choice and RT data.

(A & B) Split according to energy state and probability of foraging success.

(C & D) Split according to weather type and energy state.

See legend of **Fig S3** for more information on the logic of these plots. See **Fig 2**.

- 11) The fMRI part of the manuscript is a bit thin. For example, if they believe the ACC is involved in implementing the optimal policy, shouldn't they run a PPI with the value of the optimal policy to test whether it functionally connects more strongly to other value related areas when it tries to steer the system towards more optimal policies?**

We thank the reviewer for this suggestion, which we think an excellent for a follow-up study that is optimized for this question by selecting appropriate numbers of trials for this PPI. We have extended the fMRI part by also showing effects of the choice uncertainty of the optimal policy (and also effects of RTs). We have also included analyses according to two separate GLMs that address 1) how participants' choices alone relate BOLD signals and 2) show that including choice as an additional parametric modulator along with all other variables from the main models gives very similar results.

Minor Comments:

- A) P21 L375 “found neural representation” Isn't this the scaling of the signal by value or a transfer of the same into p forage success? It is unusual to talk about a scalar value signal as a representation, as that might have come from a pattern analysis etc. Therefore, I would suggest just describing it as “IPS and frontal pole signals scaled/increased with heuristic value/p-foraging success”. Also it should be discussed why this positive effect exists, together with negative effects related to overall survival probability in frontal pole in the forest phase (called starvation probability).**

We apologize for our incorrect use of the term “representation.” We did not want to imply that this result was obtained in some kind of pattern analyses. We improved the discussion of this issue and refrain from using the term “representation” throughout the manuscript. In the revised analyses we corrected the calculation of the optimal policy and thus the starvation probabilities under the optimal policy. With these revised calculations, the effect of overall survival probability during the “forest phase” in the frontal pole just failed to reach significance (this is the only major qualitative change in the fMRI results).

- B) I am not sure whether it is mentioned in the manuscript but I couldn't find it. Do subjects need to count the days by themselves or is this cued somehow?**

There were no explicit cues and they had to count this by themselves. We included this information in the caption of **Fig 1**.

The number of days past in a forest was not shown to participants.

The same information is also mentioned in **Table 1**.

The number of days is not explicitly depicted on the screen but participants can easily infer it by counting the number of choice phases after the last occurrence of the forest phase.

In the end participants provided ratings on how strongly they explicitly counted. This information is included in the supplementary behavioral results (IQ and questionnaire):

We also asked participants (on a scale from 1 = never to 4 = always) whether they (f) actively counted the number of past days (2.3 ± 1.0) and whether they (g) were aware whether the current weather type was good or bad (3.0 ± 0.7).

- C) P20 L357 “conflict-related brain activity” a bit empty phrase. Also, I don't think the conclusion of parallel rather than interrelated can be proven.**

In overall agreement with the reviewer, we do not use the word “conflict” and also refrain from drawing the strong conclusion of parallel computation.

D) How does the pregenual cingulate/vmPFC effect compare to McGuire and Kable's Nat Neuro paper, as they had a simple sequential paradigm and argued for positive value effects in these regions.

Thank you very much for pointing us to this interesting paper. We think that there is potentially an interesting relation between the perigenual ACC/VMPFC effects in the two studies, which we now discuss. Page 23:

Intriguingly, a positive relationship between the value of the waiting action with BOLD signals in the perigenual ACC and the VMPFC may also accord with the role of a similar region in a conceptually different sequential decision-making task: In the task used by McGuire and Kable²⁹ participants had to adaptively decide how long to keep waiting for future rewards with different, uncertain timings. The temporal unfolding of the subjective value during different waiting periods was related to the VMPFC. In both tasks, "waiting" (either as a discrete action as in our task or as temporal persistence as in the task by McGuire and Kable) trades off current (opportunity) costs against potential gains in the future. The choice uncertainty of the heuristic variable was also related to BOLD signals in an anterior part of the VMPFC, as well as dorsal MPFC and IFG.

E) Both optimal policy positive signals as the other effect are as much in the ACC as other, yet one is called MPFC and other ACC in figure 5, although this is better in the figure legends.

Our revised analyses show a more refined pattern of fMRI results. We now additionally included the choice uncertainty of the optimal policy as a parametric modulator in the GLM. This choice uncertainty scales (negatively) with almost the same region that we previously identified for the optimal policy itself. This region is slightly more dorsal than the region that we previously identified but still extends into the ACC. Please see our response above to comment 3, where we pasted parts of the fMRI results section below as well as **Figs 4 & 5**.

F) As a general note, the authors should consider larger delays between events. As it is, it is impossible to dissociate some of the stages temporally and overall the timing was rather crowded. For their current analyses it is ultimately not too bad, as the important decision event has information that isn't presented before, but other analyses would have been possible with a bit more generous spacing.

We thank the reviewer for this comment and will take this into consideration in our next study. Here, we specifically aimed at looking into the choice phase.

G) Participants are strangely bad (Figure 2). Even when they should be at 100% in bad weather for foraging they are essentially random. Is this because the overall p survival is so low that they don't care? Might be useful to plot the p survival for good and bad weather by energy state as the temperature might be effected by the value and fMRI behavioural data might have a higher temperature accentuating that modulation even further.

We now provide plots that show participants' choices split up according to energy state and the probability of foraging success as well as according to energy state and weather type (see **Fig S5** pasted above in response to comment 10). We also provide a plot below that shows the mean value difference (foraging minus waiting) of the optimal policy across all trials of the fMRI sample. (This value difference corresponds to starvation probability.) The plot shows that the average value in the bad weather in energy state 1 is above 1, i.e., prescribing foraging.

We deem it likely that the “strangely bad” behavior shows participants’ overreliance on the heuristic. In the cases the reviewer refers to, participants are above chance but clearly not close to the normative choice, which would be to choose foraging with 100%. The same plots also show that unfortunately there were a limited number of trials within the relevant condition (bad weather & energy state 1). This was actually intended by our stimulus selection: We aimed at reducing the number of trials in which participants were starved (because participants could not provide answers in these trials). Nevertheless, we conducted additional analyses to test whether energy state additionally explained participants’ behavior. Overall, we did not find evidence that choice data were consistently and decisively better explained by the energy state. We now suggest energy state as an interesting variable for follow-up studies. Page 21:

We deem it an interesting question for follow-up research whether a different selection of trials or variations of our task design would result in more (or less) complex models being identified. For example, it could well be that more challenging tasks would lead participants to abandon the optimal policy in favor of a combination of two heuristics (such as a combination of the momentary probabilities of foraging success and the current energy state).

- H) How likely is the heuristic strategy to give higher p-choice estimates than the optimal? Furthermore, is, after fitting the softmax etc one strategy more likely to be more certain the majority of trials? If so, then the unsigned difference converges with a signed variant and it is therefore important to know the degree of correlation between signed and unsigned within every subject.**

Across participants, the mean proportion of trials in which the choice probability according to the probability of foraging success was higher than the choice probability according to the optimal policy was 0.476 (SD 0.018). The mean correlation of signed and unsigned differences was 0.057 (SD 0.085, minimum: -0.091; maximum: 0.321). Thus, there is no evidence that one of the two strategies is overall more certain and the signed and unsigned versions do not converge.

- I) They claim that macroscopically different brain regions encode the two policy but never show the respective other policies effect size in the regions they are referring to making it hard to judge that statement P17 L288-9.**

We agree that we do not directly compare effect sizes between regions. This is actually not aim to show that one region but clearly NOT another one is involved in computing a given variable. We therefore drop the unqualified and misleading sentence.

- J) The abstract has an incomplete sentence. It says “resort to easy-to-heuristics”**

We corrected the sentence to
To avoid such complex computations, decision-makers may resort to easy-to-compute heuristics that approximate optimal solutions.

Reviewers' comments:

Reviewer #1 (Remarks to the Author):

Korn and Bach have completed an extensive revision of their already impressive study, which features a detailed analysis of choice, RT and prefrontal BOLD behavior in a very complicated sequential decision task. I think this is a very thought provoking article well worth publishing even as is; though I do also continue to have some reservations about the analyses in their current form and so I offer some suggestions that I hope might improve the manuscript further should additional revision be required.

1) The main thing I still find ultimately unsatisfying about the article is that while there are extensive and convincing model fitting and comparison exercises, the "posterior predictive checks" don't really serve the role of helping the reader understand what about the task and the choices actually allows the models to be distinguished. I would particularly call the authors attention to the recent Palmenteri TICS paper ("The importance of falsification") for a better sense what I feel is missing here. Figure S2 shows choice behavior is monotonic in the model predictors (and S4 that it is not monotonic in some of the other candidates); and S5 shows a lot of behavior, but still I couldn't point to what about the choices tells me that these are the right models and, in particular, the other models are falsified.

Specifically, the authors argue that part of behavior is determined by a myopic heuristic that considers only choice features of the current trial. What is unique about this heuristic relative to other obvious myopic ones like EV is that the subjects pay attention to the probability of success but ignore the amount. (By the way, see Venkatraman's 2008 Neuron and 2014 Organizational Behavior papers for a seemingly similar heuristic.) So: Can we see examples of particular situations where subjects tend to make a choice that is predicted by the heuristic but for which the opposite choice is predicted by EV?

Conversely, the model fits seem to indicate that other parts of the behavior are best explained by optimal sequential choice to the exact horizon of 5, rather than some other horizon. I find it pretty surprising that different horizons can be distinguished from one another, to be honest. What is it about the choices that reveals this consideration of future trials (let alone exactly the remaining set of them)? The intuition on p.5 of the supplement doesn't shed light on this. (It describes a myopic policy with the caveat that in some cases you should be indifferent -- is selective indifference ultimately all that's driving the effect?) Again, I'd ideally like to see some feature of choices in particular situations that shows subjects paying attention to the rest of the days in the forest. Maybe this is hidden in Figure S5 but it would really make the paper more convincing and more informative if this could be exposed better.

2) One thing I find somewhat confusing about the fMRI results is that the key value-like regressors appear to be expressed in terms of value of foraging, whereas much has considered mPFC value in terms of the value of chosen (or unchosen) options. Indeed, at one point the discussion appears to suggest reinterpreting activity in these terms. I wonder if it might clarify some of the results to directly examine the data in this frame. (ie try

regressors for the heuristic or optimal value of the chosen or chosen minus unchosen action rather than the forage action – I think this is not the same as including choice itself as an additional nuisance variable.) It has been argued (e.g. by Rushworth) that a default vs non-default frame actually does better explain activity in some areas and in tasks like this, so the current analysis may well be best, but chosen value might be worth a look.

Reviewer #2 (Remarks to the Author):

Korn and Bach have revised their manuscript substantially and gone to great lengths to allay my concerns regarding their previous draft. There is no doubt in my mind that this work provides useful insights into an interesting and important problem, namely, how do people make complex decisions?

Despite the improvements, I still have one concern with the manuscript. The concern is that I think the hybrid model is still missing a key aspect of the subject behavior. For lack of better terminology, subjects seem less willing than the best-fitting model to select actions that lead to imminent and immediate death. This is quite clear from figure S5 panels A-D; in all cases the model fits all data-points except those where the energy state is equal to 1. The energy state=1 data include substantially more foraging than would be predicted by the hybrid model, particularly when the $p(\text{foraging success})$ is low. My understanding of this is that subjects do not want to “wait” under these circumstances, as this action would lead with absolute certainty to 0 energy. No doubt, such a heuristic would be a useful one in this task, and I understand the authors’ insinuation that such a heuristic might only be in play for a very small number of trials, but it certainly suggests that the behavioral model used by the authors is incomplete. While in some cases, incomplete behavioral modeling can have dire consequences (CF. Nassar & Gold 2013) this does strike me as a case where an additional mixture component is unlikely to interfere with the primary claims in the paper. Nonetheless, the computational interpretations of the behavior and imaging data would be a bit more convincing if the authors could verify the robustness of their basic modeling assertions to the inclusion of this sort of simple binary and short-term “death avoidance” heuristic.

Minor points:

Figure 2C:

It would be useful if the authors could provide more description of exactly what they are plotting. I am assuming that the light points are simulated model data though I don't see them labeled anywhere.

Figure 5B:

These should be for optimal policy, right?

Nassar, M.R. & Gold, J.I., 2013. A healthy fear of the unknown: perspectives on the interpretation of parameter fits from computational models in neuroscience. PLoS Computational Biology, 9(4), p.e1003015.

Reviewer #3 (Remarks to the Author):

1) I did not mean to imply that the authors use of the word value is improper nor that probability of surviving isn't an appropriate value, just that it represents not a signal that scales with the magnitude of a desired outcome but rather an increase in certainty of getting a reward at an end of a chain, which is somewhat different from most other tasks. I simply felt the reader might be interested in knowing this distinction and am happy with the authors changes.

2) Fine. Interesting RT analyses showing heuristics are faster.

3) Figure 5 the figure labels are a little bit confusing and could maybe be made somewhat simpler. The figure legend doesn't split up into ABCD, which might also clarify what is shown exactly. Furthermore, I think that B in Figure 5 is wrongly labeled. Isn't it supposed to be optimal policy uncertainty (negative)?

4) Fine.

5) I apologize for the confusion. The important point here is the direction of the effect! If I understand the authors correctly, both heuristic and optimal choice uncertainty (Figure 5 A+B) where they appear to exert an in MPFC effect, they do so negatively, i.e. increased activity with increased certainty (often closely related to increased subjective value). However, previous uncertainty findings in other tasks normally have an increased signal with lower certainty, i.e. higher uncertainty. Sometimes it reads in the manuscript as if the paper found what everybody else has as there are uncertainty effects but if I am not mistaken the sign is the other way around, at least for broader ACC. This is exciting as it suggests increased signals here with certainty, inconsistent with simple conflict accounts, as those would always predict positive effects of uncertainty, not certainty. To put it simply, the signal isn't heuristic or optimal difficulty, but easiness according to interpretations of previous papers in the field regarding signal signs.

6) Fine.

7) Even from the plots it looks like the p foraging success is the strongest regressor other than optimal policy (h-1). I didn't expect all factors to have the same effect size, but it should be mentioned somewhere in the text that p foraging success was the strongest single predictor, not just behaviourally but also by design in terms of driving the optimal

value.

8) Fine

9) Fine. It's a little bit of the pity that the interactive model doesn't any longer work, but maybe in a future study with a design more optimized to answer this interaction question, the authors might be able to show such an effect.

10) Fine

11) Fine

A) Fine

B) Fine. I would also mention it somewhere in the main text, e.g. methods.

C) Fine

D) Fine.

E) Interesting. However, doesn't this positive value signal contradict McGuire and Kable's conclusion of no value or decision signals in ACC? Maybe highlight this distinction when discussing McGuire and Kable. They argued all value signals are exclusively in vmPFC.

F) Fine

G) I highlighted the feature in the data precisely as it should be relatively obvious to the participants what to do. At energy state one in bad weather participants should always forage because otherwise they definitely starve. Despite that participants only forage half the time, which I was surprised by. I however understand the explanation of the authors that precisely that bin is under-sampled and just a little bit of randomness can depress percentages. I am however not so sure about the heuristics explanation, precisely because it should be so obviously the correct choice. Either way, as long as the authors mention the sampling, I am happy.

H) Fine.

I) Fine

J) Fine

Additional Comment:

Figure S9 I am not sure whether I am just misunderstanding but foraging < waiting is the same thing mathematically as waiting > foraging, unless I misunderstand. I imagine it is only a typo and one is foraging more than waiting and the other one waiting more than

foraging? Would be interesting to know what way around it is. Does on average foraging create higher activity in dACC or waiting?

Reviewer #1 (Remarks to the Author):

Korn and Bach have completed an extensive revision of their already impressive study, which features a detailed analysis of choice, RT and prefrontal BOLD behavior in a very complicated sequential decision task. I think this is a very thought provoking article well worth publishing even as is; though I do also continue to have some reservations about the analyses in their current form and so I offer some suggestions that I hope might improve the manuscript further should additional revision be required.

Thank you very much for your favorable evaluation of our manuscript and for your helpful suggestions. We have taken care to implement your recommendations, which we believe has improved the manuscript considerably.

1) The main thing I still find ultimately unsatisfying about the article is that while there are extensive and convincing model fitting and comparison exercises, the “posterior predictive checks” don’t really serve the role of helping the reader understand what about the task and the choices actually allows the models to be distinguished. I would particularly call the authors attention to the recent Palminteri TICS paper (“The importance of falsification”) for a better sense what I feel is missing here. Figure S2 shows choice behavior is monotonic in the model predictors (and S4 that it is not monotonic in some of the other candidates); and S5 shows a lot of behavior, but still I couldn’t point to what about the choices tells me that these are the right models and, in particular, the other models are falsified.

Thank you for prompting us to clarify the “posterior predictive checks” and for pointing us to the recent article by Palminteri et al (which we now cite). We realized that in our previous submission we only provided plots showing that the *winning model does capture behavior* when data are binned according to all of the different candidate variables (these are presented in the figures mentioned by the reviewer). We now added plots showing that *alternative models do not capture behavior*. In our opinion, these plots nicely illustrate the lack of fit of these models, thereby falsifying them for our setup and data. The differences are particularly striking when comparing all models with a single predictor (see panel **A** below). Many of the models do not make sensible predictions at all. The new illustrations also demonstrate clearly that the inclusion of the optimal policy (horizon-5) considerably improves the fit to the empirical choice data (see panels **B & C** below).

Fig S4. fMRI sample: Posterior predictive checks: Comparison of choice data to different model predictions

- (A)** Posterior predictive checks show that—among the models with a single predictor—the model comprising the probability of foraging success captures choice data better than any of the other models.
- (B)** Posterior predictive checks show that—among the models with a two predictors—the model comprising the probability of foraging success and the optimal policy (at a horizon of 5 time steps) captures choice data better than any of the other models. Overall most models make quite similar predictions since they all include the probability of foraging success.
- (C)** Posterior predictive checks show that the model comprising both the probability of foraging success and the optimal policy (horizon-5) provides a better fit to the data than the model that only comprises the probability of foraging success

Error bars are SEM. Per data bin, circles depict mean empirical data points and colored lines and crosses depict mean model predictions (averaged for simulated data according to each participant’s model fit). In several cases, error bars are smaller than the marker sizes, which scale with the average number of trials contributing to the respective data points. See **Fig S5** for the same plots with data of the behavioral sample. See **Table 1** for a list of all variables. h-5: horizon of 5 days; h-1: horizon of 1 day; cont.: continuous; bin.: binary

The same was true for the behavioral sample:

Fig S5. Behavioral sample: Posterior predictive checks: Comparison of choice data to different model predictions

The plots follow the same logic as those in **Fig S4**, which show data and models of the fMRI sample. Error bars are SEM. Per data bin, circles depict mean empirical data points and colored lines depict mean model predictions (averaged for simulated data according to each participant's model fit). In several cases, error bars are smaller than the marker sizes, which scale with the average number of trials contributing to the respective data points. See **Table 1** for a list of all variables. h-5: horizon of 5 days; h-1: horizon of 1 day; cont.: continuous; bin.: binary

Of course, we refer to these illustrations in the results sections:

Page 9:

Posterior predictive checks demonstrate that, also qualitatively, no other model with a single predictor captured choice data as well as the model with the probability of foraging success (**Figs S4A & Fig S5A**), including models relying on the optimal policy or on a heuristic based on expected value (see **Supplementary results**).

Page 12:

Posterior predictive checks confirm that the winning model qualitatively captured participants' behavior (see **Figs 3C & 3D** for the winning model, and **Figs S4, S6, & S8-S10** for extended posterior predictive checks of all models and for parameter estimates of a full model including all candidate variables).

We have also revised the organization and the captions of the figures mentioned by the reviewer (which are now **Fig S4-S4**) to provide a better description of what is depicted.

Specifically, the authors argue that part of behavior is determined by a myopic heuristic that considers only choice features of the current trial. What is unique about this heuristic relative to other obvious myopic ones like EV is that the subjects pay attention to the probability of success but ignore the amount. (By the way, see Venkatraman's 2008 Neuron and 2014 Organizational Behavior papers for a seemingly similar heuristic.) So: Can we see examples of particular situations where subjects tend to make a choice that is predicted by the heuristic but for which the opposite choice is predicted by EV?

Thank you for this specific suggestion. We now selected trials in which the probability of foraging success model and the EV model made opposing predictions and then tested empirical choice in these conditions. These analyses corroborated that the probability of foraging success is by far the better predictor.

These results are now described in detail (**SI results**: page **SI 15**)

Comparison of choice data according to prescriptions of the models comprising probability of foraging success or expected values

We specifically tested behavior in situations in which probability of foraging success and EV made opposing prescriptions. We chose these two metrics in particular since they are mathematically related and since previous studies have investigated how probability and EV account for choices in another type of sequential decision-making task^{7,8}. We selected two types of trials: First, trials in which the heuristic of using the probability of foraging success prescribed foraging (i.e., model prescription > 0.5) and at the same time the heuristic of using EV prescribed waiting (i.e., model prescription < 0.5; based on logistic functions derived from mean parameter estimates from the behavioral sample). Second, trials in which the probability of foraging success prescribed waiting and the EV prescribed foraging. In both cases participants' actual choices were aligned with the prescriptions of the model including the probability of foraging success: In the first type of trials, participants were more likely to choose foraging (proportion of trials foraging chosen: mean \pm SD: 0.73 ± 0.15 , t-test against 0.5: $t(27) = 8.1$, $p < 10^{-7}$; percentage of these trials in the overall number of trials: 0.12 ± 0.01). In the second type of trials, participants chose waiting more often (proportion of trials foraging chosen: 0.11 ± 0.07 , t-test against 0.5: $t(27) = -25.0$, $p < 10^{-19}$; percentage of these trials in the overall number of trials: 0.27 ± 0.02 ; see **Table S5** for analogous analyses comparing different horizons of the optimal policy).

SI methods (pages SI 10-11)

For additional analyses, we used the relevant logistic functions derived from mean parameter estimates (from the behavioral sample) to select trials in which the heuristic policies of using the probability of foraging success and of using the EV made opposite prescriptions for choice. That is, we binarized the prescriptions of the two policies by splitting them into those above and below the midpoint of 0.5. Logistic functions were derived from the mean parameter estimates of the behavioral sample. See **Supplementary behavioral results and discussion: Comparison of choice data according to prescriptions of the models comprising probability of foraging success or expected values**).

We refer to this section in the main results (page 9):

Posterior predictive checks demonstrate that, also qualitatively, no other model with a single predictor captured choice data as well as the model with the probability of foraging success (**Figs S4A & Fig S5A**), including models relying on the optimal policy or on a heuristic based on expected value (see **Supplementary results**).

We highlight this result and the relation to the two studies by Venkatraman et al. in the discussion (page 21):

Specifically, we demonstrate that participants took advantage of a model-free heuristic available at the time point of decision-making. This heuristic of relying on the probability of foraging success approximates the optimal policy, which is hard to compute, and performed overall best in explaining choices among a large set of alternatives (including the foraging options' expected values). This finding is in line with studies showing that participants base their choices to a large degree on the overall probability of winning in another type of sequential decision-making task^{16,28}.

Conversely, the model fits seem to indicate that other parts of the behavior are best explained by optimal sequential choice to the exact horizon of 5, rather than some other horizon. I find it pretty surprising that different horizons can be distinguished from one another, to be honest. What is it about the choices that reveals this consideration of future trials (let alone exactly the remaining set of them)? The intuition on p.5 of the supplement doesn't shed light on this. (It describes a myopic policy with the caveat that in some cases you should be indifferent -- is selective indifference ultimately all that's driving the effect?) Again, I'd ideally like to see some feature of choices in particular situations that shows subjects paying attention to the rest of the days in the forest. Maybe this is hidden in Figure S5 but it would really make the paper more convincing and more informative if this could be exposed better.

To address this insightful point by the reviewer, we followed a similar approach as described above for the comparison of p foraging success and EV. We selected trials in which the winning optimal horizon-5 policy made opposing prescriptions with respect to the optimal policies with the other considered horizons (i.e., one horizon prescribed foraging and another prescribed waiting or vice versa). Again, in all of these conditions choices were in line with the horizon-5 policy. Empirical data were never better predicted by a horizon other than horizon 5. This also shows that selective indifference is *not* all that is driving the effect, since these trials were not considered in the selected subsets.

We summarized these results in **Table S5**:

Table S5. Both samples: Analyses of choice data in subsets of trials with opposing prescriptions according to different time horizons of the optimal policy

Participants' choices follow the optimal policy with a horizon of 5 time steps, which is normative in our task. That is, in all subsets of trials, in which different shorter horizons of the optimal policy made opposing prescriptions, choices were in line with the prescriptions of the optimal policy with a horizon of 5 steps.

Condition	Mean (SD) proportion of participants' choices to forage in respective conditions	t-values (test against midpoint of choice proportion, i.e., 0.5)	p-values	Mean (SD) proportion of trials in respective condition versus total number of trials
fMRI sample				
Foraging prescribed by horizon-5 optimal policy & waiting prescribed by ...				
horizon-1 policy	0.94 (0.06)	38.45	$< 10^{-24}$	0.14 (0.012)
horizon-2 policy	0.79 (0.07)	21.57	$< 10^{-17}$	0.24 (0.014)
horizon-3 policy	0.82 (0.11)	15.79	$< 10^{-14}$	0.11 (0.010)
horizon-4 policy	0.71 (0.21)	5.23	$< 10^{-4}$	0.03 (0.007)
Waiting prescribed by horizon-5 optimal policy & foraging prescribed by ...				
horizon-3 policy	0.20 (0.22)	-7.26	$< 10^{-7}$	0.02 (0.007)
horizon-4 policy	0.26 (0.22)	-5.96	$< 10^{-5}$	0.02 (0.006)
Behavioral sample				
Foraging prescribed by horizon-5 optimal policy & waiting prescribed by ...				
horizon-1 policy	0.92 (0.17)	11.58	$< 10^{-9}$	0.14 (0.019)
horizon-2 policy	0.77 (0.15)	8.13	$< 10^{-7}$	0.24 (0.015)
horizon-3 policy	0.79 (0.20)	6.52	$< 10^{-5}$	0.12 (0.012)
horizon-4 policy	0.75 (0.24)	4.83	$< 10^{-3}$	0.03 (0.010)
Waiting prescribed by horizon-5 optimal policy & foraging prescribed by ...				
horizon-3 policy	0.19 (0.25)	-5.60	$< 10^{-5}$	0.02 (0.005)
horizon-4 policy	0.39 (0.25)	-1.98	= 0.062	0.02 (0.005)

See **Fig S11** for an illustration of the differential prescriptions of the optimal policy according to different time horizons and **Fig S12** for formal model comparisons of choice data.

Additionally, we now provide a graphical illustration of how often the optimal policies according to the different horizons prescribe foraging or waiting, or are indifferent between these two options for the trials used in our setup (**Fig S11**). As the reviewer notes, indifference decreases with longer time horizons but also the percentage of prescribed waiting changes (especially for horizons 2 & 3).

Fig S11. Prescriptions of optimal policy according to different time horizons

This plot illustrates that the optimal policy makes differential prescriptions according to the time horizon employed. In the trials used in our task, the optimal policy at shorter time horizons is often indifferent between the two choice options of waiting versus foraging (e.g., when starvation is not possible on the next time step under any of the two options, a myopic optimal policy with a horizon of only 1 time step is indifferent).

SI Methods (page SI 10):

To analyze subsets of trials, in which different horizons of the optimal policy made opposing prescriptions, we simply selected trials in which the (*a priori* computable) optimal policy for a horizon of 5 time steps prescribed foraging and the optimal policy for another time horizon prescribed waiting; or vice versa (see **Table S5 & Fig S11**).

We refer to these analyses in the results section of the main text (pages 12-13):

When comparing models with different horizons, we found that participants' choices were indeed best described by a time-horizon of five days (see **Fig S11** for an illustration of the different prescriptions made by optimal policies with different time horizons and **Fig S12** for model comparisons). This finding was corroborated when analyzing subsets of trials in which policies with different horizons made opposing prescriptions (**Table S5**).

2) One thing I find somewhat confusing about the fMRI results is that the key value-like regressors appear to be expressed in terms of value of foraging, whereas much has considered mPFC value in terms of the value of chosen (or unchosen) options. Indeed, at one point the discussion appears to suggest reinterpreting activity in these terms. I wonder if it might clarify some of the results to directly examine the data in this frame. (ie try regressors for the heuristic or optimal value of the chosen or chosen minus unchosen action rather than the forage action – I think this is not the same as including choice itself as an additional nuisance variable.) It has been argued (e.g. by Rushworth) that a default vs non-default frame actually does better explain activity in some areas and in tasks like this, so the current analysis may well be best, but chosen value might be worth a look.

We have now run an additional GLM along the suggestions of the reviewer and find indeed expected regions for the regressors related to the chosen action. Specifically, these analysis show that the chosen value according to the optimal policy was positively related to a region in the VMPFC

(extending into perigenual ACC), similar to the value differences according to the optimal policy. For the heuristic, analogous results were obtained for the left frontal pole. We now describe these findings in the main text and add the relevant figures to the **SI**. These results corroborate that BOLD signals in these regions were related to variables, on which participants based their choice.

Main results section (page 19):

In a fourth GLM, we specifically analyzed how the values of the chosen options related to BOLD signals (**Fig S16 & Table S10**). The probabilities of foraging success according to the options chosen by participants scaled positively with the left frontal pole, in a similar region as described above for the probabilities of foraging success according to the presented foraging options. The values of optimal policy according to the chosen options' values were positively related to the VMPFC, in a similar region as described above for the value differences according to the optimal policy. These findings corroborate that BOLD signals in the left frontal pole and in the VMPFC were associated with variables on which participants based their choices.

SI Methods section (page **SI 13**):

A fourth GLM included parametric modulators in terms of chosen (and unchosen) options (and not in terms of the presented options as in the first three GLMs). Specifically, this fourth GLM contained the a parametric modulator for the chosen values of the employed heuristic (i.e., the current probability of foraging success for trials in which the foraging option was chosen and a probability of zero when the waiting option was chosen) and a parametric modulator for the chosen values according to the optimal policy (i.e., the current values of the foraging or the waiting options) as well as parametric modulators for the corresponding unchosen values along with parametric modulators for participants' choices and RTs.

Table S10 (page **SI 62**) provides details on all clusters.

Fig S16

Fig S16. Statistical parametric maps for the choice phase (GLM with values of chosen options as parametric modulator)

Depicted are results from a separate GLM, in which parametric modulators were framed in terms of the values of the chosen options (rather than in terms of the options presented). **(A)** The probabilities of foraging success according to the chosen options (i.e., foraging or waiting) were positively related to the left frontal pole (in a similar region as the probabilities of foraging framed in terms of the presented foraging option in the main GLM). **(B)** The values of the optimal policy according to the chosen options showed a positive relation with the VMPFC (in a similar region as the differences between the presented choice options' values according to the optimal policy in the main GLM).

Overlay on group average T1-weighted image in MNI space; clusters are whole-brain FWE corrected for multiple comparisons at $p < 0.05$ with a cluster-defining threshold of $p < 0.001$. See **Table S10** for a list of all clusters.

Reviewer #2 (Remarks to the Author):

Korn and Bach have revised their manuscript substantially and gone to great lengths to allay my concerns regarding their previous draft. There is no doubt in my mind that this work provides useful insights into an interesting and important problem, namely, how do people make complex decisions?

Thank you very much for your evaluation.

Despite the improvements, I still one concern with the manuscript. The concern is that I think the hybrid model is still missing a key aspect of the subject behavior. For lack of better terminology, subjects seem less willing than the best-fitting model to select actions that lead to imminent and immediate death. This is quite clear from figure S5 panels A-D; in all cases the model fits all data-points except those where the energy state is equal to 1. The energy state=1 data include substantially more foraging than would be predicted by the hybrid model, particularly when the p(foraging success) is low. My understanding of this is that subjects do not want to “wait” under these circumstances, as this action would lead with absolute certainty to 0 energy. No doubt, such a heuristic would be a useful one in this task, and I understand the authors’ insinuation that such a heuristic might only be in play for a very small number of trials, but it certainly suggests that the behavioral model used by the authors is incomplete. While in some cases, incomplete behavioral modeling can have dire consequences (CF. Nassar & Gold 2013) this does strike me as a case where an additional mixture component is unlikely to interfere with the primary claims in the paper. Nonetheless, the computational interpretations of the behavior and imaging data would be a bit more convincing if the authors could verify the robustness of their basic modeling assertions to the inclusion of this sort of simple binary and short-term “death avoidance” heuristic.

We thank the reviewer for this insightful suggestion to further look into how the current energy state affects participants’ behavior. We deem the reviewer’s suggestion of a “simple binary and short-term death avoidance heuristic” very valuable and have therefore extended all relevant model comparisons to include this variable (i.e., we now compare a total of 11 candidate decision variables). This “binary energy state heuristic” is set to 1 when the “continuous energy state” is one and is set to 0 for energy states of two or higher. As the reviewer notes this heuristic captures the fact that waiting leads to sure death in energy state one. We would like to stress that this “binary energy state heuristic” is thus always in line with the optimal policy (since the optimal policy always prescribes foraging when only one energy point is left). But in contrast to the optimal policy, the binary energy state heuristic can by design not make specific prescriptions in other energy states (which is also reflected in the fact that it performs poorly in minimizing overall starvation probabilities in our simulations, see updated **Fig S13**).

In the fMRI sample (n=28), results remain unchanged and robust to the addition of the binary energy state heuristic. In a comparison of models comprising three decision variables (i.e., p foraging success, optimal policy at a horizon of 5 days, and one of the remaining 9 variables), the model including the binary energy state heuristic only performs fourth-best according to relative log-Bayes factors and third-best according to protected exceedance probabilities (**Fig 3 & Tables S1 & S3**). As the reviewer mentions, the small number of trials in which participants were in energy state one (percentage: mean \pm SD: 0.07 \pm 0.01) probably limits the power to detect a strong influence of this binary energy state heuristic. Additionally, it might be that—since the situation of energy state one occurs so rarely—participants still apply in many cases the “p foraging success heuristic” and are swayed to wait by small probabilities of foraging success (although this is clearly suboptimal). This might explain why participants only chose foraging in well below 80% of these cases and not in 100% as they should.

In the smaller behavioral sample (n=21), results as indicated by log-group Bayes factors still decisively favored the model comprising p foraging success and the optimal policy at a horizon of 5 days but protected exceedance probabilities were indecisive between this model and the model comprising p foraging success and binary energy state (**Fig S3 & Tables S2 & S4**). Binary energy state also emerged as the best third predictor according to log-group Bayes factors.

Taken together, the combined evidence from both samples shows that our results are robust to the inclusion of a “binary energy state heuristic” but the fact that it performed second-best to the optimal policy in the smaller behavioral sample suggests that this heuristic is an important addition to the model space.

In line with the reviewer’s comment, we provide a graphical visualization of the model including the binary energy state variable. Qualitatively predictions improved for energy state one.

Fig S10. Both samples: Comparison of choice data to model predictions when splitting conditions for probability of foraging success and continuous energy.

These plots separate data and model predictions according to two task variables: Probability of foraging success (abscissa) and continuous energy state (color-coded). The plots illustrate the quality of fit for the choice model, which includes the probability of foraging success and the optimal policy at a horizon of 5 time steps (left panels **A & C**). This choice model captures the data well. Notable deviations of data and model fit are only observed for conditions with an energy state of one (and a rather low probability of foraging success). In our current study design, the data bin with energy state one only comprised a small fraction of trials (percentage: mean \pm SD: fMRI sample: 0.07 ± 0.01 ; behavioral sample: 0.08 ± 0.02). For additional inspection, we provide plots for a choice model that includes the “binary energy state variable” (right panels **B & D**). The more complicated model provides a better qualitative fit to the data bins with an energy state of one. However, the simpler model provides a decisively better model fit in the fMRI sample (**Tables S1 & S3**). In the behavioral sample, the model including binary energy state is decisively better according to log-group Bayes factors but not according to protected exceedance probabilities (**Tables S2 & 4**). Per data bin, mean empirical data points are depicted as circles (size according to number of data points) and mean model predictions are depicted as dashed lines.

To reflect these additions, we have expanded the results section (page 11):

The optimal policy explained additional variance in participants' choices

After accounting for the best heuristic, a decisive proportion of remaining variance in the (larger) fMRI sample was explained by the *a priori* optimal policy with a horizon of five time steps (**Fig 3B, Table S1**). This model with two predictors also won in an extensive model comparison against all 55 pairs of candidate variables (**Table S2**). Posterior predictive checks confirm that the winning model qualitatively captured participants' behavior (see **Figs 3C & 3D** for the winning model, and **Figs S4, S6, & S8-S10** for extended posterior predictive checks of all models and for parameter estimates of a full model including all candidate variables). Additionally, models including interactions between the most important heuristics did not provide a better fit than the model with the optimal policy (**Table S1**). There was no decisive evidence (according to protected exceedance probabilities) that a model with three variables explained choices better than the best model with two variables (**Table S1**).

In the (smaller) behavioral sample, we found that protected exceedance probabilities did not decisively distinguish between two-variable models including the optimal policy, or the binary energy state heuristic, respectively. However, log-group Bayes factors provided decisive evidence for the same model as in the fMRI sample, including the best heuristic and the optimal policy (**Tables S2 & S4, Figs S3, S5, & S7**). Thus, overall our model comparisons across the two groups decisively favor the optimal policy as a predictor of participants' choice in our task. Nevertheless, it is possible that on specific subsets of trial types (e.g. with energy state one), different variables predict behavior better (see **Figs 9-10**).

Taken together, model comparisons suggested that participants' choices were often predicted by a heuristic policy, but additionally choices followed the normatively optimal policy.

We extended a cautionary note on model selection in the discussion (page 21):

We did not find decisive evidence that any linear combination of two candidate policies and variables explained our choice data better than the probability of foraging success and the optimal policy. Although it is theoretically possible that participants use a yet unknown decision policy (for discussion see^{29,30}), such policies do not follow from the given task variables in an obvious way. Any such model would thus likely require higher complexity than the linear combination of the probability of foraging success and the optimal policy (or it would only apply to a less complex setting than the one investigated here). It is an interesting question for follow-up research whether on specific trial types (possibly under-sampled here) participants may have used a more (or less) complex model. Also, it appears possible that in a more (or less) challenging task, participants may abandon the optimal policy in favor of a combination of two or more heuristics.

We added an explanation of the binary energy state variable to the methods (page **SI 8**):

As *policy*, we first considered the following potential optimal or heuristic policies (see also **Table 1** for an annotated list): [...] (6) binary energy state (indicating whether waiting would lead to sure death or not). Since waiting leads to sure death when the continuous energy state is one, the optimal policy never prescribes waiting in these situations (i.e., even when the probability of foraging success is small, the optimal policy always favors a non-zero starvation probability). Therefore, the prescriptions of the binary energy state variable are always in line with the optimal policy at a continuous energy state of one.

We updated **Table 1**, which lists all variables.

	Explanation	Theoretically possible values of this variable in the task	Example value of this variable (as in Figs 1 choice phase)	Grand mean of variable across fMRI participants (mean within participant SD)
[...]	[...]	[...]	[...]	[...]
Binary energy state	When the continuous energy state is one, waiting leads to sure death. In higher energy states, waiting will never lead to starvation. The variable “binary energy state” distinguishes between these situations (1=energy state is one; 0=energy state is two or higher).	Binary variable: 1 or 0	0 “waiting does not lead to starvation”	0.07 (0.26)
[...]	[...]	[...]	[...]	[...]

We extended the statistical tables of all choice model comparisons in **Tables S1 & S3** for the fMRI sample and in **Tables S2 & S4** for the behavioral sample. For easier reference, we summarize these results in the headers of the respective tables (and we also kept this verbal description similar for the two samples).

Table S1. fMRI sample: Comparison of choice models

Among 11 candidate variables, probability of foraging success emerged as the single best predictor of participants’ choices. The model that included both the probabilities of foraging success and the optimal policy outperformed 1) all other combinations of probability of foraging success and the remaining 10 candidate variables, 2) the model with the probabilities of foraging success as sole predictor, and 3) models including the probabilities of foraging success, along with five different heuristics, and the respective interactions. No decisive evidence emerged for 1) models with a third candidate variable consistently explaining more variance (according to protected exceedance probabilities), and 2) a clear-cut influence of choice uncertainties or discrepancy between the two policies included in the main model (according to relative log-group Bayes factors and protected exceedance probabilities).

Table S2. Behavioral sample: Comparison of choice models

Among 11 candidate variables, probability of foraging success emerged as the single best predictor of participants’ choices. The model that included both the probabilities of foraging success and the optimal policy outperformed 1) all other combinations of probability of foraging success and the remaining 10 candidate variables (according to relative log-group Bayes factors; the model including the binary energy state variable performed second-best), 2) the model with the probabilities of foraging success as sole predictor, and 3) models including the probabilities of foraging success, along with five different heuristics, and the respective interactions (according to relative log-group Bayes factors; the model including the binary energy state and the respective interaction performed second-best). No decisive evidence emerged for 1) models with a third candidate variable consistently explaining more variance (according to protected exceedance probabilities), and 2) a clear-cut influence of choice uncertainties or discrepancy between the two policies included in the main model (according to relative log-group Bayes factors and protected exceedance probabilities).

Table S3. fMRI sample: Comparison of choice models with all possible combinations of two candidate variables

Model comparison of a total of 55 models with all binary combinations of the 11 candidate decision variables. Log-group Bayes factors indicate that winning model comprises the probability of foraging success and the optimal policy with a horizon of 5 time steps (protected exceedance probabilities also favor the same model).

Table S4. Behavioral sample: Comparison of choice models with all possible combinations of two candidate variables

Model comparison of a total of 55 models with all binary combinations of the 11 candidate decision variables. Log-group Bayes factors indicate that winning model comprises the probability of foraging success and the optimal policy with a horizon of 5 time steps (protected exceedance probabilities do not decisively distinguish between this model and the model comprising the probability of foraging success and the binary energy state variable).

Minor points:

Figure 2C:

It would be useful if the authors could provide more description of exactly what they are plotting. I am assuming that the light points are simulated model data though I don't see them labeled anywhere.

Thank you. We have added the following statement in the legend of **Fig 3** (page 11, previously **Fig 2**) and analogous statements to all other relevant figures.

In the right-hand panels error bars are SEM. Per data bin, circles depict mean empirical data points and lines and crosses depict mean model predictions (averaged for simulated data according to each participant's model fit). In several cases, error bars are smaller than the circles, which scale with the average number of trials contributing to the respective data points.

Figure 5B:

These should be for optimal policy, right?

Thank you. Yes, we correct this (this is now **Fig 6B**). We carefully checked all other figures and figure legends.

Nassar, M.R. & Gold, J.I., 2013. A healthy fear of the unknown: perspectives on the interpretation of parameter fits from computational models in neuroscience. PLoS Computational Biology, 9(4), p.e1003015

Thank you for suggesting this paper, which we now cite:

Discussion (page 21)

Although it is theoretically possible that participants use a yet unknown decision policy (for discussion see^{29,30}), such policies do not follow from the given task variables in an obvious way. Any such model would thus likely require higher complexity than the linear combination of the probability of foraging success and the optimal policy (or it would only apply to a less complex setting than the one investigated here).

Reviewer #3 (Remarks to the Author):

1) I did not mean to imply that the authors use of the word value is improper nor that probability of surviving isn't an appropriate value, just that it represents not a signal that scales with the magnitude of a desired outcome but rather an increase in certainty of getting a reward at an end of a chain, which is somewhat different from most other tasks. I simply felt the reader might be interested in knowing this distinction and am happy with the authors changes.

Thank you for this clarification.

2) Fine. Interesting RT analyses showing heuristics are faster.

3) Figure 5 the figure labels are a little bit confusing and could maybe be made somewhat simpler. The figure legend doesn't split up into ABCD, which might also clarify what is shown exactly. Furthermore, I think that B in Figure 5 is wrongly labeled. Isn't it supposed to be optimal policy uncertainty (negative)?

We apologize for the incorrect labeling of **Fig 5B** in our earlier version (now **Fig 6B**). The reviewer is correct that “optimal policy uncertainty (negative)” is depicted. This is now corrected and we also carefully checked all other figures and figure legends. Following the reviewer's comment, we now split up the legends of the two fMRI figures in the main text according to all subpanels.

Fig 5 (page 17; previously **Fig 4**).

Fig 5. Statistical parametric maps for the BOLD signals related to heuristic and the optimal policies during the choice phase. **(A)** The probability of foraging success, the employed heuristic policy, showed a positive relation with BOLD signals in DMPFC, extending into pre-SMA), in bilateral IPS, and the left frontal pole among other regions. **(B)** The probability of foraging success showed a negative relation in the perigenual ACC extending into VMPFC. **(C)** The optimal policy showed a positive relation in perigenual ACC and mid-cingulate. Overlay on group average T1-weighted image in MNI space; clusters are whole-brain family-wise error (FWE) corrected for multiple comparisons at $p < 0.05$ with a cluster-defining threshold of $p < 0.001$. See **Table S7** for a list of all clusters.

Fig 6 (page 20; previously Fig 5).

Fig 6. Statistical parametric maps for the respective uncertainties of heuristic and optimal policies, the discrepancies in their choice probabilities, and log-transformed RTs during the choice phase. **(A)** Choice uncertainty of the heuristic (i.e., probability of foraging success) showed a negative relation with BOLD signals in VMPFC, DMPFC, IFG, and the posterior cingulate cortex, among other regions. **(B)** Choice uncertainty of the optimal policy exhibited a negative correlation with DMPFC/ACC and IFG. **(C)** Discrepancies between the two policies showed a positive relation with DMPFC (extending into pre-SMA and ACC), bilateral dorsal striatum, and bilateral IFG. **(D)** For completeness, correlations with log-transformed RTs are depicted. Overlay on group average T1-weighted image in MNI space; clusters are whole-brain FWE corrected for multiple comparisons at $p < 0.05$ with a cluster-defining threshold of $p < 0.001$. See **Table S7** for a list of all clusters (as well as **Tables S8-10 & Figs S15-S16** for further analyses of the choice phase). See **Fig S17 & Table S11** for BOLD signals during the outcome phase.

4) Fine.

5) I apologize for the confusion. The important point here is the direction of the effect! If I understand the authors correctly, both heuristic and optimal choice uncertainty (Figure 5 A+B) where they appear to exert an in MPFC effect, the do so negatively, i.e. increased activity with increased certainty (often closely related to increased subjective value). However, previous uncertainty findings in other tasks normally have an increased signal with lower certainty, i.e.

higher uncertainty. Sometimes it reads in the manuscript as if the paper found what everybody else has as there are uncertainty effects but If I am not mistaken the sign is the other way around, at least for broader ACC. This is exciting as it suggests increased signals here with certainty, inconsistent with simple conflict accounts, as those would always predict positive effects of uncertainty, not certainty. To put it simply, the signal isn't heuristic or optimal difficulty, but easiness according to interpretations of previous papers in the field regarding signal signs.

We thank the reviewer for clarifying this point. We now highlight the direction of the effect (and relate it to some of our previous findings showing a similar relationship in the posterior cingulate cortex with second-order uncertainty; Bach et al., 2011, Journal of Neuroscience).

In the results section (page 18):

That is, we found increased BOLD signals with increasing choice certainty of both heuristic and optimal policy in regions of the MPFC.

In the discussion section (page 23):

Interestingly, the uncertainty of the heuristic variable was negatively (and not positively) related to BOLD signals in several regions, in particular in the posterior cingulate cortex, an anterior part of the VMPFC, dorsal MPFC, and IFG. Put differently, these regions showed a positive association with the “easiness” of making a decision according to the heuristic policy. Negative relations of an uncertainty metric have previously been identified a region of the posterior cingulate cortex, which was slightly more posterior than the cluster identified here^{12,34}.

[...]

In particular, we found that a dorsal region of the MPFC correlates negatively with the choice uncertainty of the optimal policy (i.e., positively with the “easiness” of making a decision according to the optimal policy).

6) Fine.

7) Even from the plots it looks like the p foraging success is the strongest regressor other than optimal policy (h-1). I didn't expect all factors to have the same effect size, but it should be mentioned somewhere in the text that p foraging success was the strongest single predictor, not just behaviourally but also by design in terms of driving the optimal value.

Thank you. This piece of information fits nicely into the beginning of results section that describes the all considered candidate variables (page 9). For clarity, we now provide shared variances and the relationships of the optimal policy to the heuristic policies in two separate figures (**Figs S1-S2**).

By design, the optimal policy shares some predictions with several of the considered heuristics most notably the probability of foraging success (**Figs S1-S2**). However, average shared variance, derived on a trial-by-trial basis across participants, was sufficiently low to dissociate which variables accounted for participants' decisions (**Figs S1-S2**).

8) Fine

9) Fine. It's a little bit of the pity that the interactive model doesn't any longer work, but maybe in a future study with a design more optimized to answer this interaction question, the authors might be able to show such an effect.

Thank you. We agree and we are indeed considering to run a follow-up study to address this.

10) Fine

11) Fine

A) Fine

B) Fine. I would also mention it somewhere in the main text, e.g. methods.

We now mention in the methods that participants did not explicitly see the remaining number of days within a forest (**SI methods**: page **SI 7**).

After a variable fixation interval (between 0.5 and 3.8 sec), a new day or a new forest was depicted. Participants were not explicitly cued about the current day within a forest; but they knew that they always remained a maximum of five days within a forest (so they could count down the number of days when entering a new forest).

C) Fine

D) Fine.

E) Interesting. However, doesn't this positive value signal contradict McGuire and Kable's conclusion of no value or decision signals in ACC? Maybe highlight this distinction when discussing McGuire and Kable. They argued all value signals are exclusively in vmPFC.

Thank you. We now mention this difference in the discussion (page 23):

Still, the fact that our analyses also identify regions outside the VMPFC that scale with the value of foraging (notably in the DMPFC) constitutes a difference between our findings and those by McGuire and Kable.

F) Fine

G) I highlighted the feature in the data precisely as it should be relatively obvious to the participants what to do. At energy state one in bad weather participants should always forage because otherwise they definitely starve. Despite that participants only forage half the time, which I was surprised by. I however understand the explanation of the authors that precisely that bin is under-sampled and just a little bit of randomness can depress percentages. I am however not so sure about the heuristics explanation, precisely because it should be so obviously the correct choice. Either way, as long as the authors mention the sampling, I am happy.

In response to your comment and (a comment by Reviewer 2), we additionally included a new “binary energy state heuristic” in all model comparisons. This variable is set to 1 when the “continuous energy state” is one and is set to 0 for energy states of two or higher. As the reviewer notes this heuristic captures the fact that waiting leads to sure death in energy state one. This “binary energy state heuristic” is thus always in line with the optimal policy (since the optimal policy always prescribes foraging when only one energy point is left). But in contrast to the optimal policy, the binary energy state heuristic can by design not make specific prescriptions in other energy states.

In the fMRI sample (n=28), results remain unchanged and robust to the addition of the binary energy state heuristic. In the smaller behavioral sample (n=21), results as indicated by log-group Bayes factors still decisively favored the model comprising p foraging success and the optimal policy at a horizon of 5 days but protected exceedance probabilities were indecisive between this model and the model comprising p foraging success and binary energy state.

Taken together, the combined evidence from both samples shows that our results do not support the inclusion of a “binary energy state heuristic,” which is likely due to the under-sampling of trials with an energy state of one. We have updated all relevant results and provide a graphical

visualization of a model including the “binary energy state heuristic.” In the legend of this figure (**Fig S10**; page **SI 28**) we highlight the under-sampling as suggested by the reviewer.

In our current study design, the data bin with energy state one only comprised a small fraction of trials (percentage: mean \pm SD: fMRI sample: 0.07 ± 0.01 ; behavioral sample: 0.08 ± 0.02).

H) Fine.

I) Fine

J) Fine

Additional Comment:

Figure S9 I am not sure whether I am just misunderstanding but foraging < waiting is the same thing mathematically as waiting > foraging, unless I misunderstand. I imagine it is only a typo and one is foraging more than waiting and the other one waiting more than foraging? Would be interesting to know what way around it is. Does on average foraging create higher activity in dACC or waiting?

We apologize for this typo! The upper panel in **A** is “foraging > waiting” and the lower panel in **B** is “waiting > foraging.” Thus, on average foraging creates higher activity in dACC (than waiting). The figure, which is now **Fig S15**, is corrected (page **SI 34**).

REVIEWERS' COMMENTS:

Reviewer #1 (Remarks to the Author):

Thanks for another very responsive and informative revision. The new analyses help to make what was already an extremely intriguing paper all the more illuminating.

Reviewer #2 (Remarks to the Author):

The authors have completely addressed my concerns, and I find the current draft an impressive piece of work on an interesting and timely problem in computational neuroscience.